

Impacts of future climate change on urban flood volumes in Hohhot City
in Northern China: benefits of climate mitigation and adaptations
Qianqian Zhou[1, 2], Guoyong Leng[2,*], Maoyi Huang[3]
[1]School of Civil and Transportation Engineering, Guangdong University of Technology,
Waihuan Xi Road, Guangzhou 510006, China
[2]Joint Global Change Research Institute, Pacific Northwest National Laboratory, College Park
MD 20740, USA
[3]Earth System Analysis and Modeling Group, Pacific Northwest National Laboratory, Richland,
WA 99352, USA
________________________
*Corresponding author address: Guoyong Leng, Joint Global Change Research Institute, Pacific
Northwest National Laboratory, College Park MD, 20740.
E-mail: guoyong.leng@pnnl.gov

**Abstract**

As China has become increasingly urbanised, flooding has become a regular occurrence in its major cities. Assessing the effects of future climate change on urban flood volumes is crucial to informing better management of such disasters given the severity of the devastating impacts of flooding (e.g., the 2016 flooding across China). Although recent studies have investigated the impacts of future climate change on urban flooding, the effects of both climate change mitigation and adaptation have rarely been accounted for together in a consistent framework. In this study, we assess the benefits of mitigating climate change by reducing greenhouse gas emissions and locally adapting to climate change by modifying drainage systems to reduce urban flooding under various climate change scenarios through a case study conducted in Northern China. The urban drainage model—Storm Water Management Model—was used to simulate urban flood volumes using current and two adapted drainage systems (i.e., pipe enlargement and low-impact development), driven by bias-corrected meteorological forcing from five general circulation models in the Coupled Model Intercomparison Project Phase 5 archive. Results indicate that urban flood volume is projected to increase by 52% in 2020–2040 compared to the volume in 1971–2000 under the business-as-usual scenario (i.e., Representative Concentration Pathway (RCP) 8.5). The magnitudes of urban flood volumes are found to increase nonlinearly with changes in precipitation intensity. On average, the projected flood volume under RCP 2.6 is 13% less than that under RCP 8.5, demonstrating the benefits of global-scale climate change mitigation efforts in reducing local urban flood volumes. Comparison of reduced flood volumes between climate change mitigation and local adaptation (by improving the drainage system) scenarios suggests that local adaptation is more effective than climate change mitigation in reducing future flood volumes. This has broad implications for the research community relative

to drainage system design and modelling in a changing environment. This study highlights the
importance of accounting for local adaptation when coping with future urban floods.

**Keywords:** Climate change, urban floods, mitigation, adaptation, drainage systems

1.      Introduction
Floods are one of the most hazardous and frequent disasters in urban areas and can cause
enormous impacts on the economy, environment, city infrastructure, and human society (Chang
et al., 2013; Ashley et al., 2007; Zhou et al., 2012). Urban drainage systems have been
constructed to provide carrying and conveyance capacities at a desired frequency to prevent
urban flooding. The design of drainage systems is generally based on historical precipitation
statistics for a certain period of time, without considering the potential changes in precipitation
extremes for the designed return periods (Yazdanfar and Sharma, 2015; Peng et al., 2015;
Zahmatkesh et al., 2015). For example, in Danish design guidelines for urban drainage, a 30%
and 40% increase in the precipitation intensity is expected for the 10- and 100-year return
periods, respectively (Arnbjerg-Nielsen, 2012). The systems are, however, likely to be
overwhelmed by additional runoff effects induced by climate change, which may lead to
increased flood frequency and magnitude, disruption of transportation systems, and increased
human health risk (Chang et al., 2013; Abdellatif et al., 2015). Therefore, it is important to
investigate the performance of drainage systems in a changing environment and to assess the
potential urban flooding under various scenarios to achieve better adaptations (Mishra, 2015;
Karamouz et al., 2013; Yazdanfar and Sharma, 2015; Notaro et al., 2015).

Impacts of climate change on extreme precipitation and urban flooding have been well
documented in a number of case studies. For example, Ashley et al. (2005) showed that flooding
risks may increase by almost 30 times in comparison to current situations, and effective
adaptation measures are required to cope with the increasing risks in the UK. Larsen et al. (2009)
estimated that future extreme one-hour precipitation will increase by 20%~60% throughout
Europe. Willems (2013) found that in Belgium the current design storm intensity for the 10-year
return period is projected to increase by 50% by the end of this century. Several studies have also
investigated the role of climate change mitigation and adaptation in reducing urban flood
damages and risks under climate change scenarios (Alfieri et al., 2016; Arnbjerg-Nielsen et al.,
2015; Moore et al., 2016; Poussin et al., 2012). To date, however, limited work has been done to
investigate the relationship between changes in precipitation intensity and flood volume to
provide additional insights into drainage design strategies. More importantly, investigations of
the benefits of climate change mitigation (by reducing greenhouse gas emissions [GHG]) and
local adaptation (by improving drainage systems) in reducing future urban flood volumes are
typically conducted separately, rather than within a consistent framework.

As China has become increasingly urbanised, flooding has become a regular occurrence in its
cities; 62% of Chinese cities surveyed experienced floods and direct economic losses of up to
$100 billion between 2011 and 2014 (China Statistical Yearbook 2015). The 2016 flooding
affected more than 60 million people—more than 200 people were killed and $22 billion in
losses were suffered across China. Hence, assessing future changes in urban flooding is very
important for managing urban floods by designing new and re-designing existing urban
infrastructures to be resilient in response to the impacts of future climate change. While urban
floods are speculated to increase in the future (Yang 2000; Ding et al., 2006), their magnitudes
are hard to assess because of uncertainties associated with future climate change scenarios, as
well as the under-representation of plausible climate change mitigation and adaptation strategies
in the models.

In this study, we chose a drainage system in a typical city in Northern China to illustrate the role
of climate change mitigation and local adaptation in coping with future urban flood volumes.
Such an investigation of the performance of the present-day drainage system also has important
implications for local governments responsible for managing urban flood disasters in the study
region. Specifically, we first quantified the effects of future climate change on urban flood
volumes as a result of extreme precipitation events for various return periods using the present-
day drainage system. We then designed two plausible adaptation strategies for the study region
and investigated how much urban flood volume can be reduced by the adapted systems. We also
compared the benefits of global-scale climate change mitigation and local adaptation in reducing
urban flood volumes to advance our understanding of the effective measures for coping with
future urban floods.

## 2.    Materials and Methods
a.  Study region
The study region (Hohhot City) is located in the south-central portion of Inner Mongolia, China.
It lies between the Great Blue Mountains to the north and the Hetao Plateau to the south, which
has a north-to-south topographic gradient. The drainage area in year 2010 was about 210.72 $km^2$
and it served a residential population of 1.793 million (Figure 1a). The land use types in the
region can be classified into five categories: agricultural land (8%), residential areas (38%),
industrial land (13%), green spaces (7%), and other facilities (34%, including municipal squares,
commercial districts, institutions). The planned drainage area in 2020 is about 307.83 km$^2$, which
is 50% larger than the current drainage area. The land use categories and distribution are shown
in Figure 1b.

The region is in a cold semi-arid climate zone, characterised by cold and dry winters and hot and
humid summers. The regional annual mean precipitation is approximately 396 mm and it
exhibits large intra-seasonal variations. Most rain storms fall between June and August, a period
that accounts for more than 65% of the annual precipitation. According to local water authorities,
the major soil type of the area is a mixture of loam and clay. The current drainage system can be
divided into three large sub-basins (Figure 1c) and 326 sub-catchments with a total pipeline
length of 249.36 km. The drainage network has a higher pipeline cover rate in the central part,
but a rather low design standard for extreme rainfall events with a return period of less than 1
year. Historical records of stormwater drainage and flood damage indicate that the region has
experienced an increase in flood frequency and magnitude within the context of climate change
and urbanisation. During the major flood event on 11 July 2016, the city, especially the western
portion of the watershed, was hit by an extreme rainfall event that featured more than 100 mm of
rain in 3 hours. The flood event led to the cancellation of at least 8 flights and 17 trains, and
delays of several transportation systems. In particular, in the central area, the flood event caused
severe traffic jams on major streets and resulted in a number of flooded residential buildings. A
new drainage system is therefore required to cope with increasing urban flood volumes and
frequencies in the future.

b.   Climate change scenarios
Climate projections by five general circulation models (GCMs) from Phase 5 of the Coupled
Model Intercomparison Project (CMIP5) archive were obtained from the Inter-Sectoral Impact
Model Intercomparison Project (ISI-MIP) (Warszawski et al., 2014). The CMIP5 climate
projections were bias-corrected against observed climate for the overlapping period 1950–2000
using a quantile mapping method (Piani et al., 2010; Hempel et al., 2013). The bias-corrected
CMIP5 climate projections represent a complete climate change picture that includes both the
mean property and variation of future climate. Several studies have demonstrated the value of the
bias-corrected climate projections in quantifying climate change impacts on global and regional
hydrology (e.g., Piontek et al., 2014; Elliott et al., 2014; Haddeland et al., 2014; Leng et al.,
2015a,b). In this study, we used the bias-corrected climate from all five GCMs (HadGEM2-ES,
GFDL-ESM2M, IPSLCM5A-LR, MIROC-ESM-CHEM, and NorESM1-M) under two
Representative Concentration Pathways (RCPs) (i.e., RCP 2.6 and RCP 8.5). The projected
urban flood volumes under the business-as-usual scenario RCP 8.5 are compared with those
under the climate change mitigation scenario RCP 2.6 to explore the benefits of climate change
mitigation in reducing regional urban flood volumes. The possible land-surface-atmosphere
interactions that would indirectly affect rainfall and flooding are not considered in this study.

c.   Urban drainage modelling
The Storm Water Management Model (SWMM 5.1) developed by the U.S. Environmental
Protection Administration is a widely used urban stormwater model that can simulate rainfall-
runoff routing and pipe dynamics under single or continuous events (Rossman and Huber, 2016).
SWMM can be used to evaluate the variation in hydrological and hydraulic processes and the
performance of drainage systems under specific mitigation and adaptation scenarios in the
context of global warming. The hydrological component requires inputs of precipitation and
subcatchment properties including drainage area, subcatchment width, and imperviousness. The
pipe network requires inputs from manholes, pipelines, outfalls, and connections to sub-
catchments (Zahmatkesh et al., 2015; Chang et al., 2013). Basic flow-routing models include
steady flow, kinematic, and dynamic wave methods. Infiltration can be described by the Horton,
Green-Ampt, or Curve Number (SCS-CN) methods. The dynamics of pipe flow are calculated
based on the continuity equation and Saint-Venant equations (Rossman and Huber, 2016).
Overflow occurs once the surface runoff exceeds the pipe capacity and is expressed as the value
of total flood volume (TFV) at each overloaded manhole; i.e., the excess water from manholes
after completely filling the pipe system without taking into account the outlet discharges. Other
types of model outputs include catchment peak flows, maximum flow rates of pipelines, and
flooded hours of manholes. It should be noted that SWMM is not capable of simulating surface
inundation dynamics and cannot provide accurate estimation of the inundated zones and depths.
The TFV value is thus used to approximately reflect the flood condition and drainage system
overloading status. Nevertheless, surface inundation models (e.g., Apel et al., 2009; Horritt and
Bates, 2002; Vojinovic and Tutulic, 2009) are applicable if more accurate information about
overland flow characteristics is available. In this study, the kinematic wave routing and the
Horton infiltration model are used for model simulations. The infiltration capacity parameters for
the category of "Dry loam soils with little or no vegetation" are used in the hydrological model to
be consistent with the local soil type (Akan, 1993; Rossman and Huber, 2016) (Table 1).

Rainfall inputs are calculated based on the regional storm intensity formula (SIF) using historical
climatic statistics (Zhang and Guan, 2012) (see Equation 1). Application of the SIF is a standard
practice for determining design rainfalls in urban drainage modelling in China, and is well
documented in the National Guidance for Design of Outdoor Wastewater Engineering
(MOHURD, 2011). In fact, the SIF represents an Intensity-Duration-Frequency (IDF)
relationship, which is a common approach in literature for estimating design rainfall hydrographs
using the Chicago Design Storms (CDS) approach (Berggren et al., 2014; Willems et al., 2012;
Zhou et al., 2013).

$$q = \frac{A(1 + Dlg(P))}{(t + b)^c} \qquad \text{Eq. (1)}$$

where $q$ is the average rainfall intensity, and $P$ and $t$ are the design return period and duration of
storm, respectively. The typical temporal resolution considered in SIF for urban drainage
modelling is minutes. $A$, $b$, $c$, and $D$ are regional parameters governing the IDF relationship
among rainfall intensity, return period, and storm duration. For the study region, the values of $A$,
$b$, $c$, and $D$ were obtained from the local weather bureau and are equal to 635, 0, 0.61, and 0.841,
respectively.

The procedure for applying SIF to obtain CDS is outlined in the National Technical Guidelines
for Establishment of Intensity-Duration-Frequency Curve and Design Rainstorm Profile
(MOHURD, 2014; Zhang et al., 2008; Zhang et al., 2015). Specifically, for a given return period,
the SIF is fitted into the Horner's equation as:

$$i = \frac{a}{(t + b)^c} \qquad \text{Eq. (2)}$$


The synthetic hyetograph based on the Chicago method is computed using Equation 2 and an
additional parameter $r$ (where $0 < r < 1$), which determines the relative time step of the peak
intensity, $t_p = r*t$. The time distribution of rainfall intensity is then described after the peak $t_a =$
$(1-r)*t$ and before the peak $t_b = r*t$ using Equations 3 and 4, respectively, where $i_b$ and $i_a$ are the
instantaneous rainfall intensity before and after the peak:

$$i_a = \frac{a[\frac{(1-c)t_a}{(1-r)} + b]}{(\frac{t_a}{(1-r)} + b)^{1+c}} \qquad \text{Eq. (3)}$$

$$i_b = \frac{a[\frac{(1-c)t_b}{r} + b]}{(\frac{t_b}{r} + b)^{1+c}} \qquad \text{Eq. (4)}$$

In this study, we considered 10 return periods, i.e., the 1-, 2-, 3-, 10-, 20-, 50-, 100-, 200-, 500-,
and 1000-year events. A 4-hour rainfall time series was generated for each return period at 10-
minute intervals based on Equations 1−4. We assumed that the SIF was constant without
considering the non-stationary features in a changing climate. That is, the IDF relationships were
assumed to remain stable in the future and only changes in the daily mean intensity were
considered because of the limited data availability in future sub-hourly climate projections from
which to derive the parameters.

As for future climate, the projected changes (i.e., change factors) in precipitation intensity at
various return periods were calculated for each GCM-RCP combination (Table 2). Specifically,
for each year, the annual maximum daily precipitation was determined for both historical and
future periods. The generalised extreme value (GEV) distribution was then fitted separately to
the two sets of daily values (Coles 2001; Katz et al. 2002). The goodness-of-fit was tested by
calculating the Kolmogorov–Smirnov and Anderson–Darling statistics. The value corresponding
to each return period was estimated based on the GEV distribution and the changes between
future and historical periods were calculated as the change factors. The derived change factor for
each return period was then multiplied by the historical design CDS rainfall time series to derive
future climate scenarios. We acknowledge that the estimation of changes in extreme precipitation
events involves inevitable uncertainties and therefore caution should be exercised when
interpreting the relevant results.

d.   Flood volume assessment
The TFV values of given rainfall events were simulated by the SWMM. A log-linear relationship
is assumed to characterize the changes in flood volume with the increase in precipitation
intensity as indicated by return periods (Figure 2a) following Zhou et al. (2012) and Olsen et al.
(2015). Generally, more intense rainfall will induce higher TFVs. The TFVs were further linked
to their occurrence frequencies to derive the expected flood volume for a flood event at a specific
probability (Figure 2b). The total grey area under the curve represents the average total TFVs per
year for all floods at various return periods. The contribution of an individual flood event to total
TFVs is dependent not only on the flood volume, but also its corresponding probability of
occurrence. Intensified precipitation is expected to increase the magnitude of system overflow,
resulting in an upward trend in the TFV-return period relationship and increased total TFVs.
Mitigation and adaptation are aimed at reducing or preventing the impacts of global warming on
flood volumes.

e.   Design of adaptation scenarios
In this study, two adaptation scenarios were designed to explore the role of adaptation in
reducing urban flood volume within the context of climate change. The first scenario adapted the
drainage system as planned by the water authorities to cope with the designed standard of a 3-
year design event. It involved two main improvements of the current drainage system—
enhancing the pipeline diameter and expanding the pipe network. The design was implemented
in the SWMM model as shown in Figure 1c. The number of pipelines of the present-day and
adapted systems was 323 and 488, with a total pipe length of 251.6 km and 375.4 km,
respectively. In the adapted scenarios, the mean pipeline diameter was about 1.73 m, which
increased by 53% compared to that of the present-day system.

A variety of site-specific factors, such as the imperviousness of land area in the drainage basin,
can also influence the performance of a drainage system in managing surface runoff. The second
adaptation scenario was to increase the permeable surfaces (e.g., green spaces) and reduce the
regional imperviousness in the study region on the basis of pipe capacity enhancement. This
scenario is referred to as the Low Impact Development (LID) scenario, and it was used to
explore the effectiveness of urban green measures, such as the use of permeable pavements,
infiltration trenches, and green roofs. Due to a lack of detailed information about the permeable
soil and coverage rates in the study region, the effects of these specific measures cannot be
modelled individually. Here, we used a simplified approach by altering the subcatchment
imperviousness to reflect the combined effects of infiltration-related measures. We derived such
information by comparing the current and planned land use maps using a geographical
information system (GIS) and incorporated the changes in land use and imperviousness into the
designed LID scenarios. Figure 1d shows the difference in weighted mean imperviousness (WMI)
calculated for each subcatchment in the current and planned maps, using the commonly applied
impervious factors (Pazwash, 2011; Butler and Davies, 2004) for each land use type. The
difference in WMI was used to indicate the area potential for adaptation based on the city plan.
For example, a subcatchment with higher positive changes in the WMI indicates that the area is
planned to have a land use type with lower imperviousness and therefore is assumed to be more
suitable for LID planning, and vice versa.

3.      Results
a.   Impacts of future climate change on urban flood volumes
Figure 3 shows the projected climate change impacts on urban flooding using the present-day
drainage system of the near future (i.e., 2020–2040) under the RCP 8.5 scenario. Without climate
change mitigation or adaptation, the TFV was projected to increase significantly with the
increase of extreme rainfall events for most of investigated return periods (Table 2). Note that the
lower bounds for return periods of 1, 3, and 1000 years fall below the current TFV curve due to
the decrease in precipitation intensities. Despite the large uncertainty associated with climate
projections, in particular with the 1-, 10-, and 1000-year return periods, the poor service
performance of the current system in coping with urban flooding was evident. Overall, the urban
flood volume was projected to increase by 52% on average by the multi-model ensemble median
by 2020–2040; the largest increase (258%) was projected for the 1-year event and the smallest
increase (12%) for the 100-year event.

b.   Benefits of climate change mitigation in reducing urban flood volumes
Figure 4 shows the comparison of TFVs under the RCP 8.5 scenario (i.e., a business-as-usual
scenario) and the RCP 2.6 scenario (i.e., a climate change mitigation scenario). Although large
uncertainties exist arising from climate models, it is clear that the simulated TFVs are much
smaller under the RCP 2.6 scenario than under the RCP 8.5 scenario, demonstrating the benefits
of climate mitigation in reducing local urban flood volumes. Such benefits are especially evident
for floods for smaller return periods. For example, an increase of 936 $m^3$ in flood volume is
projected with the increase in 1-year extreme rainfall under the business-as-usual climate change
scenario (i.e., RCP 8.5), 52% of which would be reduced if climate change mitigation is in place
(i.e., under RCP 2.6). Overall, climate change mitigation can reduce future flood volumes by 13%
compared to the scenario without mitigation, as indicated by the multi-model ensemble median.
Notably, the peak of the total TFV curve was even projected to shift from the 1-year event under
the RCP 8.5 scenario to the 3-year event under the RCP 2.6 scenario (Figure 4b). Such a shift in
the peak toward smaller return periods combined with a flatter curve demonstrates the important
role of climate mitigation in regulating local urban flood volumes.

c.  Benefits of adaptation in reducing urban flood volumes
Figure 5 shows the overloaded pipelines (red colour) with and without adaptation. The simulated
results under the present 3-year event (recommended service level) and 50-year event (one
typical extreme event) were selected to illustrate the role of adaptation in coping with floods in
the historical period. As shown in Figure 5a, the simulated locations of overloaded pipelines are
in good agreement with historical flood points as recorded by local water authorities. Overall, the
percentage of overloaded manholes (POM) and the ratio of flood volume (RFV) are up to 37%
and 35% in the current drainage system (Figure 5a), respectively. When experiencing a 50-year
extreme rainfall, the POM and RFV increase to 67% and 38%, respectively. This indicates that
current pipe capacities are insufficient to cope with extreme rainfall events (Figure 5b). Spatially,
the central portion of the city is the most affected region due to the low service level in the area.
With proposed adaptations, urban floods can be reduced to zero under a 3-year flood event. Such
benefits of local adaptations are also evident when experiencing more intense precipitation
events (e.g., 50-year events), for which the POM and RFV reduced from 67% and 50% to 49%
and 17%, respectively.

Figure 6 shows the future changes in urban flood volume (CTFVs) ($CTFV=(TFV_c - TFV_{nc})/TFV_{nc}$,
where $c$ and $nc$ represent the results with and without climate change, respectively) with changes
in extreme rainfall for various return periods. The performance of the current drainage system
(no adaptation) was found to be less sensitive to future climate change, as indicated by the flatter
slope in Figure 6. For example, a similar magnitude of changes in flood volume was projected
given changes in extreme rainfall for the return periods of 3, 50, and 500 years; the CTFV is 0.62,
0.32 and 0.35 for these periods, respectively. This is because the capacity of the current system is
too small to handle extreme rainfall events with return periods larger than 1 year—a condition
under which the current drainage system would be flooded completely, not to mention the
situations with increased rainfall intensity in the future. Mathematically, the low sensitivity of
the current drainage system to changes in extreme rainfall intensity could be attributed to the
large value of the denominator in the calculation of $CTFV$.

With adaptations in place, the flood volume becomes much smaller than that in the current
system due to capacity upgrading to hold more water. For example, when experiencing a 10-year
extreme rainfall event, the urban flood volumes for the present period (i.e., $TFVnc$) are 1041,230,
274,650 and 180,610 $m^3$ in the current and two adapted systems, respectively, while in the future
period, the magnitude of flood volume (i.e., *TFVc*) is relatively similar among the three drainage
systems. Therefore, future CTFVs relative to the historical period are much larger in the adapted
systems than in the current system. The larger CTFVs in the adapted systems do not mean a
worsened drainage system performance. Rather, they imply that the capacity (i.e., service level)
of adapted drainage systems tends to become lower with climate change, while the current
drainage system has already reached its peak capacity in handling extreme rainfall events in the
historical period and thus shows a low sensitivity to future increases in rainfall intensity under
climate change scenarios. Notably, the considerable increases in the CTFVs for return periods of
less than 10 years in the adapted systems imply that the designed adaptations can effectively
attenuate extreme rainfall events for small return periods. For more extreme rainfall events of
return periods ≥50 years, more consistent results were found for both adaptation scenarios. This
indicates that although the performances of adapted drainage systems are significantly improved
compared to that of the current system, the flood volume remains large when experiencing
extreme rainfall events with return periods larger than 50 years, because flooding in such cases
will push the adapted drainage systems to their upper limits.

d.   Climate mitigation versus drainage adaptation
Figure 7 shows the reduced TFVs by climate change mitigation and drainage system adaptation
as functions of return period. It is evident that both mitigation and adaptation measures are
effective in reducing future urban flood volumes. However, such benefits are projected to
weaken gradually with the increase in rainfall intensity (i.e., larger return periods). Importantly,
our results show that the two adaptation systems proposed in this study are found to be more
effective in reducing urban floods than climate change mitigation. In most cases, the benefits of
local adaptation are more than double those of mitigation. In extreme cases, the reduction in TFV
achieved by adaptation is five times more than that achieved by climate change mitigation (i.e.,
for the return periods of 2–3 years). Such effectiveness of urban flood reduction through
drainage system adaptations has profound implications for local governments charged with
managing urban flooding in the future. Notably, the second scenario (LID+pipe) exhibited a
higher level of flood volume reduction than the pipe scenario in coping with extreme rainfall
events for all investigated return periods. This implies that implementation of LID measures to
augment drainage system capacity is more effective through reducing upstream loadings
compared to updating the pipe system alone.

It is noted that local soil characteristics could affect the performance of the designed adaptation
systems, in particular the LID measures. However, information about soil properties was not
available at the subcatchment level in the study region. Here, a set of sensitivity experiments
were conducted by adopting different parameters (e.g., infiltration values) associated with
possible soil conditions (i.e., dry sand, loam, and clay soils with little or no vegetation in Table 1)
for the area. The boundary bars in Figure 7 show the uncertainty range arising from the
representation of different soil conditions in the drainage model. The benefits of the designed
adaptation measures in reducing urban flood volumes were found to be robust regardless of soil
conditions, and such benefits exceeded those of climate change mitigation, confirming our major
conclusions found in this study.

## 4.    Uncertainties and Limitations

A number of uncertainties and limitations arise from the model structure, parameter inputs, emission scenarios, GCMs, climate downscaling/bias-correction approaches, etc. Specifically, climate projections by GCMs are subject to large uncertainties, in particular regarding precipitation (Covey et al., 2003) at spatial scales, which are relevant for urban flood modelling. An alternative approach is to simulate future climate using a regional climate model (RCM) nested within a GCM. Such climate projections by RCMs have added value in terms of higher spatial resolution, which can provide more detailed regional climate information. However, various levels of bias would still remain in RCM simulations (Teutschbein and Seibert 2012) and bias corrections of RCM projections would be required; e.g., the European project ENSEMBLES (Hewitt and Griggs 2004; Christensen et al. 2008). To run a RCM was not within the scope of this study; instead, we tended to use publicly available climate projections. Here, we obtained the climate projections from the ISI-MIP (Warszawski et al. 2014), which provides spatially downscaled climate data for impact models. The climate projections were also bias-corrected against observations (Hempel et al. 2013) and have been widely used in climate change impact studies on hydrological extremes such as floods and droughts (e.g., Dankers et al. 2014; Prudhomme et al. 2014; Leng et al. 2015a). It should be noted that we used the delta change factor to derive future climate scenarios as inputs into our drainage model instead of using GCM climate directly. This is because the relative climate change signal simulated by GCMs is argued to be more reliable than the simulated absolute values (Ho et al. 2012). Moreover, we used an ensemble of GCM simulations rather than one single climate model in order to characterise the uncertainty range arising from climate projections. However, disadvantages of this method are that transient climate changes cannot be represented and that changes in intra-seasonal or daily

climate variability are not taken into account (Leng and Tang, 2014). Such sources of uncertainty
can be explored when improved climate models at finer scales become available (Jaramillo and
Nazemi 2017).

In addition, the SIF parameters were assumed to remain stable in the future and only changes in
the daily mean intensity were considered, because future sub-hourly climate projections were not
readily available. The full climate variability range would also be under-sampled, although we
used five climate models to show the possible range. Given the above limitations, we
acknowledge that the modelling results represent the first-order potential climate change impacts
on urban floods. Future efforts should be devoted to the representation of dynamic rainfall
changes at hourly time steps with consideration of non-stationary climate change.

Moreover, several assumptions had to be made due to limitations of the current modelling
structure and approach. For example, the conveyance capacities of the drainage system and flood
volume would largely depend on the state of drainage systems. Hence, a drainage system
obstructed by vegetation, waste, or artefacts (cables, pipes, temporary constructions) can make
the outcomes of the SWMM calculation significantly different from observations. However,
quantifying the impacts of drainage system states on urban flood volumes is not trivial because
of the difficulties involved in collecting field data and selecting and using appropriate methods
for reasonable assessment of pipe conditions (Ana and Bauwens, 2007; Fenner, 2000), and was
not within the scope of this study. With deterioration, such as ageing network, pipe deterioration,
blockage, and construction failures, drainage systems were shown to become more vulnerable to
extreme rainfalls as demonstrated in previous studies (Dawson et al., 2008; CIRIA, 1997; Davies
et al., 2001). It is very likely that our simulated urban flood volumes would be underestimated
without considering the changes in drainage conditions (Pollert et al., 2005).

Further, constrained by the one-dimensional modelling approach using SWMM, the
performances of LID measures were mainly evaluated according to their effects in reducing
water volume from overloaded manholes (Oraei Zare et al., 2012; Lee et al., 2013). That is, the
LID adaptation measure was mainly designed to reduce the amount of water rather than slowing
down the water speed, which has been demonstrated to be effective in reducing urban floods
(Messner et al., 2006; Ashley et al., 2007; Floodsite, 2009). However, it should be noted that
most LID measures can reduce runoff volume and flow speed at the same time, although some of
the LID measures are primarily designed to slow down the flow speed, i.e., vegetated swales. To
examine whether flood retention of a given event is induced by runoff volume or the internal
speed control function in the model is difficult and requires detailed data for model validations.
Specifically, the required information about surface roughness, soil conductivity, and seepage
rate were unavailable at the subcatchment scale in the study region. Therefore, a simplified
modelling approach was used to take advantage of existing data, especially for the design of LID
measures. With the aid of more detailed field data and planning documents, the design of LID
measures could be significantly improved by implementing more advanced approaches (Elliott
and Trowsdale, 2007; Zoppou, 2001). Evaluation of other potential adaptation strategies, such as
flood retention by rain gardens and green roofs, can be explored in the future to gain additional
insights into the performance of LID systems. In particular, the cost-effectiveness of the
proposed adaptation measures should be accounted for. Nevertheless, given these limitations,
this study stands out from previous climate impact assessment studies of urban flood volumes by
having proposed two feasible adaptation strategies and compared their benefits to those from
global-scale climate change mitigations through GHG reductions within a consistent framework.
Depending on the progress on data collection and the demands of local authorities, more
advanced methods for pipe assessment (e.g., considering the changing pipe conditions), LID
measures (detailed modelling of LID control), and two-dimensional surface flooding for
assessment of flood damage and risk are planned in a future study to provide a more
comprehensive analysis of the adaptation measures.

## 5.    Summary and Conclusions

The potential impacts of future climate change on current urban drainage systems have received
increasing attention during recent decades because of the devastating impacts of urban flooding
on the economy and society (Chang et al., 2013; Zhou et al., 2012; Abdellatif et al., 2015).
However, few studies have explored the role of both climate change mitigation and drainage
adaptations in coping with urban flooding in a changing climate. This study investigated the
performance of a drainage system in a typical city in Northern China in response to various
future scenarios. In particular, we assessed the potential changes in urban flood volume and
explored the role of both mitigation and adaptation in reducing urban flood volumes in a
consistent manner.

Our results show significant increases in urban flood volumes due to increases in precipitation
extremes, especially for return periods of less than 10 years. Overall, urban flood volume in the
study region is projected to increase by 52% by the multi-model ensemble median in the period
of 2020–2040. Such increases in flood volume can be reduced considerably by climate change
mitigation through reduction of GHG emissions. For example, the future TFVs under 1-year
extreme rainfall events can be reduced by 50% when climate change mitigation is in place.
Besides global-scale climate change mitigation, regional/local adaptation can be implemented to
cope with the adverse impacts of future climate change on urban flood volumes. Here, the
adaptation measures as designed in this study were demonstrated to be much more effective in
reducing future flood volumes than climate change mitigation measures. In general, the reduced
flood volumes achieved by adaptation were more than double those achieved by climate change
mitigation.

Through a comprehensive investigation of future urban floods, this study provides much-needed
insights into urban flood management for similar urban areas in China, most of which are
equipped with highly insufficient drainage capacities. By comparing the reduction of flood
volume by climate change mitigation (via reduction of GHG emissions) and local adaptation (via
improvement of drainage systems), this study highlights the effectiveness of system adaptations
in reducing future flood volumes. This has important implications for the research community
and decision-makers involved in urban flood management. We emphasise the importance of
accounting for both global-scale climate change mitigation and local-scale adaptation in
assessing future climate impacts on urban flood volumes within a consistent framework.

**Acknowledgements**
This research was supported by the Natural Science Foundation of Guangdong Province, China
(No. 2014A030310121) and the Scientific Research Foundation for the Returned Overseas
Chinese Scholars, State Education Ministry. G. Leng and M. Huang were supported by the

Integrated Assessment Research program through the Integrated Multi-sector, Multi-scale Modeling (IM$^3$) Scientific Focus Area (SFA) sponsored by the Biological and Environmental Research Division of Office of Science, U.S. Department of Energy. The Pacific Northwest National Laboratory (PNNL) is operated for the U.S. DOE by Battelle Memorial Institute under contract DE-AC05-76RL01830

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

664     **Table 1** Infiltration parameters for three categories of soil in the SWMM simulation

| Soil category | Infiltration parameters | | | |
|---|---|---|---|---|
| | MaxRate | MinRate | Decay rate | DryTime |
| | [in/hr] | [in/hr] | [1/hr] | [days] |
| Dry loam with little or no vegetation | 3 | 0.5 | 4 | 7 |
| Dry sand with little or no vegetation | 5 | 0.7 | 5 | 5 |
| Dry clay with little or no vegetation | 1 | 0.3 | 3 | 9 |

665

666

667     **Table 2** Projected changes in precipitation intensity under return periods ranging from 1 year to 1000
668     years by five Global Climate Models under two Representative Concentration Pathways (RCPs)

| | | 1 | 2 | 3 | 10 | 20 | 50 | 100 | 200 | 500 | 1000 |
|---|---|---|---|---|---|---|---|---|---|---|---|
| GFDL-ESM2M | RCP8.5 | 2.12 | 1.23 | 1.34 | 1.25 | 1.27 | 1.21 | 1.08 | 1.12 | 1.24 | 1.23 |
| | RCP2.6 | 1.74 | 1.08 | 1.03 | 1.11 | 1.07 | 1.15 | 1.14 | 1.15 | 1.19 | 1.16 |
| HadGEM2-ES | RCP8.5 | 0.62 | 1.08 | 1.09 | 1.06 | 1.01 | 1.03 | 1.17 | 1.26 | 1.23 | 1.14 |
| | RCP2.6 | 0.36 | 1.2 | 1.19 | 1.04 | 1.02 | 1.11 | 1.31 | 1.26 | 1.37 | 1.24 |
| IPSL-CM5A-LR | RCP8.5 | 1.44 | 1.17 | 1.28 | 1.17 | 1.08 | 1.09 | 1.02 | 1.1 | 1.12 | 1.13 |
| | RCP2.6 | 0.74 | 1.04 | 1.18 | 1.01 | 1.06 | 1.03 | 1.01 | 0.99 | 0.95 | 1 |
| MIROC-ESM-CHEM | RCP8.5 | 2.13 | 1.38 | 1.3 | 1.51 | 1.32 | 1.23 | 1.17 | 1.27 | 1.16 | 1.31 |
| | RCP2.6 | 0.71 | 1.12 | 1.14 | 1.18 | 1.1 | 1.07 | 1.01 | 1.09 | 1.01 | 1.09 |
| NorESM1-M | RCP8.5 | 2.11 | 0.96 | 0.8 | 1.63 | 1.35 | 1.15 | 1.08 | 1.01 | 1.04 | 0.97 |
| | RCP2.6 | 0.11 | 1.09 | 1.05 | 1.28 | 1.17 | 1.08 | 1.1 | 1.18 | 1.09 | 1.2 |

669

## List of Figures

**Figure 1** Land use of the study region for the year 2010 (a) and 2020 (b). Pipe network description of current and planned drainage systems (c). Difference in Weighted Mean Imperviousness (WMI) between year 2010 and 2020 (d).

**Figure 2** Illustration of flood volume and average total expected total flood volume (TFVs) as a function of return period under a stationary drainage system. The grey area denotes the average total expected TFVs per year considering all kinds of floods.

**Figure 3** Projected TFV with changes in precipitation intensity at various return periods under the RCP8.5 scenario for the period of 2020–2040.

**Figure 4** Comparison of (a) flood volume, (b) total TFVs (i.e., the piece-wise integral of flood volume versus the expected frequency with changes in precipitation intensity of various return periods under RCP8.5 (blue) and RCP2.6 (red). (c) is for the reduced TFVs in percentage (i.e., benefits of climate mitigation) in RCP2.6 relative to RCP8.5 at various return periods.

**Figure 5** Spatial distribution of overloaded pipelines (red colour) induced by the 3-year (left column) and 50-year extreme events (right column) without and with adaptations. The total percentage of overloaded manholes (POM) and ratio of flood volume (RFV) are summarised for each scenario. Descriptions of local land use, mainly the traffic network and green spaces, are provided as the background image in (a).

**Figure 6** Future changes in flood volumes (CTFVs) relative to historical conditions under the current drainage system (yellow) and two adaptation scenarios (i.e., Pipe in red and Pipe+LID in green) at various return periods.

**Figure 7** Comparison of benefits of climate mitigation and two adaptation strategies in reducing urban flood volumes with changes in precipitation intensities for various return periods, and with related variations (boundary bars) as a result of uncertainty arising from local soil conditions.

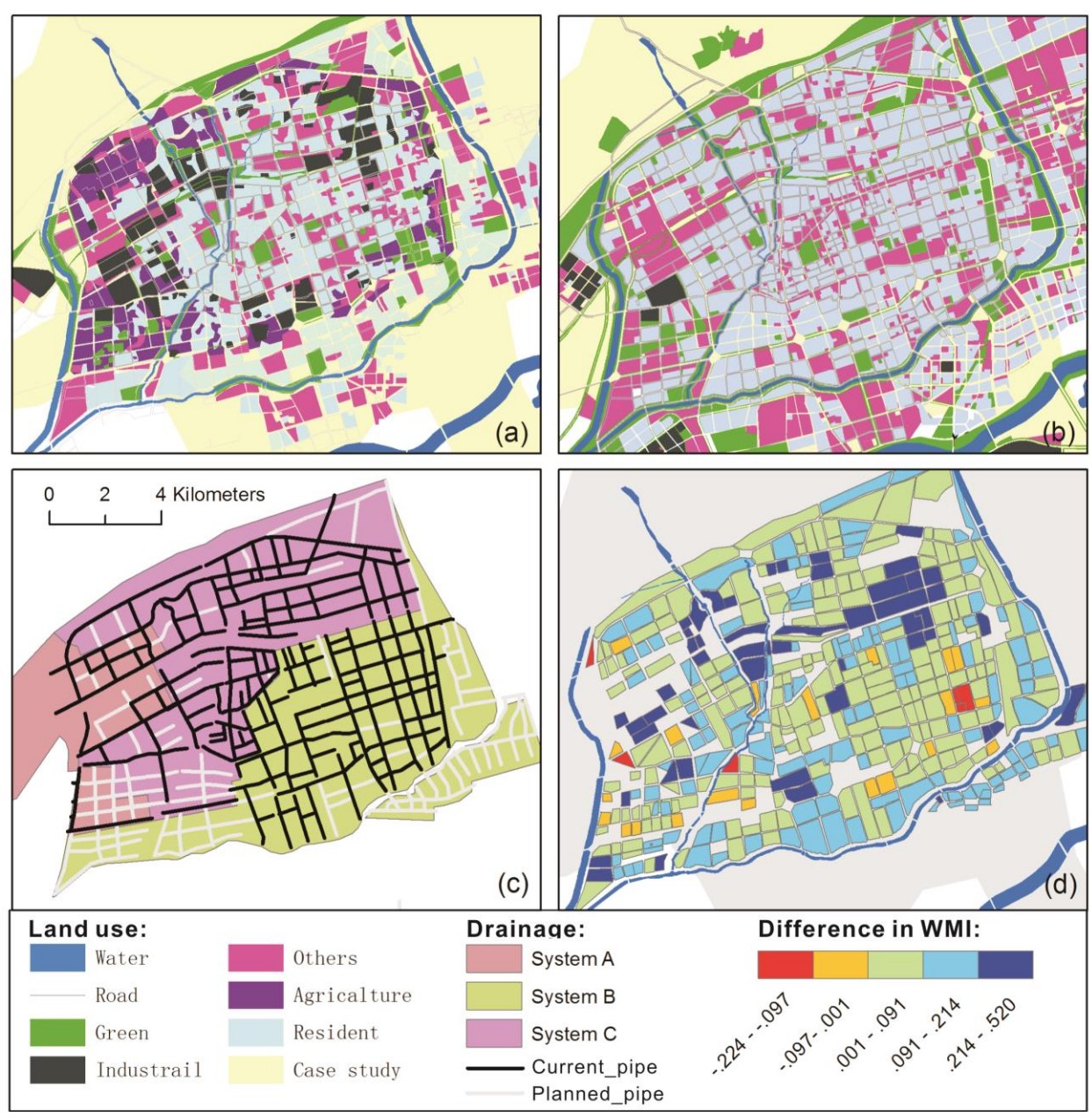


**Figure 1** Land use of the study region for the year 2010 (a) and 2020 (b). Pipe network description of current and planned drainage systems (c). Difference in Weighted Mean Imperviousness (WMI) between year 2010 and 2020 (d).


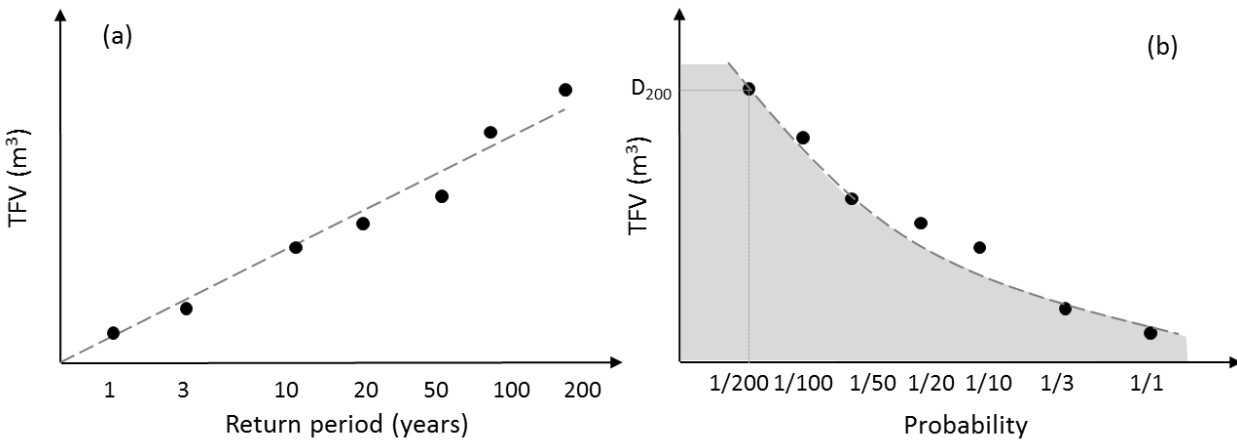


**Figure 2** Illustration of flood volume and average total expected total flood volumes (TFVs) as a
function of return period under a stationary drainage system. The grey area denotes the average
total expected TFVs per year considering all kinds of floods.

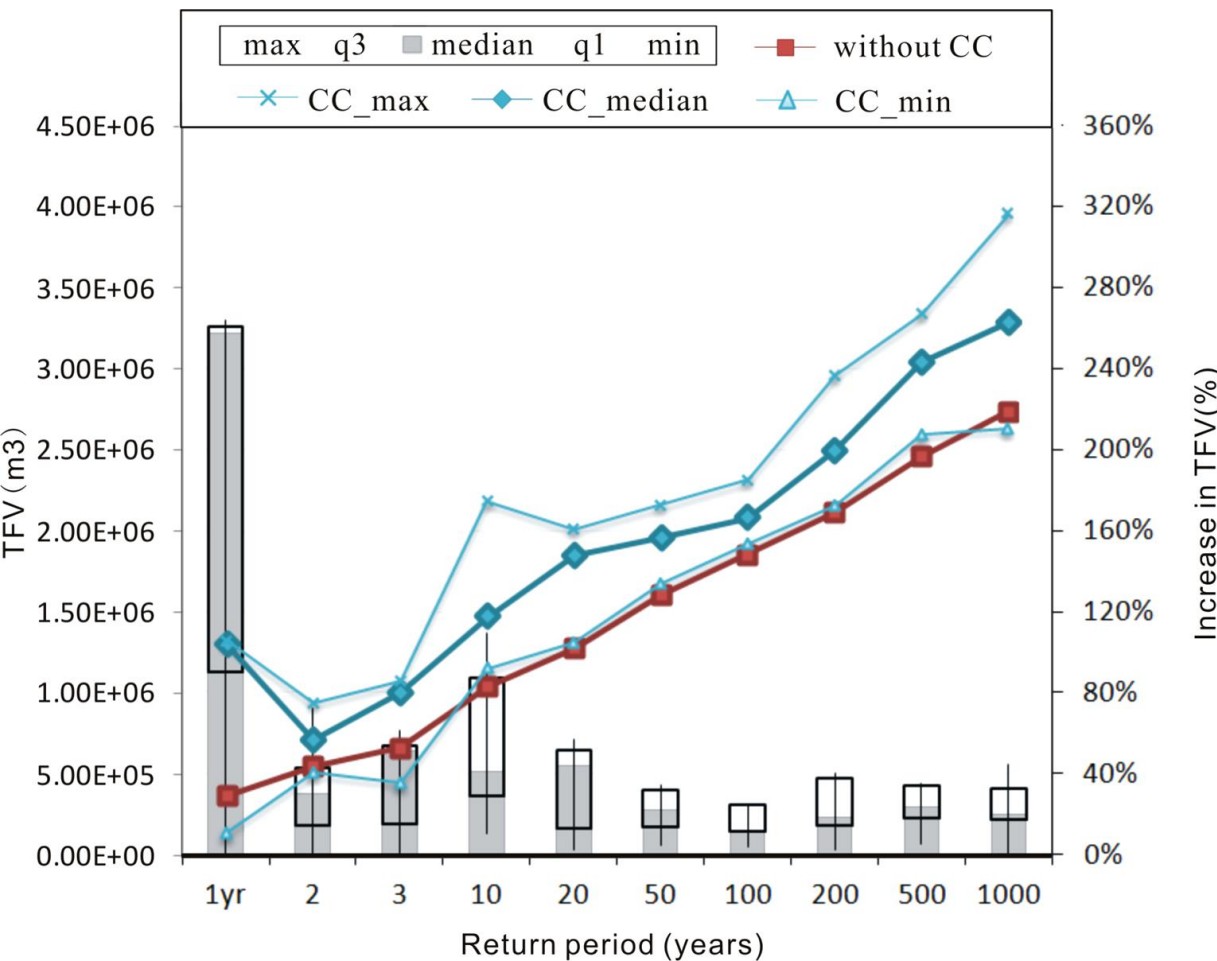

**Figure 3** Projected TFV with changes in precipitation intensity at various return periods under
the RCP8.5 scenario for the period of 2020–2040.

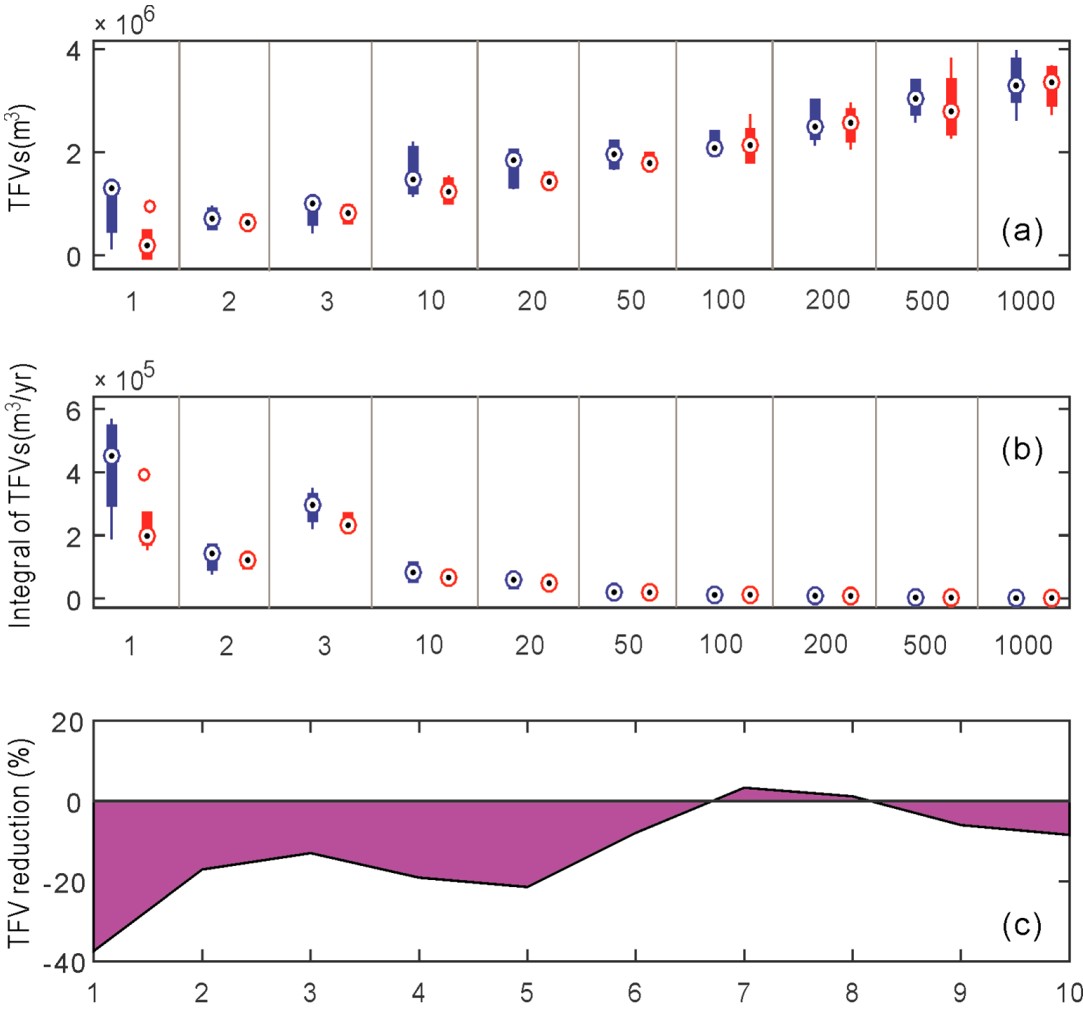


**Figure 4** Comparison of (a) flood volume, (b) total TFVs (i.e., the piece-wise integral of flood
volume versus the expected frequency with changes in precipitation intensity of various return
periods under RCP8.5 (blue) and RCP2.6 (red). (c) is for the reduced TFVs in percentage (i.e.,
benefits of climate mitigation) in RCP2.6 relative to RCP8.5 at various return periods.

714

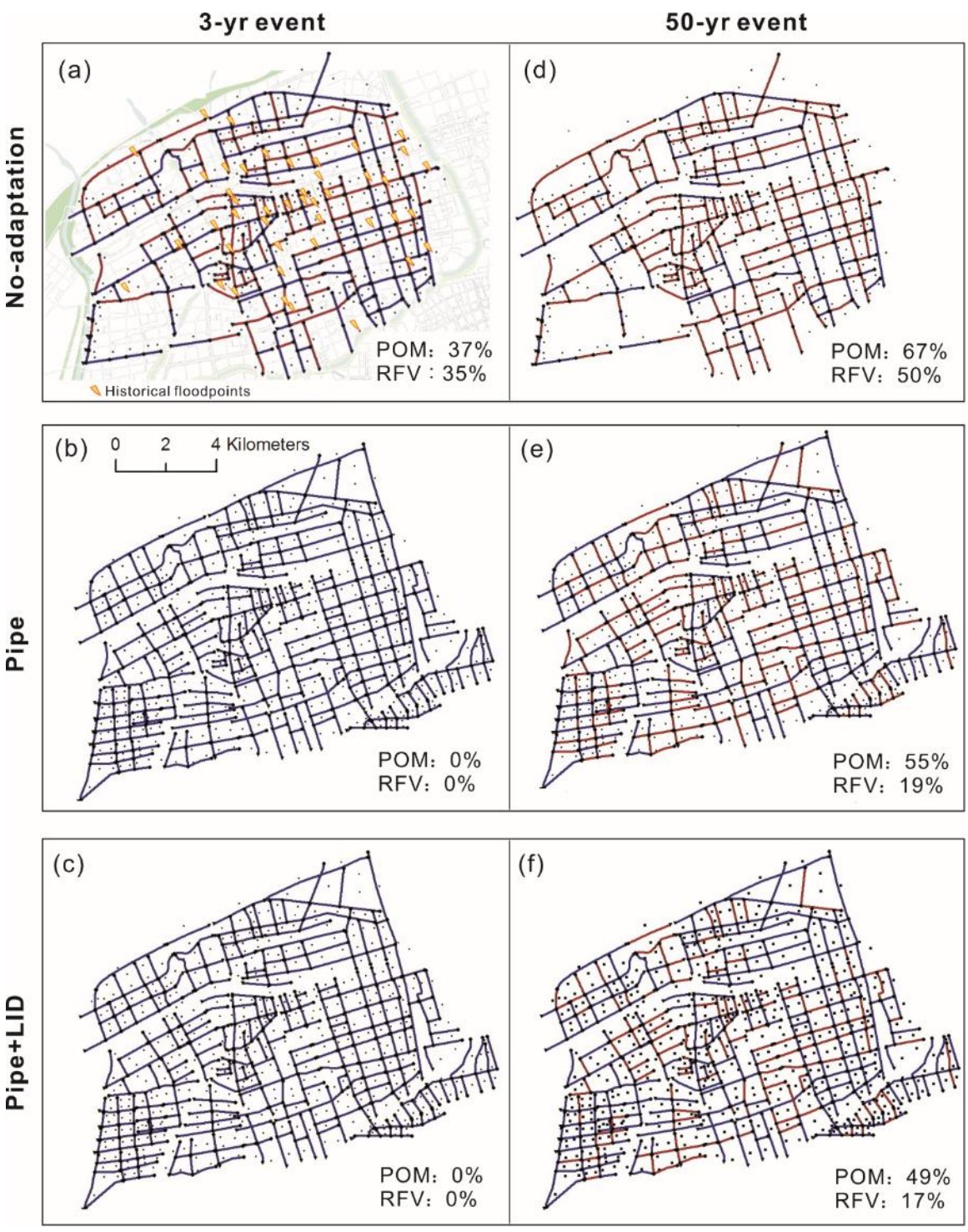

715

**Figure 5** Spatial distribution of overloaded pipelines (red colour) induced by the 3-year (left column) and 50-year extreme events (right column) without and with adaptations. The total percentage of overloaded manholes (POM) and ratio of flood volume (RFV) are summarised for each scenario. Descriptions of local land use, mainly the traffic network and green spaces, are provided as the background image in (a).


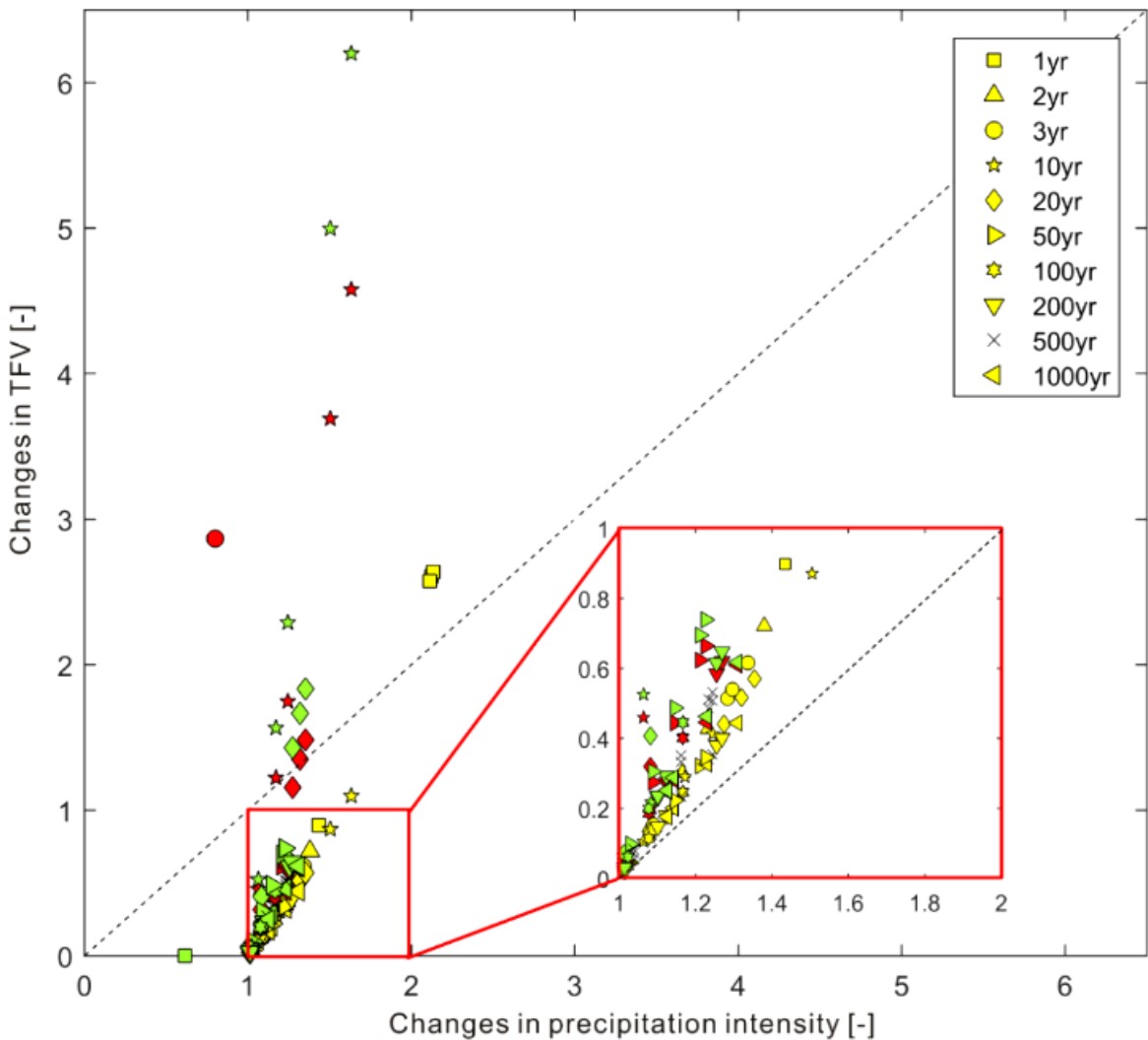


**Figure 6** Future changes in flood volumes (CTFVs) relative to historical conditions under the current drainage system (yellow) and two adaptation scenarios (i.e., Pipe in red and Pipe+LID in green) at various return periods.


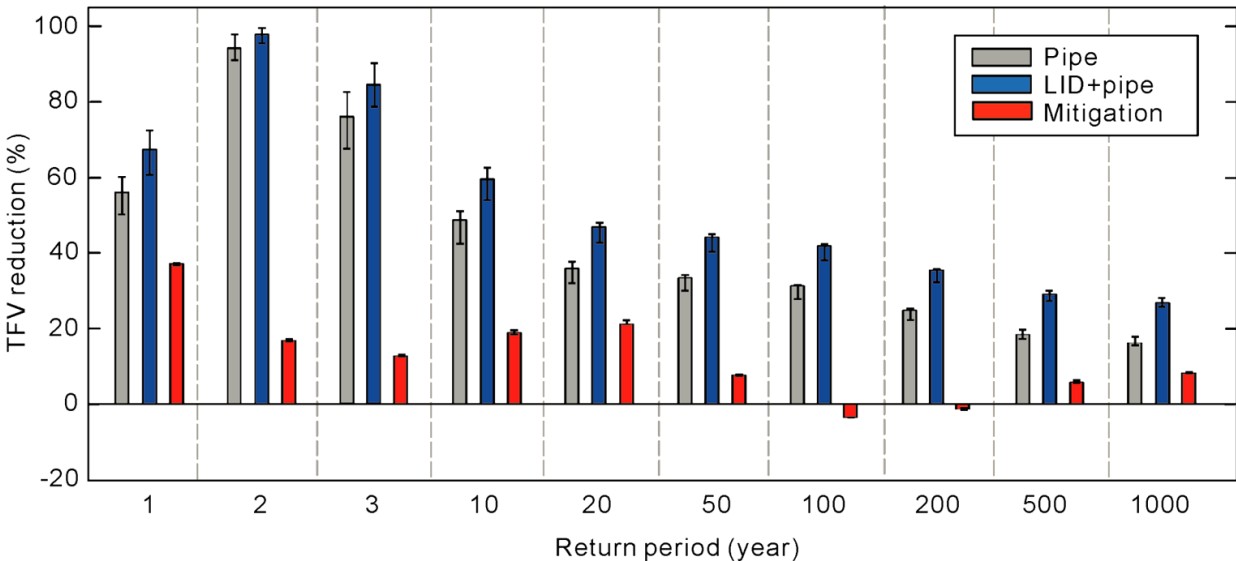


**Figure 7** Comparison of benefits of climate mitigation and two adaptation strategies in reducing
urban flood volumes with changes in precipitation intensities for various return periods, and with
related variations (boundary bars) as a result of uncertainty arising from local soil conditions.