# Peer review of "Impacts of future climate change on urban flood volumes in Hohhot City in Northern China: benefits of climate mitigation and adaptations"

_Hydrology and Earth System Sciences, 2016_

## Referee Comment (RC1) · Anonymous Referee #1 · 11 Nov 2016

The article by Zhou at al tackles a very topical issue in the field of flood risk assessment, which deals with climate change, mitigation and adaptation measures. The research questions that the authors investigate is sound and meaningful, and it is particularly interesting as the benefits of adaptation and mitigation measures are evaluated numerically through a modelling framework (though their associate cost is not assessed). Now the bad news: the structure of the article is sometimes not so clear, due to missing links, lack of details in the methods, questionable assumptions and unclear interpretation of results. Also, the use of English, although sufficient, is sometimes sub-optimal, and could do with a revision by a native speaker. Please pay careful attention to the use of prepositions and of the "s" for plurals. I found a number of mistakes and inappropriate use. Nonetheless, I think that the article had good potential for being published, provided that the following comments are adequately addressed. Please pay special attention to the general comments, where substantial work is needed to improve parts of the description of methods, assumptions and evaluation of results.

General comments

L 131-146: I would like to see some comments by the authors on the suitability of CMIP5 data for studies on urban flooding. Given the coarse resolution of CMIP5 (as they are global models), I'm sure that the entire study region is considerably smaller than 1 model grid cell. This poses some questions on how well extreme precipitation for modeling urban flooding is adequately represented by such datasets, given that such models are not able to represent local and short-lived storms commonly inducing flooding in small catchments. Intuitively one would say that downscaled projections with high resolution would be more suitable for this work, though that clearly depends on the data availability. Perhaps the authors can comment on that.

L 169-182: I suggest expanding this section as I think there are some unclear points which prevents the reader from understanding some modeling steps, underlying assumptions, as well as from making the approach reproducible. For example, is q in eq. 1 the peak intensity? Which is the temporal resolution considered? Most climate datasets have 1 day as highest temporal resolution, but that would probably be rather coarse for urban flooding applications. How are then the hyetographs calculated from the q? Is it a simple rescaling based on their peak, keeping the same shape? Also, I see a lack of information on how climatic data is handled statistically to estimate storms/volumes with selected return period between 1 and 1000 years. For example, I see that the considered period for assessing future scenarios is 2020-2040, hence 21 years of data. Does it mean that return periods in the order of 1000 years are estimated from 21 years of data? Could the authors clarify on this? Can they provide ranges of uncertainty due to the undersampling of the climate variability in such long periods? Also, this should be mentioned in Sect. 4 as a further uncertainty source.

[Figure]

Final comment is about eq. 1: could you briefly comment on how the parameters A, b, c, D are valid under a non-stationary climate? 4 parameters and just 2 variables sounds a lot for an empirical formula.

L235-250: Despite the authors' efforts to link the flood volume with flood risk and damage, I find inappropriate to call results in Figure 4 as "risk" and "damage". There is clearly a missing step in linking flood volume with some socio-economic indicator on the impact of floods. This also results in a biased evaluation of what is called "flood risk", which suggests in Figure 4 that the largest contribution is given by floods with 1-2 year return period. In reality, it may well be that a single 100-year flood induces a damage which is larger than 100 1-year floods. For this reason, I do not agree with the statement in lines 239-242. The authors should definitely clarify this part and spend some words on what are the consequences of their assumptions, if that is retained at all. In addition, the authors should clarify the relations between Fig 4a and 4b. I have the feeling that values in 4b are simply obtained by dividing numbers in 4a by their theoretical expected annual frequency indicated below each column. This would be incorrect as in this way you would be double counting all probabilities smaller than each considered class. You should instead apply the formula for piece-wise integral of flood damage versus the expected frequency of each class, hence considering the width of each bar (e.g., for the second column is 1/2-1/3, for the third one is 1/3 -1/10 and so forth).

L 265-286: I find this part rather difficult to understand and suggest the authors to clarify some points and describe more thoroughly Figure 6 and its usefulness. First, the way changes (CTFV) are defined is not intuitive, as it is now defined as a multiplicative factor. Changes should be CTFV=(TFVc-TFVnc)/TFVnc. Also, why the current system is less sensitive to climate change than the adapted system (l 268-269)? I'm a bit puzzled by seeing that small changes in the 10-year precipitation intensity lead up to a 7-fold increase in TFV under the case of adaptation. Does it mean 7 times worse conditions or simply that the adapted system can hold more water, also because the

catchment area is larger? Then I get confused on the definition of TFV: is it the total volume or simply the excess volume after filling completely the pipes system? I thought it's the second option, but now I'm confused. Please clarify in sect. 2c. In both cases it's difficult to assess how worse the conditions (i.e., the damage) would be under larger TFV in the adapted system, though I think a graph with such information is currently missing and could be added. Finally, please avoid 4 decimals in numbers at lines 270-271; 2 decimal digits are surely enough.

Specific comments

L 31: given the delay between submission and publishing I suggest removing "current" from the text. Same for line 81.

L 32: I suggest removing "existing" in favor of "past", "recent," "literature" or similar

L 40: "Based on the results" –> "Results indicates that"

L45: This is an outcome of your research, hence I would not say it is "obvious" but rather something like "very likely" or "results clearly indicates..." or similar.

L 46: "greenhouse gas emissions"

L 62: The sentence is not clear. Please specify units of the change and in relation to what (e.g., flood peak, precipitation intensity?)

L 66-69 is again not clear. E.g., non-stationary changes reads awkward. Also, what do you mean by future hydroclimate?

L71-77: As the article has a strong focus on mitigation and adaptation I suggest adding some relevant references in those areas. See the work by (Alfieri et al., 2016; Arnbjerg-Nielsen et al., 2015; Moore et al., 2016; Poussin et al., 2012) among others. The few ones currently listed in the article are somehow hidden in the conclusions.

L136-137: the sentence is currently hard to read. Please reformulate.

L 140-144: The sentence is rather misleading, first because there is now a wealth of studies using ensembles of several GCMs, and second because "all five GCMs" sounds like if there were only five, while CMIP5 includes way more than that.

L 151: Rainfall is a climatic data. Please clarify.

L 176: there are –> we considered

L 181-182: This sentence should be supported by data, graphs or a reference to publications showing the validation work against historical records.

L 186-191: This part is difficult to read and understand. Please clarify and add some detail on how the TFV – return period relationship was derived. Figure 2 currently doesn't help a lot as it is too general, with no units nor tick marks. For example, if it the grey area is meant to indicate those events that contribute the most to the annual damage, then it should take at least 50% of the area under the curve in Figure 2, as its integral is proportional to the total flood risk.

L 191- 195: This statement indicates a strong assumption which is not justified at this stage and sounds like a speculation. Perhaps the authors want to introduce what is later on indicated by their findings, but I think at this point this is unjustified, unless the point is supported by stronger evidence and/or some references.

L204-205: What is the extent of the enhancement of pipeline diameters in the adapted scenario? I couldn't find it anywhere in the text.

L230-231: Is this 52% a simple average of the percent changes shown in Figure 3? Then I suggest to clarify, as it doesn't necessarily mean the overall projected change in flood risk.

L 254: More correctly "10 magnitudes of rainfall events".

L 263: 19% should be 49%.

L 332-333: Not just uncertainties but modeling assumptions as well.

L 328-329: That's true but perhaps out of the scope of this article, as anyways there is no real damage model to evaluate economic flood losses.

L 358-363: Following the discussions above one should be careful in calling these numbers "flood risk". Please adapt according to the indications in the discussion points above.

L 605-606: I suggest including the period "2020-2040" in the caption for better understanding the graph.

Table 1: Which are the units in the table? Please specify units and the storm duration related to the precipitation intensity values listed (key parameter to understand such values).

Figure 5: Please choose a more visible way of indicating overloaded pipelines, perhaps with a thicker line and/or a different color. Also the POM is currently mistakenly written as "NOM" in the 6 panels.

Figure 6: Add units in the axis labels. E.g.: "[-]" for dimensionless. Also, note the typo in the x-axis label.

Figure 7: Negative values for risk reduction means increasing risk. Please reverse graphs with positive values (plus fix the typo rish -> risk)

References

Alfieri, L., Feyen, L. and Baldassarre, G. D.: Increasing flood risk under climate change: a pan-European assessment of the benefits of four adaptation strategies, Clim. Change, 136(3), 507–521, doi:10.1007/s10584-016-1641-1, 2016.

Arnbjerg-Nielsen, K., Leonardsen, L. and Madsen, H.: Evaluating adaptation options for urban flooding based on new high-end emission scenario regional climate model simulations, Clim. Res., 64(1), 73–84, doi:10.3354/cr01299, 2015.

Moore, T. L., Gulliver, J. S., Stack, L. and Simpson, M. H.: Stormwater management

and climate change: vulnerability and capacity for adaptation in urban and suburban contexts, Clim. Change, 138(3–4), 491–504, doi:10.1007/s10584-016-1766-2, 2016.

Poussin, J. K., Bubeck, P., H. Aerts, J. C. J. and Ward, P. J.: Potential of semi-structural and non-structural adaptation strategies to reduce future flood risk: Case study for the Meuse, Nat. Hazards Earth Syst. Sci., 12(11), 3455–3471, doi:10.5194/nhess-12-3455-2012, 2012.

---

## Referee Comment (RC2) · Anonymous Referee #2 · 27 Feb 2017

SHORT COMMENTS IN THE JOURNAL STYLE Scientific questions. Adaptation effects on drainage performance in a context of climate change (CC) is relevant. Novel concepts. Try to quantify the impact adaptation measures is potenti<lly new if appropriately developed in single case studies. Substantial conclusions. Not attended yet, due to insufficiently explained datasets and methods. Scientific methods and assumptions. Not clearly outlined. Results vs interpretations/conclusions. Unattended. Description. Pretty obscure. Authors proper credit. Ok! but not all is new. Title. OK! but to be revised in case of revision. Summary. Unbalanced on Climate trends when the most interesting part is adaptation. Overall presentation. Lacking of context outline. Language. To be revised by a mothertongue, that I am not. Formulae. Not expert enough

to say. Parts to modify. Develop 1, 4 & 5, Clarify 2a & 2e, Reduce 2b, Delete 3b, Modify Fig. 1 & 5. References. Ok.

EXTENDED COMMENTS

1 Introduction All key definitions should be provided here. Flood riskis the probability an hazard has to generate damages (UNISDR, ISO etc...), not a probability of a disastrous flood only (that is hazard occurrence). Should be wise to specify to whom this work is addressed, since very essential information of the case study is missing (see next sections).

2. Material and Methods 2a) (i) A characterization of the hazard (rainfall) in Hohhot City ismissing. (ii) A detailed description of watershed soils is reccomended. Rocky, lateritic, clay, sandy, or... soils perform differently in semi-arid contexts than in wet contexts. Even where infiltration seems possible some pervious looking soils after the first minutes turn into impervious. Context matter in this type of study. (iii) Authors consider permeable pavements, infiltration trenches and green roofs as possible adaptation measures. Which are the permeable soil and coverage rates in the different parts of the watershed considered?

(iv) Can the authors provide some information about last disastrous floods in the case study? Areas affected the most, etc.

2b) (i) It's quite normal to use more than one GCMs . (ii) Reader expects to learn from the expected changes in rainfall (mm and in which month) but no information is provided on this topic.

2c) (i) Which rainfall information has been used to run SWMM [dataset length (years) and type (daily, three hourly, hourly, etc.)]?

2e) (i) The adaptation measures considered are to reduce the amount of water that run off. This is one side of the problem. The other one is to slow down the water speed. And for this no measure is considered: there is a wide range of measures for semi-arid

contexts commonly used for this. I recommend to consider it or explain why you don't.

(ii) How Authors have determined the impact of individual adaptation measures (permeable pavements, trenches, green roof) over run off reduction? This should be explained.

3) Results 3b) (i) I don't understand the approach: Mitigation is expected to impact on CC at long term (decades. . .). Drainage system is expected to reduce CC impacts at short-medium term (1-5 years). Is obvious that adapting we can't expect to see effects on rainfall. . .

4) Uncertainities & Limitations (i) The consideration of the state of drainage system could be a limitation of this study? A drainage system obstructed by vegetation, waste or artefacts (cables, pipes, temporary constructions) can make the outcomes of the SWMM quite distant from the real world. And change also recommendations. . . that need to be extended to waste sector.

5. Could the Authors consider to show us what is their way forward?

Figures 1 & 5: scale is not showed: how large are blocs contoured by drainage network?

0. Manuscript's title Show the name of the case study and the country. Limit to Adaptation, delete mitigation, delete risk.

END

---

## Author Comment (AC1) · 27 Mar 2017

**Impacts of future climate change on urban flood risks: benefits of climate mitigation and adaptations [MS No.: hess-2016-369]**

**Responses to review comments**

**REFEREE REPORT(S):**

**Anonymous Referee #1:**

The article by Zhou at al tackles a very topical issue in the field of flood risk assessment, which deals with climate change, mitigation and adaptation measures. The research questions that the authors investigate is sound and meaningful, and it is particularly interesting as the benefits of adaptation and mitigation measures are evaluated numerically through a modelling framework (though their associate cost is not assessed). Now the bad news: the structure of the article is sometimes not so clear, due to missing links, lack of details in the methods, questionable assumptions and unclear interpretation of results. Also, the use of English, although sufficient, is sometimes sub-optimal, and could do with a revision by a native speaker. Please pay careful attention to the use of prepositions and of the "s" for plurals. I found a number of mistakes and inappropriate use. Nonetheless, I think that the article had good potential for being published, provided that the following comments are adequately addressed. Please pay special attention to the general comments, where substantial work is needed to improve parts of the description of methods, assumptions and evaluation of results.

**Response**: *We greatly appreciate the reviewer for the constructive comments and suggestions to improve our manuscript. In the revision, we have 1) added more details on the datasets and methods, 2) added more discussions on the assumptions and limitations, 3) modified the relevant statements and figures which are unclear or inaccurate, 4) invited a native speaker to proof-read the paper. More details of our responses to each comment are provided as follows.*

**General comments**

L 131-146: I would like to see some comments by the authors on the suitability of CMIP5 data for studies on urban flooding. Given the coarse resolution of CMIP5 (as they are global models), I'm sure that the entire study region is considerably smaller than 1 model grid cell. This poses some questions on how well extreme precipitation for modeling urban flooding is adequately represented by such datasets, given that such models are not able to represent local and short-lived storms commonly inducing flooding in small

catchments. Intuitively one would say that downscaled projections with high resolution would be more suitable for this work, though that clearly depends on the data availability. Perhaps the authors can comment on that.

**Response**: *Thanks for the comments. As pointed out by the reviewer, bias would exist in global climate model (GCM) simulations especially at the local and regional scales. An alternative approach is to simulate the future climate using regional climate model (RCM) nested within a GCM. Such climate projections by RCM have added value in terms of higher spatial resolution which can provide more detailed regional information. However, various level of bias would still remain in RCM simulations (Teutschbein and Seibert 2012) and bias correction of RCM projections are required, e.g. the European project ENSEMBLES (Hewitt and Griggs 2004; Christensen et al. 2008). To run regional climate model is not within the scope of this study. Instead, we tend to use publicly available climate projection dataset. Here, we obtain climate projections from the ISI-MIP (Warszawski et al. 2014), which provides spatially-downscaled climate data for impact models. The climate projections were also bias-corrected against observations (Hempel et al. 2013) and have been widely used in climate change impact studies on hydrological extremes such as floods and droughts (e.g. Dankers et al. 2014; Prudhomme et al. 2014; Giuntoli et al. 2015; Leng et al. 2015).*

*It should be noted that we used the delta change factor to derive the climate scenarios as inputs into our flood drainage model instead of using the climate projections directly. Specifically, we calculate the change factor between current and future climate projection simulated by GCMs and multiply them to observed time series to derive future climate scenario into our flood drainage model. This is because the relative climate change signal simulated by GCMs are argued to be more reliable than the simulated absolute values (Ho et al. 2012). What's more, we use an ensemble of GCM simulations rather than one single climate model in order to characterize the uncertainty range arising from climate projections. In the revision, we have added more discussions on this.*

*Reference*
*Warszawski, L., Frieler, K., Huber, V., Piontek, F., Serdeczny, O., & Schewe, J. (2014). The inter-sectoral impact model intercomparison project (ISI–MIP): project framework. Proceedings of the National Academy of Sciences, 111(9), 3228-3232.*

*Dankers, R., Arnell, N. W., Clark, D. B., Falloon, P. D., Fekete, B. M., Gosling, S. N., ... & Stacke, T. (2014). First look at changes in flood hazard in the Inter-Sectoral Impact Model Intercomparison Project ensemble. Proceedings of the National Academy of Sciences, 111(9), 3257-3261.*

*Prudhomme, C., Giuntoli, I., Robinson, E. L., Clark, D. B., Arnell, N. W., Dankers, R., ... & Hagemann, S. (2014). Hydrological droughts in the 21st century, hotspots and uncertainties from a global multimodel ensemble experiment. Proceedings of the National Academy of Sciences, 111(9), 3262-3267.*

*Leng, G., Tang, Q., & Rayburg, S. (2015). Climate change impacts on meteorological, agricultural and hydrological droughts in China. Global and Planetary Change, 126, 23-34.*

*Giuntoli, I., Vidal, J. P., Prudhomme, C., Hannah, D. M. (2015). Future hydrological extremes: the uncertainty from multiple global climate and global hydrological models. Earth System Dynamics, 6(1), 267.*

*Teutschbein, C., & Seibert, J. (2012). Bias correction of regional climate model simulations for hydrological climate-change impact studies: Review and evaluation of different methods. Journal of Hydrology, 456, 12-29.*

*Hempel, S., Frieler, K., Warszawski, L., Schewe, J., & Piontek, F. (2013). A trend-preserving bias correction–the ISI-MIP approach. Earth System Dynamics, 4(2), 219-236.*

*Christensen, J. H., Boberg, F., Christensen, O. B., & Lucas‑Picher, P. (2008). On the need for bias correction of regional climate change projections of temperature and precipitation. Geophysical Research Letters, 35(20).*

*Hewitt, C. D., and D. J. Griggs (2004), Ensembles-based predictions of climate changes and their impacts, Eos Trans. AGU, 85, 566.*

*Ho, C. K., Stephenson, D. B., Collins, M., Ferro, C. A., & Brown, S. J. (2012). Calibration strategies: a source of additional uncertainty in climate change projections. Bulletin of the American Meteorological Society, 93(1), 21.*

L 169-182: I suggest expanding this section as I think there are some unclear points which prevents the reader from understanding some modeling steps, underlying assumptions, as well as from making the approach reproducible. For example, is q in eq. 1 the peak intensity? Which is the temporal resolution considered? Most climate datasets have 1 day as highest temporal resolution, but that would probably be rather coarse for urban flooding applications. How are then the hyetographs calculated from the q? Is it a simple rescaling based on their peak, keeping the same shape? Also, I see a lack of information on how climatic data is handled statistically to estimate storms/volumes with selected return period between 1 and 1000 years. For example, I see that the considered period for assessing future scenarios is 2020-2040, hence 21 years of data. Does it mean that return periods in the order of 1000 years are estimated from 21 years of data? Could the authors clarify on this? Can they provide ranges of uncertainty due to the undersampling of the climate variability in such long periods? Also, this should be mentioned in Sect. 4 as a further uncertainty source. Final comment is about eq. 1: could you briefly comment on how the

parameters A, b, c, D are valid under a non-stationary climate? 4 parameters and just 2 variables sounds a lot for an empirical formula.

**Response**: *Thanks for the comments. In this study, we adopt the storm intensity formula (SIF) to derive the precipitation input into our drainage model. The SIF is a standard approach for rainfall design in urban drainage modeling in China, as well documented in the National Guidance for the Design of Outdoor Wastewater Engineering (MOHURD, 2011). Specifically, the SIF is used to describe an Intensity-Duration-Frequency (IDF) relationship, which is well used in the literature for estimating rainfall design hydrographs through the Chicago Design Storms (CDS) approach (Berggren et al., 2014; Cheng and AghaKouchak, 2014; Panthou et al., 2014; Willems, 2000; Zhou et al., 2012). More details can refer to Smith (2004) for the derivation of CDS from an IDF relationship. In China, the procedures for applying SIF to obtain CDS design storms are outlined in the National Technical Guidelines for Establishment of Intensity-Duration-Frequency Curve and Design Rainstorm Profile (MOHURD, 2014) and have been well adopted for Chinese urban drainage designs (Wu et al., 2016; Yin et al., 2016; Zhang et al., 2008; Zhang et al., 2015). Therefore, the method for using the SIF to generate CDS design storms for our SWMM modeling study is reproducible and valid for drainage modeling.*

*The technical details of SIF and derivation of CDS rainfall are given as follows. As shown in the Equation 1, the q is the average rainfall intensity, t is the storm duration and P is the design return period. The typical temporal resolution in SIF is minutes for urban drainage modeling. A, b, c and D are the regional parameters governing the IDF relations among rainfall intensity, return period and storm duration. For a given return period, the SIF can be fitted into the Horner's equation (2004) as shown in Equation 2:*

$$q = \frac{A(1 + Dlg(P))}{(t + b)^c} \qquad\qquad Eq.\ (1)$$

$$i = \frac{a}{(t + b)^c} \qquad\qquad Eq.\ (2)$$

*The synthetic hyetograph based on the Chicago method is computed using Equation 2 and an additional parameter r (where 0< r <1) which determines the relative location of peak intensity (with respect to time), $t_p=r*t$. The time distribution of rainfall intensity is described after the peak $t_a = (1-r)*t$ and before the peak $t_b=r*t$ by Equation (3) and (4), respectively. Specially, $i_b$ is the instantaneous rainfall intensity before the peak, and $i_a$ is the instantaneous rainfall intensity after the peak.*

$$i_a = \frac{a[\frac{(1-c)t_a}{(1-r)} + b]}{(\frac{t_a}{(1-r)} + b)^{1+c}}$$

Eq. (3)

$$i_b = \frac{a[\frac{(1-c)t_b}{r} + b]}{(\frac{t_b}{r} + b)^{1+c}}$$

Eq. (4)

*In this study, we considered 10 return periods, i.e., the 1, 2, 3, 10, 20, 50, 100, 200, 500 and 1000-year events. A 4-hour rainfall time series is generated for each return period at a 10-minute interval based on Equations 1-4. The A, b, c and D parameters governing the SIF shape were obtained from the local weather bureau, which fits the historical precipitation distribution for the study region. In the revision, we have added more details about the methods.*

*As for the generation of future climate scenarios, we first calculate the change factor for each return period. Specifically, for each year, the annual maximum daily precipitation was determined for both historical and future periods. Then, the generalized extreme value (GEV) distribution is fitted separately to the two sets of daily values (Coles 2001; Katz et al. 2002). The goodness of fit was tested by calculating the Kolmogorov–Smirnov and Anderson–Darling statistics. The value corresponding to each return period is derived based on the GEV distribution and the changes between future and historical periods are calculated as the change factors (as shown in Table 1 in the text). The change factor for each return period is then multiplied to the historical design CDS rainfall time series to derive future climate scenarios for the model. We acknowledge that to estimate the changes in extreme precipitation events involves inevitable uncertainties especially for return periods beyond the length of the data, e.g. 1000yrs as pointed by the reviewer. Hence, caution should be exercised when interpreting the results for return levels beyond the data length. However, we'd like to mention that "return period" is intrinsically a statistical measurement derived based on probability density function (PDF) of historical data in extended period. That is, it represents a recurrence interval which is an estimate of the likelihood of an event (in our case, a flood) indicated by the PDF. Depending on the historical period used, the return period could vary if the time series is not stationary. Nevertheless, a 1000-year return period can be derived from 21-year time series based on its definition by using a PDF. We have added discussions on this in the revision.*

*We agree that climate variability range would be under-sampled, although five climate models are used to show the possible ranges. In the revision, we use the boot-strap sampling technique to address the uncertainty range of under-sampling climate variability. We have added discussions on this in the revision.*

*The parameters A, b, c, D are derived from sub-hourly rainfall data and provided by local weather bureau. The four parameters which describe the Intensity-Duration-Frequency (IDF) relationship in the study region are assumed to be constant without considering its non-stationary features in a changing climate. To derive the parameter in the future period requires hourly precipitation data, which are not readily available. Hence, the IDF relationship is assumed to remain stable in the future and only changes in the daily mean intensity are considered. Given the above limitations, we acknowledge that our modeling results mainly represent the first-order potential climate change impacts on urban floods. Future efforts should be devoted to the representation of dynamic rainfall changes at hourly time step taking into account of non-stationary climate change. We have added more discussions in the revised manuscript.*

*Reference:*

*Berggren, K., Packman, J., Ashley, R., and Viklander, M.: Climate changed rainfalls for urban drainage capacity assessment, Urban Water Journal, 11, 543-556, 10.1080/1573062X.2013.851709, 2014.*

*Cheng, L. Y., and AghaKouchak, A.: Nonstationary Precipitation Intensity-Duration-Frequency Curves for Infrastructure Design in a Changing Climate, Scientic Report, 4, 10.1038/srep07093, 2014.*

*Coles S (2001) An Introduction to Statistical Modeling of Extreme Values, Springer Series in Statistics (Springer, London).*

*Katz RW, Parlange MB, Naveau P (2002) Statistics of extremes in hydrology. Adv Water Resour 25(8-12):1287–1304*

*MOHURD: AQSIQ. Code for Design of Outdoor Wastewater Engineering (GB 50014-2006), Ministry of Housing and Urban-Rural Development, General Administration of Quality Supervision, Inspection and Quarantine of the People's Republic of China: Beijing, China (In Chinese), 2011.*

*MOHURD: Technical Guidelines for Establishment of Intensity-Duration-Frequency Curve and Design Rainstorm Profile (In Chinese), Ministry of Housing and Urban-Rural Development of the People's Republic of China and China Meteorological Administration. Accessed on November 2016 from http://www.mohurd.gov.cn/wjfb/201405/W020140519104225.pdf, 2014.*

*Panthou, G., Vischel, T., Lebel, T., Quantin, G., and Molinie, G.: Characterising the space-time structure of rainfall in the Sahel with a view to estimating IDAF curves, Hydrology and Earth System Sciences, 18, 5093-5107, 10.5194/hess-18-5093-2014, 2014.*

*Smith, A. A.: MIDUSS Version 2, Reference Manual, Version 2.00 Rev2.00., Alan A. Smith Inc., Dundas, Ontario, Canada, 2004.*

*Willems, P.: Compound intensity/duration/frequency-relationships of extreme precipitation for two seasons and two storm types, Journal of Hydrology, 233, 189-205, http://dx.doi.org/10.1016/S0022-1694(00)00233-X, 2000.*

*Wu, H., Huang, G., Meng, Q., Zhang, M., and Li, L.: Deep Tunnel for Regulating Combined Sewer Overflow Pollution and Flood Disaster: A Case Study in Guangzhou City, China, Water, 8, 329, 2016.*

*Yin, J., Yu, D. P., Yin, Z., Liu, M., and He, Q.: Evaluating the impact and risk of pluvial flash flood on intra-urban road network: A case study in the city center of Shanghai, China, Journal of Hydrology, 537, 138-145, 10.1016/j.jhydrol.2016.03.037, 2016.*

*Zhang, Y.-q., Lv, M., and Wang, Q.-g.: Formula method design of drainage pipe network and analysis of model simulation, Water Resour. Power, 33, 105-107, 2015.*

*Zhang, D., Zhao, D. q., Chen, J. n., and Wang, H. z.: Application of Chicago approach in urban drainage network modeling, Water & Wastewater Engineering, 34, 354-357, 2008.*

*Zhou, Q., Mikkelsen, P. S., Halsnaes, K., and Arnbjerg-Nielsen, K.: Framework for economic pluvial flood risk assessment considering climate change effects and adaptation benefits, Journal of Hydrology, 414, 539-549, 10.1016/j.jhydrol.2011.11.031, 2012.*

L235-250: Despite the authors' efforts to link the flood volume with flood risk and damage, I find inappropriate to call results in Figure 4 as "risk" and "damage". There is clearly a missing step in linking flood volume with some socio-economic indicator on the impact of floods. This also results in a biased evaluation of what is called "flood risk", which suggests in Figure 4 that the largest contribution is given by floods with 1-2 year return period. In reality, it may well be that a single 100-year flood induces a damage which is larger than 100 1-year floods. For this reason, I do not agree with the statement in lines 239-242. The authors should definitely clarify this part and spend some words on what are the consequences of their assumptions, if that is retained at all. In addition, the authors should clarify the relations between Fig 4a and 4b. I have the feeling that values in 4b are simply obtained by dividing numbers in 4a by their theoretical expected annual frequency indicated below each column. This would be incorrect as in this way you would be double counting all probabilities smaller than each considered class. You should instead apply the formula for piece-wise integral of flood damage versus the expected frequency of each class, hence considering the width of each bar (e.g., for the second column is 1/2-1/3, for the third one is 1/3 -1/10 and so forth).

**Response**: *Thanks for the comments. We agree with the reviewer that results in Figure 4 refer to the flood volume rather than "damage" or "risk" due to the missing linkage to the socio-economic conditions. We also agree that a single 100-year flood event could have larger impacts than 100 1-year floods. In the revision, we have deleted the word "damage" or "risk" and revised the statements in lines 239-242 and other relevant statements accordingly. The original Figure 2 which is used to illustrate the conceptual flood risks is also revised.*

*Following the suggestion by the reviewer, we have revised the Figure 4b to show the piece-wise integral of flood volume corresponding to each frequency class (e.g. the width of first class is 1/1-1/2 and so forth).*

[Figure]

***Figure 4*** *Comparison of (a) flood volume, (b) total TFVs (i.e., the piece-wise integral of flood volume versus the expected frequency with changes in precipitation intensity of various return periods under RCP8.5 (blue) and RCP2.6 (red). (c) is for the reduced TFVs in percentage (i.e., benefits of climate mitigation) in RCP2.6 relative to RCP8.5 at various return periods.*

L 265-286: I find this part rather difficult to understand and suggest the authors to clarify some points and describe more thoroughly Figure 6 and its usefulness. First, the way changes (CTFV) are defined is not intuitive, as it is now defined as a multiplicative factor. Changes should be CTFV=(TFVc-TFVnc)/TFVnc. Also, why the current system is less sensitive to climate change than the adapted system (l 268-269)? I'm a bit puzzled by seeing that small changes in the 10-year precipitation intensity lead up to a 7-fold increase in TFV under the case of adaptation. Does it mean 7 times worse conditions or simply that the adapted system can hold more water, also because the catchment area is larger? Then I get confused on the definition of TFV: is it the total volume or simply the excess volume after filling

completely the pipes system? I thought it's the second option, but now I'm confused. Please clarify in sect. 2c. In both cases it's difficult to assess how worse the conditions (i.e., the damage) would be under larger TFV in the adapted system, though I think a graph with such information is currently missing and could be added. Finally, please avoid 4 decimals in numbers at lines 270-271; 2 decimal digits are surely enough.

**Response**: *Thanks for the comments. We are sorry for the confusion. The TFV is defined as the total volume flooded from manholes without taking into account the outlet discharges, i.e., the excess water after filling completely the pipe system. As pointed out by the reviewer, the current drainage system is less sensitive to climate change. This is because the capacity of current drainage system is small, i.e. the excess water after filling completely the pipe system (i.e., TFVnc) is large. Given extreme rainfall events, the current system would be flooded completely, thus exhibiting less sensitivity to larger extreme rainfall events in the future. Therefore, the magnitude of changes in excess flood volume is smaller in the current system than the adapted system due to its large value of denominator in the calculation of CTFV (CTFV=(TFVc-TFVnc)/TFVnc).*

*In order to better clarify this point, we have provided a table below summarizing the flood volumes of current and adapted drainage systems, with and without climate change. It is evident that for the present time, the flood volume of the adapted systems are much smaller than that in the current system due to capacity upgrades in the adapted systems to hold more water. For example, given a 10-year event, the flood volume for the present period (i.e., TFVnc) is 1041,230, 274,650 and 180,610 (m3) in current and the two adapted systems (highlighted in blue), respectively, while in the future period with climate change, the magnitude of flood volume (i.e., TFVc, highlighted in yellow) is similar among the three drainage systems. Therefore, the calculated changes in flooded volume (CTFV) in the future relative to the present period are much smaller in current system than adapted systems due to the larger value of denominator in the equation.*

*In the revision, we have 1) clarified the definition of TFV; 2) re-defined $CTFV=(TFV_c-TFV_{nc})/TFV_{nc}$ following the suggestion, and  updated Figure 6 accordingly (see Figure 6 below); 3) added more discussions on projected changes on TFV;  4) used 2 decimal digits for the numeric results throughout the text.*

*Based on the suggested formula, the calculated CTFV for the three systems are 0.41, 1.75 and 2.29, respectively. The larger CTFVs in the adapted systems does not mean the worsened conditions. Rather, it indicates that the capacity (i.e., service level) of adapted system tends to become lower with climate changes while the current system has already reached its peak capacity in the present period and thus shows small sensitivity to climate change.*

**Table 1**: *TFVs of current and adapted systems with and without climate changes*

| Return period | | 1 | 2 | 3 | 10 | 20 | 50 | 100 | 200 | 500 | 1000 |
|---|---|---|---|---|---|---|---|---|---|---|---|
| Current system | NC | 363434 | 545594 | 662399 | 1041230 | 1280598 | 1604223 | 1855559 | 2113083 | 2464388 | 2740033 |
| | C1 | 1311483 | 779030 | 1070807 | 1471180 | 1845707 | 2120890 | 2081960 | 2494516 | 3337794 | 3635804 |
| | C2 | 138358 | 625172 | 763944 | 1151120 | 1309407 | 1676813 | 2313744 | 2916433 | 3302794 | 3292205 |
| | C3 | 689945 | 710016 | 1003205 | 1343650 | 1447074 | 1819748 | 1922111 | 2424542 | 2907221 | 3224196 |
| | C4 | 1322311 | 939202 | 1020153 | 1948310 | 1942896 | 2158862 | 2312024 | 2961595 | 3040893 | 3957185 |
| | C5 | 1299874 | 508016 | 447533 | 2184984 | 2011414 | 1961587 | 2068387 | 2155563 | 2598096 | 2631549 |
| | | | | | | | | | | | |
| Pipe | NC | 0 | 0 | 0 | 274650 | 545548 | 902639 | 1191761 | 1454490 | 1825663 | 2107541 |
| | C1 | 579100 | 66820 | 307628 | 754782 | 1177608 | 1465530 | 1424433 | 1853479 | 2753620 | 3048692 |
| | C2 | 0 | 14683 | 58510 | 400927 | 576342 | 988731 | 1672038 | 2305916 | 2711960 | 2700636 |
| | C3 | 30911 | 39643 | 236010 | 610572 | 720015 | 1151135 | 1260383 | 1791006 | 2295501 | 2631907 |
| | C4 | 586820 | 175700 | 254039 | 1287942 | 1283153 | 1502586 | 1670054 | 2356962 | 2432769 | 3392554 |
| | C5 | 564627 | 1288 | 647 | 1531861 | 1355232 | 1304201 | 1413665 | 1500109 | 1960429 | 1999834 |
| | | | | | | | | | | | |
| Pipe+ LID | NC | 0 | 0 | 0 | 180610 | 403742 | 735983 | 994636 | 1239575 | 1571403 | 1833913 |
| | C1 | 435235 | 31853 | 205783 | 594395 | 981183 | 1247661 | 1207291 | 1602282 | 2407278 | 2683353 |
| | C2 | 0 | 4374 | 27315 | 275503 | 432434 | 808381 | 1439073 | 2002787 | 2375242 | 2362011 |
| | C3 | 10832 | 13901 | 152559 | 463675 | 568173 | 960769 | 1056741 | 1531386 | 1993485 | 2295640 |
| | C4 | 442271 | 106856 | 165356 | 1082850 | 1077049 | 1280177 | 1437899 | 2042621 | 2123354 | 2966933 |
| | C5 | 423441 | 723 | 536 | 1300494 | 1145087 | 1094680 | 1193045 | 1277930 | 1703625 | 1738962 |

[Figure]

*Figure 6 Future changes in flood volumes (CTFV) relative to historical conditions under current drainage system (yellow) and two adaptation scenarios (i.e. Pipe in red and Pipe+LID in green) at various return periods.*

**Specific comments**

L 31: given the delay between submission and publishing I suggest removing "current" from the text. Same for line 81.

**Response**: *Done.*

L 32: I suggest removing "existing" in favor of "past", "recent," "literature" or similar

**Response**: *Done.*

L 40: "Based on the results" –> "Results indicates that"

**Response**: *Done.*

L45: This is an outcome of your research, hence I would not say it is "obvious" but rather something like "very likely" or "results clearly indicates::" or similar.

**Response**: *Thanks for the suggestion. We have revised it to "results clearly indicate"*

L 46: "greenhouse gas emissions"

**Response**: *Done.*

L 62: The sentence is not clear. Please specify units of the change and in relation to what (e.g., flood peak, precipitation intensity?)

**Response**: *Thanks for the suggestion. We have revised this sentence to "a 30% and 40% increase in the precipitation intensity is expected for the 10- and 100-year return period respectively ...."*

L 66-69 is again not clear. E.g., non-stationary changes reads awkward. Also, what do you mean by future hydroclimate?

**Response***: Thanks for the suggestion. We have revised this sentence to "Therefore, it is important to investigate the performance of drainage systems in a changing environment and assess the potential flood damage for better adaptations"*

L71-77: As the article has a strong focus on mitigation and adaptation I suggest adding some relevant references in those areas. See the work by (Alfieri et al., 2016; Arnbjerg-Nielsen et al., 2015; Moore et al., 2016; Poussin et al., 2012) among others. The few ones currently listed in the article are somehow hidden in the conclusions.

**Response**: *Thanks for the suggestion. We have expanded literature review and incorporated the suggested references in the revision.*

*References*

*Alfieri, L., Feyen, L. and Baldassarre, G. D.: Increasing flood risk under climate change: a pan-European assessment of the benefits of four adaptation strategies, Clim. Change, 136(3), 507–521, doi:10.1007/s10584-016-1641-1, 2016.*

*Arnbjerg-Nielsen, K., Leonardsen, L. and Madsen, H.: Evaluating adaptation options for urban flooding based on new high-end emission scenario regional climate model simulations, Clim. Res., 64(1), 73–84, doi:10.3354/cr01299, 2015.*

*Moore, T. L., Gulliver, J. S., Stack, L. and Simpson, M. H.: Stormwater management and climate change: vulnerability and capacity for adaptation in urban and suburban contexts, Clim. Change, 138(3–4), 491–504, doi:10.1007/s10584-016-1766-2, 2016.*

*Poussin, J. K., Bubeck, P., H. Aerts, J. C. J. and Ward, P. J.: Potential of semi-structural and non-structural adaptation strategies to reduce future flood risk: Case study for the Meuse, Nat. Hazards Earth Syst. Sci., 12(11), 3455–3471, doi:10.5194/nhess-12-3455-2012, 2012.*

L136-137: the sentence is currently hard to read. Please reformulate.

**Response**: *Done.*

L 140-144: The sentence is rather misleading, first because there is now a wealth of studies using ensembles of several GCMs, and second because "all five GCMs" sounds like if there were only five, while CMIP5 includes way more than that.

**Response**: *Thanks for the suggestion. In the revision, we have deleted the statement "Unlike most previous studies that only used data from one or two GCM in climate change impact studies on urban floods".*

L 151: Rainfall is a climatic data. Please clarify.

**Response**: *Done.*

L 176: there are –> we considered

**Response**: *Corrected.*

L 181-182: This sentence should be supported by data, graphs or a reference to publications showing the validation work against historical records.

**Response**: *Thanks for the suggestion. In the revision, we have updated Figure 5a (attached below) by adding a graph on the city land use condition (e.g., green spaces and traffic network) and records of historical flood locations obtained from local water authorities. It is shown that the simulated locations of overloaded pipelines are in good agreement with historical records of flood points.*

[Figure]

**Figure 5** *Spatial distribution of overloaded pipelines (red color) induced by the 3-yr (left column) and 50-yr extreme events (right column) without and with adaptations. The total percentage of overloaded manholes (POM) and ratio of flood volume (RFV) are summarized.*

L 186-191: This part is difficult to read and understand. Please clarify and add some detail on how the TFV – return period relationship was derived. Figure 2 currently doesn't help a lot as it is too general, with no units nor tick marks. For example, if it the grey area is meant to indicate those events that

contribute the most to the annual damage, then it should take at least 50% of the area under the curve in Figure 2, as its integral is proportional to the total flood risk.

**Response**: *Thanks for the suggestion. As responded to the third general comment, we have revised these sentences to make it more clear and concise. Figure 2 is also updated following the suggestions:*

[Figure]

*Figure 2 Illustration of flood volume and average total expected TFVs as a function of return period under stationary drainage system. The grey area denotes the average total expected TFVs per year considering all kinds of floods.*

L 191- 195: This statement indicates a strong assumption which is not justified at this stage and sounds like a speculation. Perhaps the authors want to introduce what is later on indicated by their findings, but I think at this point this is unjustified, unless the point is supported by stronger evidence and/or some references.

**Response**: *Thanks for the comment. In the revision, we have moved these statements into the discussion section with reference added.*

L204-205: What is the extent of the enhancement of pipeline diameters in the adapted scenario? I couldn't find it anywhere in the text.

**Response**: *Thanks for the comment. The number of pipelines of the current and adapted system is 323 and 488, with a total pipe length of 251.6km and 375.4 km, respectively. In the adapted scenarios, the mean pipeline diameter is about 1.73m, which has increased by 53% compared to that in current system. We have clarified this in the revision.*

L230-231: Is this 52% a simple average of the percent changes shown in Figure 3? Then I suggest to clarify, as it doesn't necessarily mean the overall projected change in flood risk.

**Response**: *Thanks for the comments. In the revision, we have added more details on the changes, rather than showing the overall average value.*

L 254: More correctly "10 magnitudes of rainfall events".

**Response**: *Corrected.*

L 263: 19% should be 49%.

**Response**: *Corrected.*

L 332-333: Not just uncertainties but modeling assumptions as well.

**Response**: *Thanks for the suggestion. We have added more discussions on the assumptions in the revision.*

L 328-329: That's true but perhaps out of the scope of this article, as anyways there is no real damage model to evaluate economic flood losses.

**Response**: *Thanks for the comment. Yes, flood damage is not addressed in this study. We have added discussions on this in the revision.*

L 358-363: Following the discussions above one should be careful in calling these numbers "flood risk". Please adapt according to the indications in the discussion points above.

**Response**: *Thanks for the suggestion. We have changed "flood damage" or "flood risk" to "flood volume" throughout the text in the revision.*

L 605-606: I suggest including the period "2020-2040" in the caption for better understanding the graph.

Table 1: Which are the units in the table? Please specify units and the storm duration related to the precipitation intensity values listed (key parameter to understand such values).

**Response**: *Thanks for the suggestion. We have added the period "2020-2040" in the caption. Table 1 shows the future changes of precipitation at various return periods. It is dimensionless. The changes are multiplied to the present rainfall time series to obtain climate change scenarios as inputs to our model (see response to general comment 2).*

Figure 5: Please choose a more visible way of indicating overloaded pipelines, perhaps with a thicker line and/or a different color. Also the POM is currently mistakenly written as "NOM" in the 6 panels.

**Response**: *Thanks for the pointing out the typo. We have replaced "NOM" with POM. The illustration of overloaded pipelines is a direct output from the SWMM model. At present, it is not easy to highlight the pipelines given the hard-coded model user interface. Instead, we tried to update the figure with larger color contrast for better illustration. In addition, we have added city land use information (i.e., green spaces and traffic network) and records of historical flood pints obtained from the local water authorities in the updated figure.*

[Figure]

**Figure 5** *Spatial distribution of overloaded pipelines (red colour) induced by the 3-yr (left column) and 50-yr extreme events (right column) without and with adaptations. The total percentage of overloaded manholes (POM) and ratio of flood volume (RFV) are summarized.*

Figure 6: Add units in the axis labels. E.g.: "[-]" for dimensionless. Also, note the typo in the x-axis label.

**Response**: *Thanks for the suggestion. We have updated the figure in the revision.*

Figure 7: Negative values for risk reduction means increasing risk. Please reverse graphs with positive values (plus fix the typo rish -> risk)

**Response**: *Thanks for the suggestion. We have updated the figure and corrected the typo in the revision.*

---

## Author Comment (AC2) · 27 Mar 2017

**Impacts of future climate change on urban flood risks: benefits of climate mitigation and adaptations [MS No.: hess-2016-369]**

**Responses to review comments**

**Anonymous Referee #2:**

**SHORT COMMENTS IN THE JOURNAL STYLE Scientific questions:**
Adaptation effects on drainage performance in a context of climate change (CC) is relevant. Novel concepts. Try to quantify the impact adaptation measures is potentially new if appropriately developed in single case studies. Substantial conclusions. Not attended yet, due to insufficiently explained datasets and methods. Scientific methods and assumptions. Not clearly outlined. Results vs interpretations / conclusions. Unattended. Description. Pretty obscure. Authors proper credit. Ok! but not all is new. Title. OK! but to be revised in case of revision. Summary. Unbalanced on Climate trends when the most interesting part is adaptation. Overall presentation. Lacking of context outline. Language. To be revised by a mother tongue, that I am not. Formulae. Not expert enough to say. Parts to modify. Develop 1, 4 & 5, Clarify 2a & 2e, Reduce 2b, Delete 3b, Modify Fig. 1 & 5. References. Ok.

**Response***: We greatly appreciate the reviewer for the constructive comments and suggestions to improve our manuscript. In the revision, we have 1) added more details on the datasets and methods, 2) added more discussions on the assumptions and limitations, 3) modified the relevant statements and figures which are unclear or inaccurate, 4) revised the specific sections as suggested, 5) invited a native speaker to proof-read the paper. More details of our responses to each comment are provided as follows.*

**EXTENDED COMMENTS**
1 Introduction All key definitions should be provided here. Flood risk is the probability an hazard has to generate damages (UNISDR, ISO etc: : :), not a probability of a disastrous flood only (that is hazard occurrence). Should be wise to specify to whom this work is addressed, since very essential information of the case study is missing (see next sections).

**Response***: Thanks for the comments. We agree with the reviewer that flood risk refers to the probability of a hazard to cause damage. In this study, we investigate the potential changes in flood volume (TFV) under various scenarios of climate changes and explore the role of adaptations in regulating such changes. We acknowledge that the TFV is a hazard indicator, while flood damage is tightly linked to socio-economic conditions which is not addressed in this study. We have clarified this concept in the revision.*

*This study investigates the performance of drainage system under climate change scenarios, which has great implications for adaptation and mitigation strategies in a typical city in Northern China. Comparing the reduction of flood volume by climate mitigations (via reduction of greenhouse gas emissions) and local adaptations (via improvement of the drainage system) indicates that local adaptations are more effective than climate mitigations in reducing future flood volumes. This has broad implications for the research community on drainage system design and modeling in a changing environment. We emphasize the importance of considering adaptions in assessing climate change impacts on future urban floods. In the revision, we have provided more detailed information on the case study region, research background and the implications following the suggestions.*

2. Material and Methods 2a) (i) A characterization of the hazard (rainfall) in Hohhot City is missing. **Response***: Thanks for the comments. In the study region, most of rain storms fall between June and August, which accounts for more than 65% of annual precipitations. In the revision, we have provided more descriptions on the rainfall characterizations and flood hazards in the study region.*

*It should be noted that the input rainfall time series for the model are not the original historical observations. Rather, it is based on the storm intensity formula (SIF), which is used to estimate the design rainfall for each return period. The modeling practice mainly follows the standard procedure in urban drainage modeling in China, as documented in the national code for design of outdoor wastewater engineering (MOHURD, 2011). Specifically, the SIF represents an Intensity-Duration-Frequency (IDF) relationship, and is commonly used in the literature to estimate design rainfall hydrographs (Berggren et al., 2014; Cheng and AghaKouchak, 2014; Panthou et al., 2014; Willems, 2000; Zhou et al., 2012). Subsequently, the Chicago Design Storms (CDS) approach is applied to derive the design storms from the local SIF for the SWMM model as used in this study. The detailed procedures in using SIF to obtain the CDS design storms can be found in Chinese National Technical Guidelines for Establishment of Intensity-Duration-Frequency Curve and Design Rainstorm Profile (MOHURD, 2014) and have been well adopted in a number of Chinese urban drainage designs (Wu et al., 2016;Yin et al., 2016; Zhang et al., 2008; Zhang et al., 2015).*

*For the case study, the local rainfall is characterized by the SIF ($q=635*(1+0.841*lg(P))/t^{0.61}$), which is obtained from local weather bureau. 10 return periods are considered in the paper and a 4-hour rainfall time series is generated for each return period at a 10-minute interval. The technical details in using SIF to derive the CDS rainfall are given in the following. As shown in the Equation 1, the q is the average rainfall intensity, t is the storm duration and P is the design return period. The typical temporal*

*resolution considered in SIF for urban drainage simulations is minutes. A, b, c and D are regional parameters governing the IDF relations among rainfall intensity, return period and storm duration. For a given return period, the SIF is fitted into the Horner's equation as Eq.2:*

$$q = \frac{A(1 + Dlg(P))}{(t + b)^c} \qquad\qquad Eq.\ (1)$$

$$i = \frac{a}{(t + b)^c} \qquad\qquad Eq.\ (2)$$

*The synthetic hyetograph based on the Chicago method is computed using Eq. 2 and an additional parameter r (where 0< r <1) which determines the relative location of peak intensity (with respect to time), $t_p=r*t$. The time distribution of rainfall intensity is described after the peak $t_a = (1-r)*t$ and before the peak $t_b=r*t$ by Eq. (3) and (4). $i_b$ is the instantaneous rainfall intensity before the peak , $i_a$ is the instantaneous rainfall intensity after the peak.*

$$i_a = \frac{a[\frac{(1-c)t_a}{(1-r)} + b]}{(\frac{t_a}{(1-r)} + b)^{1+c}} \qquad\qquad Eq.\ (3)$$

$$i_b = \frac{a[\frac{(1-c)t_b}{r} + b]}{(\frac{t_b}{r} + b)^{1+c}} \qquad\qquad Eq.\ (4)$$

*By following the above procedure, a 4-hour rainfall time series can be generated for each return period with the peak located in the center of the period. In the revision, we have added more details about the rainfall and methods.*

***Reference:***

*Berggren, K., Packman, J., Ashley, R., and Viklander, M.: Climate changed rainfalls for urban drainage capacity assessment, Urban Water Journal, 11, 543-556, 10.1080/1573062X.2013.851709, 2014.*

*Cheng, L. Y., and AghaKouchak, A.: Nonstationary Precipitation Intensity-Duration-Frequency Curves for Infrastructure Design in a Changing Climate, Scientic Report, 4, 10.1038/srep07093, 2014.*

*MOHURD: AQSIQ. Code for Design of Outdoor Wastewater Engineering (GB 50014-2006), Ministry of Housing and Urban-Rural Development, General Administration of Quality Supervision, Inspection and Quarantine of the People's Republic of China: Beijing, China (In Chinese), 2011.*

*MOHURD: Technical Guidelines for Establishment of Intensity-Duration-Frequency Curve and Design Rainstorm Profile (In Chinese), Ministry of Housing and Urban-Rural Development of the People's*

*Republic of China and China Meteorological Administration. Accessed on November 2016 from http://www.mohurd.gov.cn/wjfb/201405/W020140519104225.pdf, 2014.*

*Panthou, G., Vischel, T., Lebel, T., Quantin, G., and Molinie, G.: Characterising the space-time structure of rainfall in the Sahel with a view to estimating IDAF curves, Hydrology and Earth System Sciences, 18, 5093-5107, 10.5194/hess-18-5093-2014, 2014.*

*Willems, P.: Compound intensity/duration/frequency-relationships of extreme precipitation for two seasons and two storm types, Journal of Hydrology, 233, 189-205, http://dx.doi.org/10.1016/S0022-1694(00)00233-X, 2000.*

*Wu, H., Huang, G., Meng, Q., Zhang, M., and Li, L.: Deep Tunnel for Regulating Combined Sewer Overflow Pollution and Flood Disaster: A Case Study in Guangzhou City, China, Water, 8, 329, 2016.*

*Yin, J., Yu, D. P., Yin, Z., Liu, M., and He, Q.: Evaluating the impact and risk of pluvial flash flood on intra-urban road network: A case study in the city center of Shanghai, China, Journal of Hydrology, 537, 138-145, 10.1016/j.jhydrol.2016.03.037, 2016.*

*Zhang, Y.-q., Lv, M., and Wang, Q.-g.: Formula method design of drainage pipe network and analysis of model simulation, Water Resource Power, 33, 105-107, 2015.*

*Zhang, D., Zhao, D. q., Chen, J. n., and Wang, H. z.: Application of Chicago approach in urban drainage network modeling, Water & Wastewater Engineering, 34, 354-357, 2008.*

*Zhou, Q., Mikkelsen, P. S., Halsnaes, K., and Arnbjerg-Nielsen, K.: Framework for economic pluvial flood risk assessment considering climate change effects and adaptation benefits, Journal of Hydrology, 414, 539-549, 10.1016/j.jhydrol.2011.11.031, 2012.*

(ii) A detailed description of watershed soils is recommended. Rocky, lateritic, clay, sandy, or: : : soils perform differently in semi-arid contexts than in wet contexts. Even where infiltration seems possible some pervious looking soils after the first minutes turn into impervious. Context matter in this type of study.

**Response***: Thanks for the comments. We agree with the reviewer that soil conditions matter in this type of study. In this study, three general soil categories are considered, i.e., the sand, loam and clay. According to the limited data on local soil conditions from local water authorities, the major soil type of the study region is a mixture of loam and clay. Based on the Horton's infiltration method (Rossman and Huber, 2016) and the values suggested by (Akan, 1993) as shown in the table below, we used the values under the category of "Dry loam soils with little or no vegetation" to represent the maximum infiltration capacity in the model. We have added more descriptions on this in the revision.*

| Maximum (Initial) Infiltration Capacity (Akan, 1993) | | |
|---|---|---|
| **Soil Type** | *(in/hr)* | *(mm/hr)* |

| | | |
|---|---|---|
| Dry sandy soils with little or no vegetation | 5.0 | 127 |
| Dry loam soils with little or no vegetation | 3.0 | 76.2 |
| Dry clay soils with little or no vegetation | 1.0 | 25.4 |
| Dry sandy soils with dense vegetation | 10.0 | 254 |
| Dry loam soils with dense vegetation | 6.0 | 152 |
| Dry clay soils with dense vegetation | 2.0 | 51 |
| Moist sandy soils with little or no vegetation | 1.7 | 43 |
| Moist loam soils with little or no vegetation | 1.0 | 25 |
| Moist clay soils with little or no vegetation | 0.3 | 7.6 |
| Moist sandy soils with dense vegetation | 3.3 | 84 |
| Moist loam soils with dense vegetation | 2.0 | 5.1 |
| Moist clay soils with dense or no vegetation | 0.7 | 18 |

*To further address the concern, we have conducted a set of sensitivity experiments in the revision. Specially, we used three possible infiltration values corresponding to the first three soil types (i.e., dry sand, loam and clay soils with little or no vegetation) as listed in the above table. The parameters associated with each possible infiltration value are shown in the table below:*

| **Infiltration parameters*** | MaxRate | MinRate | Decay rate | DryTime |
|---|---|---|---|---|
| | [in/hr] | [in/hr] | [1/hr] | [days] |
| *Dry loam with little or no vegetation* | 3 | 0.5 | 4 | 7 |
| *Dry sand with little or no vegetation* | 5 | 0.7 | 5 | 5 |
| *Dry clay with little or no vegetation* | 1 | 0.3 | 3 | 9 |
| *\* To describe Horton infiltration method in SWMM, four basic infiltration parameters are required (Rossman and Huber, 2016) :*
• *MaxRate: Maximum infiltration rate on Horton curve*
• *MinRate: Minimum infiltration rate on Horton curve*
• *Decay: Decay rate constant of Horton curve*
• *DryTime: Time it takes for fully saturated soil to dry* | | | | |

*The original Figure 7 shows the comparison of benefits of climate mitigation and two adaptation strategies in reducing flood volume, based on the soil category 'Dry loam with little or no vegetation'. Here, we re-plotted the figure to show the uncertainty range arising from the representation of soil conditions in the drainage model, using the possible infiltration values of the aforementioned three categories. It is shown that magnitudes of estimated benefits differ to some degree, but the performance of adaptation measures is better than that of climate mitigation, which is supportive of the major conclusion of this study. We have included the relevant descriptions and results in the revised manuscript.*

[Figure]

***Original Figure 7*** *Comparison of benefits of climate mitigation and two adaptation strategies in reducing urban floods with changes in precipitation intensities at various return periods.*

[Figure]

***Revised Figure 7_with uncertainty***: *Estimated benefits of climate mitigation and two adaptation strategies in reducing urban floods risks with changes in precipitation intensities at various return periods. Infiltration values corresponding to three soil categories are used and the boxplot is used to show the uncertainties arising from soil conditions.*

***References:***

*Akan, O. A.: Urban stormwater hydrology: a guide to engineering calculations, CRC Press, 1993.*

*Rossman, L., and Huber, W.: Storm Water Management Model Reference Manual EPA/600/R-15/162A, 2016.*

(iii) Authors consider permeable pavements, infiltration trenches and green roofs as possible adaptation measures. Which are the permeable soil and coverage rates in the different parts of the watershed considered?

**Response***: Thanks for the comments. We agree that explicit consideration of permeable pavements, infiltration trenches and green roofs would make the designed adaptation measures more specific and realistic (Elliott and Trowsdale, 2007; Zoppou, 2001). However, there is no such detailed information on the permeable soil and coverage rates in the study region, which prevents us from representing these individual features/parameters in the model. Instead, the second adaptation scenario is designed to investigate the effects of increased permeable surfaces on flood volume, and is reflective of the combined effects of infiltration-related measures, including permeable pavements, infiltration trenches and green roofs. That is, a simplified approach by altering the subcatchment imperviousness was adopted due to the limitation of data availability in the study region. Specifically, we derived such information by comparing the current and planned landuse maps and incorporated the changes in landuse and imperviousness (see the updated Figure 1d) in the adaptation scenario. The figure 1d shows the difference in weighted mean imperviousness (WMI) calculated for each subcatchments (different parts of the watersheds) in the current and planned maps, which is used to indicate the areas that are more suitable for adaptation measures based on the city plan. For example, a subcatchment with a higher positive change in the WMI indicates that the area is planned to have a landuse type with lower imperviousness and therefore is assumed to be more suitable for lid planning, and vice versa. We have clarified this with more discussions in the revision.*

[Figure]

(iv) Can the authors provide some information about last disastrous floods in the case study? Areas affected the most, etc.

**Response**: *Thanks for the comment. On 11th July 2016, the city, especially the western part of the watershed, was hit by an extreme rainfall event with more than 100 mm of rain within 3 hours. The local meteorological department issued the red warning of rainstorm. Floods were reported in multiple sites in the central area and caused severe traffic jams in major streets (see the photos below). Some residential buildings were also flooded. The rain forced cancellations of at least 8 flights and 17trains, and delays of dozens of transportation systems. We have added more descriptions on this in the introduction of the study region in the revision.*

[Figure]

(Photos: yjhlnews.com(left) and chinanews.com(right))

*To provide more background information on historical flood events in the study area, we have included a map describing historical flood records and city traffic network in Figure 5a. It is obvious that the central portion of the city is the most affected region due to the low service level of its drainage system. We have updated the Figure 5 and added more descriptions in the revised manuscript.*

[Figure]

***Figure 5****: Spatial distribution of overloaded pipelines (red colour) induced by the current 3-yr (left column) and 50-yr extreme events (right column) without and with adaptations. The total percentage of overloaded*

*manholes (POM) and ratio of flood volume (RFV) are summarized. A summary of historical flood points are given in (a).*

2b) (i) It's quite normal to use more than one GCMs .

**Response***: Agreed. We have deleted relevant statements in the revision.*

(ii) Reader expects to learn from the expected changes in rainfall (mm and in which month) but no information is provided on this topic.

**Response***: Thanks for the comments. Readers can refer to the table 1 which summarizes the changes in extreme precipitation intensity of various return periods. In the revision, we have added additional statistics on the monthly and annual precipitation changes.*

2c) (i) Which rainfall information has been used to run SWMM [dataset length (years) and type (daily, three hourly, hourly, etc.)]?

**Response***: Thanks for the comments. The rainfalls as inputs for the model are based on artificial rains in the format of Chicago Design storms derived from historical rainfall records following the standard by the local weather bureau and the national code for design of outdoor wastewater engineering (please see response to comment 2a for details). The rainfall period is 4 hours at sub-hour (i.e., 10 minute) time step.*

2e) (i) The adaptation measures considered are to reduce the amount of water that run off. This is one side of the problem. The other one is to slow down the water speed. And for this no measure is considered: there is a wide range of measures for semi-arid contexts commonly used for this. I recommend to consider it or explain why you don't.

**Response***: Thanks for the comments. We agree that slowing down the water speed could be an alternative adaptation approach for attenuating runoff peak and reducing flood volume (Messner et al., 2006; Floodsite, 2009). We note that the water speed is influenced by, among others, the gradient and flow resistance of the bed of the water course (Ashley et al., 2007) and such information is not available at the sub-catchment scale in the study region.*

*There are two main reasons that we did not consider the measures by attenuating the water speed in our designed adaptation approach. First of fall, although some of the LID measures are primarily designed to slow down the flow speed, i.e., vegetated swales, most of the LID measures can reduce both runoff volume and flow speed at the same time. Constrained by the one-dimensional SWMM modelling approach in this study, the performances of LID measures are mainly evaluated according to their effects on reducing water volume from overloaded manholes (Oraei Zare et al., 2012;Lee et al., 2013). And it is difficult to examine*

*whether a reduced flood event for each manhole is induced by the runoff volume or inherent speed control function in the model. Second, there is a lack of data for us to consider and validate this specific measure in the case study. Especially, the required information on surface roughness, soil conductivity, seepage rate are unavailable at the subcatchment scale in the study region. Based on the available datasets on current and future landuse maps, this study tends to apply and assess adaptation measures that mainly affect the surface imperviousness. We have added the discussions in the revised manuscript.*

*Reference:*

*Ashley, R., Garvin, S., Pasche, E., Vassilopoulos, A., and Zevenbergen, C.: Advances in Urban Flood Management, in, edited by: Ashley, R., Garvin, S., Pasche, E., Vassilopoulos, A., and Zevenbergen, C., Taylor & Francis/Balkema, London, UK, 2007.*

*Floodsite: Flood risk assessment and flood risk management. An introduction and guidance based on experiences and findings of FLOODsite (an EU-funded Integrated Project), Deltares/Delft Hydraulics. ISBN 978 90 8 |4067|0, 2009.*

*Lee, J. M., Hyun, K. H., and Choi, J. S.: Analysis of the impact of low impact development on runoff from a new district in Korea, Water Science and Technology, 68, 1315-1321, 10.2166/wst.2013.346, 2013.*

*Messner, F., Penning-Rowsell, E., Green, C., Meyer, V., Tunstall, S., and Van der Veen, A.: Guidelines for Socio-economic Flood Damage Evaluation, Report Nr. T9-06-01, in, FLOOD site, HR Wallingford, UK, 2006.*

*Oraei Zare, S., Saghafian, B., Shamsai, A., and Nazif, S.: Multi-objective optimization using evolutionary algorithms for qualitative and quantitative control of urban runoff, Hydrol. Earth Syst. Sci. Discuss., 9, 777-817, 10.5194/hessd-9-777-2012, 2012.*

(ii) How Authors have determined the impact of individual adaptation measures (permeable pavements, trenches, green roof) over run off reduction? This should be explained.

**Response**: *Thanks for the comments. As clarified in the response to 2a (iii), the second adaptation measure is mainly designed to investigate the impacts of increased permeable surfaces on flood volume reductions by altering the imperviousness of subcatchments to represent the infiltrated and detained water volume in the runoff-generation process. That is, the individual measures related to permeable pavements, trenches, green roof are not considered separately but represented in a combined and simplified approach. Thus, we are not able to explicitly assess the performance of these individual measures on flood reductions in details. We have added discussions on this in the revision.*

3) Results 3b) (i) I don't understand the approach: Mitigation is expected to impact on CC at long term (decades: : :). Drainage system is expected to reduce CC impacts at short-medium term (1-5 years). Is obvious that adapting we can't expect to see effects on rainfall: : :

**Response**: *Thanks for the comments. Mitigation refers to climate mitigations via reduction of greenhouse gas emissions. The mitigation effects are assessed here by comparing the results based on RCP8.5 emission scenario (which is a business-as-usual scenario) and RCP2.6 scenario (which considers the reduction of greenhouse gas emission). Climate mitigation via reducing greenhouse gas emissions is expected to influence precipitation characteristics and thus the subsequent flood hazards (i.e., flood volume in this study). Adaptation measures are localized and here refer to the specific design/update of drainage system. The possible land surface-atmosphere interactions which would indirectly affect the rainfall and floods are not considered in this study. We have clarified this in the revised manuscript.*

4) Uncertainties & Limitations (i) The consideration of the state of drainage system could be a limitation of this study? A drainage system obstructed by vegetation, waste or artefacts (cables, pipes, temporary constructions) can make the outcomes of the SWMM quite distant from the real world. And change also recommendations: : : that need to be extended to waste sector.

**Response**: *Thanks for the comments. We agree with the reviewer that the state of drainage system could affect its conveyance capacity and thus the system performance to various degrees. In some cases, floods are not induced by the exceedance of drainage capacity, but by the deterioration of drainage system itself, e.g., aging network, pipe deterioration, blockage, construction failures and local external factors (Dawson et al., 2008; CIRIA, 1997; Davies et al., 2001). Previous studies with a focus on sewer inspection and condition assessment, maintenance and rehabilitation strategies, have highlighted the need for labor-intensive field investigations for collecting information on the waste status and relocations, and such studies are often limited to certain areas (Ana and Bauwens, 2007; Fenner, 2000). In fact, assessment of drainage conditions requires detailed datasets, which has been recognized as a great challenge in applications. For example, in Europe, water service data collections mainly cover pipe length, age, material, diameter and location (Stone et al., 2002; Ana and Bauwens, 2007), while the assessment of pipe conditions are often managed by separate and specialized programs.*

*It is beyond the scope of this study to take into account the actual state of the pipe system due to difficulties in collecting field data and selecting and utilizing appropriate methods for reasonable assessment of the current pipe conditions. Such studies usually require comprehensive efforts on the material, data and method, (e.g., Dawson et al. 2008; Chae and Abraham 2001; Chughtai and Zayed 2008), which is not the focus of this paper. We acknowledge that the hydraulic performance may be overestimated without considering the drainage conditions and the waste section in the SWMM modeling approach (Pollert et al., 2005). In the revision, we have added more discussions on the impacts of pipe conditions on system performance, which should be addressed in the future study.*

*References:*

*Ana, E., and Bauwens, W.: Sewer network asset management decision-support tools: a review, International Symposium on New Directions in Urban Water Management, 2007, 1-8, 12-14 September 2007, UNESCO Paris*

*Chae, M., and Abraham, D.: Neuro-fuzzy approaches for sanitary sewer pipeline condition assessment, J. Comput. Civ. Eng., 15, 4, 2001.*

*Chughtai, F., and Zayed, T.: Infrastructure Condition Prediction Models for Sustainable Sewer Pipelines, Journal of Performance of Constructed Facilities, 22, doi:10.1061/(ASCE)0887-3828(2008)22:5(333), 2008.*

*CIRIA: Risk Management for Real Time Control in Urban Drainage Systems: Scoping Study, Project Report 45. CIRIA, London., 1997.*

*Davies, J. P., Clarke, B. A., Whiter, J. T., and Cunningham, R. J.: Factors influencing the structural deterioration and collapse of rigid sewer pipes, Urban Water, 3, 73-89, http://dx.doi.org/10.1016/S1462-0758(01)00017-6, 2001.*

*Dawson, R. J., Speight, L., Hall, J. W., Djordjevic, S., Savic, D., and Leandro, J.: Attribution of flood risk in urban areas, Journal of Hydroinformatics, 10, 275-288, 2008.*

*Fenner, R. A.: Approaches to sewer maintenance: a review, Urban Water, 2, 343-356, http://dx.doi.org/10.1016/S1462-0758(00)00065-0, 2000.*

*Pollert, J., Ugarelli, R., Saegrov, S., Schilling, W., and Di Federico, V.: The hydraulic capacity of deteriorating sewer systems, Water Science and Technology, 52, 207-214, 2005.*

*Stone, S., Dzuray, E. J., Meisegeier, D., Dahlborg, A. S., and Erickson, M.: Decision-Support Tools for Predicting the Performance of Water Distribution and Wastewater Collection Systems., US EPA. 97pp., 2002.*

5. Could the Authors consider to show us what is their way forward?

**Response***: Thanks for the comments. As demonstrated in this study, adaptation is found to be more effective in reducing future flood volumes than climate mitigations. However, several simplified approaches were adopted in the modeling and assessments as commented by the reviewer. Depending on the progress on data collection and needs of local authorities, we plan to conduct a more comprehensive analysis of the adaptation measures in a more localized area by applying more advanced methods for pipe assessment (e.g., considering the changing pipe condition), LID measures (detailed modeling of LID control), two-dimensional surface flooding for the assessment of flood damage and risk, and those interesting points as raised by the reviewer. We have added discussions on this in the revision.*

Figures 1 & 5: scale is not showed: how large are blocs contoured by drainage network?

**Response***: Thanks for the comments. In the revision, relevant figures have been updated by including a scale bar (see the attached figures below).*

[Figure]

**Figure 1** *Land use of the study region for the year 2010 (a) and 2020 (b). Pipe network description of current and planned drainage systems (c). Difference in Weighted Mean Imperviousness (WMI) between year 2010 and 2020 (d).*

[Figure]

**Figure 5** *Spatial distribution of overloaded pipelines (red colour) induced by the current 3-yr (left column) and 50-yr extreme events (right column) without and with adaptations. The total percentage of overloaded manholes (POM) and ratio of flood volume (RFV) are summarized.*

6. Manuscript's title Show the name of the case study and the country. Limit to Adaptation, delete mitigation, delete risk.

**Response**: *Thanks for the suggestion. In the revision, we deleted the "risk' and added the study region name and country. But we tend to keep "mitigation" in the title as we believe it is important although we emphasize the importance of considering adaptions in assessing climate change impacts on future urban floods. This is because the role of adaptations in reducing flood volume is highlighted through comparing with the reduced floods by climate mitigation. Indeed, comparing the reduction of flood volume by climate mitigations (via reduction of greenhouse gas emissions) and local adaptations (via improvement of the drainage system) indicates that local adaptations are more effective than climate mitigations in reducing future flood volumes.*

---

## Author Response (AR1)

**Impacts of future climate change on urban flood risks: benefits of climate mitigation and adaptations [MS No.: hess-2016-369]**

**Responses to reviewer comments**

**REFEREE REPORT(S):**

**Anonymous Referee #1:**

The article by Zhou at al tackles a very topical issue in the field of flood risk assessment, which deals with climate change, mitigation and adaptation measures. The research questions that the authors investigate is sound and meaningful, and it is particularly interesting as the benefits of adaptation and mitigation measures are evaluated numerically through a modelling framework (though their associate cost is not assessed). Now the bad news: the structure of the article is sometimes not so clear, due to missing links, lack of details in the methods, questionable assumptions and unclear interpretation of results. Also, the use of English, although sufficient, is sometimes sub-optimal, and could do with a revision by a native speaker. Please pay careful attention to the use of prepositions and of the "s" for plurals. I found a number of mistakes and inappropriate use. Nonetheless, I think that the article had good potential for being published, provided that the following comments are adequately addressed. Please pay special attention to the general comments, where substantial work is needed to improve parts of the description of methods, assumptions and evaluation of results.

**Response**: *We greatly appreciate the reviewer for the constructive comments and suggestions to improve our manuscript. In the revision, we have 1) added more details on the datasets and methods, 2) added more discussions on the assumptions and limitations, 3) modified the relevant statements and figures which are unclear or inaccurate, 4) invited a native speaker to proof-read the paper. More details of our responses to each comment are provided as follows.*

*Note: the line numbers as mentioned in the response below refer to those in the cleaned version of manuscript.*

**General comments**

L 131-146: I would like to see some comments by the authors on the suitability of CMIP5 data for studies
on urban flooding. Given the coarse resolution of CMIP5 (as they are global models), I'm sure that the
entire study region is considerably smaller than 1 model grid cell. This poses some questions on how well
extreme precipitation for modeling urban flooding is adequately represented by such datasets, given that
such models are not able to represent local and short-lived storms commonly inducing flooding in small
catchments. Intuitively one would say that downscaled projections with high resolution would be more
suitable for this work, though that clearly depends on the data availability. Perhaps the authors can
comment on that.

**Response**: *Thanks for the comments. As pointed out by the reviewer, bias would exist in global climate*
*model (GCM) simulations especially at the local and regional scales. An alternative approach is to*
*simulate the future climate using regional climate model (RCM) nested within a GCM. Such climate*
*projections by RCM have added value in terms of higher spatial resolution which can provide more*
*detailed regional information. However, various level of bias would still remain in RCM simulations*
*(Teutschbein and Seibert 2012) and bias correction of RCM projections are required, e.g. the European*
*project ENSEMBLES (Hewitt and Griggs 2004; Christensen et al. 2008). To run regional climate model*
*is not within the scope of this study. Instead, we tend to use publicly available climate projection dataset.*
*Here, we obtain climate projections from the ISI-MIP (Warszawski et al. 2014), which provides spatially-*
*downscaled climate data for impact models. The climate projections were also bias-corrected against*
*observations (Hempel et al. 2013) and have been widely used in climate change impact studies on*
*hydrological extremes such as floods and droughts (e.g. Dankers et al. 2014; Prudhomme et al. 2014;*
*Giuntoli et al. 2015; Leng et al. 2015).*

*It should be noted that we used the delta change factor to derive the climate scenarios as inputs into our*
*flood drainage model instead of using the climate projections directly. Specifically, we calculate the*
*change factor between current and future climate projection simulated by GCMs and multiply them to*
*observed time series to derive future climate scenario into our flood drainage model. This is because the*
*relative climate change signal simulated by GCMs are argued to be more reliable than the simulated*
*absolute values (Ho et al. 2012). What's more, we use an ensemble of GCM simulations rather than one*
*single climate model in order to characterize the uncertainty range arising from climate projections.*

*In the revision, we have added more discussions on this (Lines 388-420).*

*Reference*

*Warszawski, L., Frieler, K., Huber, V., Piontek, F., Serdeczny, O., & Schewe, J. (2014). The inter-sectoral*

*impact model intercomparison project (ISI–MIP): project framework. Proceedings of the National*

*Academy of Sciences, 111(9), 3228-3232.*

*Dankers, R., Arnell, N. W., Clark, D. B., Falloon, P. D., Fekete, B. M., Gosling, S. N., ... & Stacke, T.*

*(2014). First look at changes in flood hazard in the Inter-Sectoral Impact Model Intercomparison Project*

*ensemble. Proceedings of the National Academy of Sciences, 111(9), 3257-3261.*

*Prudhomme, C., Giuntoli, I., Robinson, E. L., Clark, D. B., Arnell, N. W., Dankers, R., ... & Hagemann, S.*

*(2014). Hydrological droughts in the 21st century, hotspots and uncertainties from a global multimodel*

*ensemble experiment. Proceedings of the National Academy of Sciences, 111(9), 3262-3267.*

*Leng, G., Tang, Q., & Rayburg, S. (2015). Climate change impacts on meteorological, agricultural and*

*hydrological droughts in China. Global and Planetary Change, 126, 23-34.*

*Giuntoli, I., Vidal, J. P., Prudhomme, C., Hannah, D. M. (2015). Future hydrological extremes: the*

*uncertainty from multiple global climate and global hydrological models. Earth System Dynamics, 6(1),*

*267.*

*Teutschbein, C., & Seibert, J. (2012). Bias correction of regional climate model simulations for*

*hydrological climate-change impact studies: Review and evaluation of different methods. Journal of*

*Hydrology, 456, 12-29.*

*Hempel, S., Frieler, K., Warszawski, L., Schewe, J., & Piontek, F. (2013). A trend-preserving bias*

*correction–the ISI-MIP approach. Earth System Dynamics, 4(2), 219-236.*

*Christensen, J. H., Boberg, F., Christensen, O. B., & Lucas‑Picher, P. (2008). On the need for bias*

*correction of regional climate change projections of temperature and precipitation. Geophysical*

*Research Letters, 35(20).*

*Hewitt, C. D., and D. J. Griggs (2004), Ensembles-based predictions of climate changes and their*

*impacts, Eos Trans. AGU, 85, 566.*

*Ho, C. K., Stephenson, D. B., Collins, M., Ferro, C. A., & Brown, S. J. (2012). Calibration strategies: a*

*source of additional uncertainty in climate change projections. Bulletin of the American Meteorological*

*Society, 93(1), 21.*

L 169-182: I suggest expanding this section as I think there are some unclear points which prevents the reader from understanding some modeling steps, underlying assumptions, as well as from making the approach reproducible. For example, is q in eq. 1 the peak intensity? Which is the temporal resolution considered? Most climate datasets have 1 day as highest temporal resolution, but that would probably be rather coarse for urban flooding applications. How are then the hyetographs calculated from the q? Is it a simple rescaling based on their peak, keeping the same shape? Also, I see a lack of information on how climatic data is handled statistically to estimate storms/volumes with selected return period between 1 and

1000 years. For example, I see that the considered period for assessing future scenarios is 2020-2040, hence 21 years of data. Does it mean that return periods in the order of 1000 years are estimated from 21

years of data? Could the authors clarify on this? Can they provide ranges of uncertainty due to the undersampling of the climate variability in such long periods? Also, this should be mentioned in Sect. 4

as a further uncertainty source. Final comment is about eq. 1: could you briefly comment on how the parameters A, b, c, D are valid under a non-stationary climate? 4 parameters and just 2 variables sounds a lot for an empirical formula.

**Response**: *Thanks for the comments. In this study, we adopt the storm intensity formula (SIF) to derive*

*the precipitation input into our drainage model. Application of the SIF is a standard practice for*

*determining design rainfalls in urban drainage modelling in China, and is well documented in the*

*National Guidance for Design of Outdoor Wastewater Engineering (MOHURD, 2011). Specifically, the*

*SIF represents an Intensity-Duration-Frequency (IDF) relationship, which is a common approach in*

*literature for estimating design rainfall hydrographs using the Chicago Design Storms (CDS) approach*

*(Berggren et al., 2014; Cheng and AghaKouchak, 2014; Panthou et al., 2014; Willems, 2000; Zhou et al.,*

*2012). More details can refer to Smith (2004) for the derivation of CDS from an IDF relationship. In*

*China, the procedure for applying SIF to obtain CDS is outlined in the National Technical Guidelines for*

*Establishment of Intensity-Duration-Frequency Curve and Design Rainstorm Profile (MOHURD, 2014)*

*and have been well adopted for Chinese urban drainage designs (Wu et al., 2016; Yin et al., 2016; Zhang*

*et al., 2008; Zhang et al., 2015). Therefore, the method for using the SIF to generate CDS design storms*

*for our SWMM modelling study is reproducible and valid for drainage modelling.*

*The technical details of SIF and derivation of CDS rainfall are given as follows. As shown in the*

*Equation 1, the q is the average rainfall intensity, t is the storm duration and P is the design return*

*period. The typical temporal resolution in SIF is minutes for urban drainage modelling. A, b, c and D are*

*the regional parameters governing the IDF relations among rainfall intensity, return period and storm*

*duration. For a given return period, the SIF can be fitted into the Horner's equation (2004) as shown in*

*Equation 2:*

$$q = \frac{A(1 + Dlg(P))}{(t + b)^c} \qquad\qquad Eq.\ (1)$$

$$i = \frac{a}{(t + b)^c} \qquad\qquad Eq.\ (2)$$

*The synthetic hyetograph based on the Chicago method is computed using Equation 2 and an additional*

*parameter r (where 0< r <1) which determines the relative time step of the peak intensity, $t_p=r*t$. The*

*time distribution of rainfall intensity is described after the peak $t_a = (1-r)*t$ and before the peak $t_b=r*t$ by*

*Equation (3) and (4), respectively. Specially, $i_b$ is the instantaneous rainfall intensity before the peak, and*

*$i_a$ is the instantaneous rainfall intensity after the peak.*

$$i_a = \frac{a[\frac{(1-c)t_a}{(1-r)} + b]}{(\frac{t_a}{(1-r)} + b)^{1+c}} \qquad\qquad Eq.\ (3)$$

$$i_b = \frac{a[\frac{(1-c)t_b}{r} + b]}{(\frac{t_b}{r} + b)^{1+c}} \qquad\qquad Eq.\ (4)$$

*In this study, we considered 10 return periods, i.e., the 1-, 2-, 3-, 10-, 20-, 50-, 100-, 200-, 500-, and*

*1000-year events. A 4-hour rainfall time series was generated for each return period at 10-minute*

*intervals based on Equations 1−4. The A, b, c and D parameters governing the SIF shape were obtained*

*from the local weather bureau, which fits the historical precipitation distribution for the study region. In*

*the revision, we have added more details about the methods (Lines 189-220).*

*As for the generation of future climate scenarios, we first calculate the change factor for each return*

*period. Specifically, for each year, the annual maximum daily precipitation was determined for both*

*historical and future periods. Then, the generalized extreme value (GEV) distribution is fitted separately*

*to the two sets of daily values (Coles 2001; Katz et al. 2002). The goodness of fit was tested by*

*calculating the Kolmogorov–Smirnov and Anderson–Darling statistics. The value corresponding to each*

*return period is derived based on the GEV distribution and the changes between future and historical*

*periods are calculated as the change factors (as shown in Table 2 in the text). The change factor for each*

*return period is then multiplied to the historical design CDS rainfall time series to derive future climate*

*scenarios for the model. We acknowledge that to estimate the changes in extreme precipitation events*

*involves inevitable uncertainties especially for return periods beyond the length of the data, e.g. 1000yrs*

*as pointed by the reviewer.  Hence, caution should be exercised when interpreting the results for return*

*levels beyond the data length. However, we'd like to mention that "return period" is intrinsically a*

*statistical measurement derived based on probability density function (PDF) of historical data in*

*extended period. That is, it represents a recurrence interval which is an estimate of the likelihood of an*

*event (in our case, a flood) indicated by the PDF. Depending on the historical period used, the return*

*period could vary if the time series is not stationary. Nevertheless, a 1000-year return period can be*

*derived from 21-year time series based on its definition by using a PDF. We have added discussions on*

*this in the revision. We agree that climate variability range would be under-sampled, although five*

*climate models are used to show the possible ranges. In the revision, we have added discussions on this in*

*the revision (Lines 222-233; 416-420).*

*The parameters A, b, c, D are derived from sub-hourly rainfall data and provided by local weather*

*bureau. The four parameters which describe the Intensity-Duration-Frequency (IDF) relationship in the*

*study region are assumed to be constant without considering its non-stationary features in a changing*

*climate. To derive the parameter in the future period requires hourly precipitation data, which are not*

*readily available. Hence, the IDF relationship is assumed to remain stable in the future and only changes*

*in the daily mean intensity are considered. Given the above limitations, we acknowledge that our*

*modeling results mainly represent the first-order potential climate change impacts on urban floods.*

*Future efforts should be devoted to the representation of dynamic rainfall changes at hourly time step*

*taking into account of non-stationary climate change. We have added more discussions on this in the*

*revised manuscript (Lines 414-420).*

*forth).*

[Figure]

***Figure 4*** *Comparison of (a) flood volume, (b) total TFVs (i.e., the piece-wise integral of flood volume*
*versus the expected frequency with changes in precipitation intensity of various return periods under*
*RCP8.5 (blue) and RCP2.6 (red). (c) is for the reduced TFVs in percentage (i.e., benefits of climate*
*mitigation) in RCP2.6 relative to RCP8.5 at various return periods.*

L 265-286: I find this part rather difficult to understand and suggest the authors to clarify some points and describe more thoroughly Figure 6 and its usefulness. First, the way changes (CTFV) are defined is not intuitive, as it is now defined as a multiplicative factor. Changes should be *CTFV=(TFVc-TFVnc)/*

*TFVnc*. Also, why the current system is less sensitive to climate change than the adapted system (l 268-

269)? I'm a bit puzzled by seeing that small changes in the 10-year precipitation intensity lead up to a 7- fold increase in TFV under the case of adaptation. Does it mean 7 times worse conditions or simply that the adapted system can hold more water, also because the catchment area is larger? Then I get confused on the definition of TFV: is it the total volume or simply the excess volume after filling completely the pipes system? I thought it's the second option, but now I'm confused. Please clarify in sect. 2c. In both cases it's difficult to assess how worse the conditions (i.e., the damage) would be under larger TFV in the adapted system, though I think a graph with such information is currently missing and could be added.

Finally, please avoid 4 decimals in numbers at lines 270-271; 2 decimal digits are surely enough.

**Response**: *Thanks for the comments. We are sorry for the confusion. The TFV is defined as the total*

*volume flooded from manholes without taking into account the outlet discharges, i.e., excess water from*

*manholes after completely filling the pipe system. As pointed out by the reviewer, the current drainage*

*system is less sensitive to climate change. This is because the capacity of current drainage system is*

*small, i.e. the excess water after filling completely the pipe system (i.e., TFVnc) is large. Given extreme*

*rainfall events, the current system would be flooded completely, thus exhibiting less sensitivity to larger*

*extreme rainfall events in the future. Therefore, the magnitude of changes in excess flood volume is*

*smaller in the current system than the adapted system due to its large value of denominator in the*

*calculation of CTFV (CTFV=(TFVc-TFVnc)/TFVnc).*

*In order to better clarify this point, we have provided a table below summarizing the flood volumes of*

*current and adapted drainage systems, with and without climate change. It is evident that for the present*

*time, the flood volume of the adapted systems are much smaller than that in the current system due to*

*capacity upgrades in the adapted systems to hold more water. For example, when experiencing a 10-year*

*extreme rainfall event, the urban flood volumes for the present period (i.e., TFVnc) are 1041,230,*

*274,650 and 180,610 m3 in the current and two adapted systems(highlighted in blue), respectively, while*

*in the future period, the magnitude of flood volume (i.e., TFVc, highlighted in yellow) is relatively similar*

*among the three drainage systems. Therefore, future CTFVs relative to the historical period are much*

*larger in the adapted systems than in the current system. Mathematically, the low sensitivity of the*

*current drainage system to changes in extreme rainfall intensity could be attributed to the large value of*

*the denominator in the calculation of CTFV.*

*In the revision (Lines 175-177, 326-357), we have 1) clarified the definition of TFV; 2) re-defined*

$CTFV=(TFV_c-TFV_{nc})/TFV_{nc}$ *following the suggestion, and updated Figure 6 accordingly (see Figure 6*

*below); 3) added more discussions on projected changes on TFV; 4) used 2 decimal digits for the*

*numeric results throughout the text. Based on the suggested formula, the calculated CTFV for the three*

*systems are 0.41, 1.75 and 2.29, respectively. The larger CTFVs in the adapted systems does not mean the*

*worsened conditions. Rather, it indicates that the capacity (i.e., service level) of adapted system tends to*

*become lower with climate changes while the current system has already reached its peak capacity in the*

*present period and thus shows small sensitivity to climate change.*

**Table S1**: *TFVs of current and adapted systems with and without climate changes*

| Return period | | 1 | 2 | 3 | 10 | 20 | 50 | 100 | 200 | 500 | 1000 |
|---|---|---|---|---|---|---|---|---|---|---|---|
| Current system | NC | 363434 | 545594 | 662399 | 1041230 | 1280598 | 1604223 | 1855559 | 2113083 | 2464388 | 2740033 |
| | C1 | 1311483 | 779030 | 1070807 | 1471180 | 1845707 | 2120890 | 2081960 | 2494516 | 3337794 | 3635804 |
| | C2 | 138358 | 625172 | 763944 | 1151120 | 1309407 | 1676813 | 2313744 | 2916433 | 3302794 | 3292205 |
| | C3 | 689945 | 710016 | 1003205 | 1343650 | 1447074 | 1819748 | 1922111 | 2424542 | 2907221 | 3224196 |
| | C4 | 1322311 | 939202 | 1020153 | 1948310 | 1942896 | 2158862 | 2312024 | 2961595 | 3040893 | 3957185 |
| | C5 | 1299874 | 508016 | 447533 | 2184984 | 2011414 | 1961587 | 2068387 | 2155563 | 2598096 | 2631549 |
| | | | | | | | | | | | |
| Pipe | NC | 0 | 0 | 0 | 274650 | 545548 | 902639 | 1191761 | 1454490 | 1825663 | 2107541 |
| | C1 | 579100 | 66820 | 307628 | 754782 | 1177608 | 1465530 | 1424433 | 1853479 | 2753620 | 3048692 |
| | C2 | 0 | 14683 | 58510 | 400927 | 576342 | 988731 | 1672038 | 2305916 | 2711960 | 2700636 |
| | C3 | 30911 | 39643 | 236010 | 610572 | 720015 | 1151135 | 1260383 | 1791006 | 2295501 | 2631907 |
| | C4 | 586820 | 175700 | 254039 | 1287942 | 1283153 | 1502586 | 1670054 | 2356962 | 2432769 | 3392554 |
| | C5 | 564627 | 1288 | 647 | 1531861 | 1355232 | 1304201 | 1413665 | 1500109 | 1960429 | 1999834 |
| | | | | | | | | | | | |
| Pipe+LID | NC | 0 | 0 | 0 | 180610 | 403742 | 735983 | 994636 | 1239575 | 1571403 | 1833913 |
| | C1 | 435235 | 31853 | 205783 | 594395 | 981183 | 1247661 | 1207291 | 1602282 | 2407278 | 2683353 |
| | C2 | 0 | 4374 | 27315 | 275503 | 432434 | 808381 | 1439073 | 2002787 | 2375242 | 2362011 |
| | C3 | 10832 | 13901 | 152559 | 463675 | 568173 | 960769 | 1056741 | 1531386 | 1993485 | 2295640 |
| | C4 | 442271 | 106856 | 165356 | 1082850 | 1077049 | 1280177 | 1437899 | 2042621 | 2123354 | 2966933 |
| | C5 | 423441 | 723 | 536 | 1300494 | 1145087 | 1094680 | 1193045 | 1277930 | 1703625 | 1738962 |

[Figure]

*Figure 6* *Future changes in flood volumes (CTFVs) relative to historical conditions under the current drainage system (yellow) and two adaptation scenarios (i.e., Pipe in red and Pipe+LID in green) at various return periods.*

**Specific comments**

L 31: given the delay between submission and publishing I suggest removing "current" from the text. Same for line 81.

**Response**: *Done.*

L 32: I suggest removing "existing" in favor of "past", "recent," "literature" or similar

**Response**: *Done.*

L 40: "Based on the results" –> "Results indicates that"

**Response**: *Done.*

L45: This is an outcome of your research, hence I would not say it is "obvious" but rather something like
"very likely" or "results clearly indicates::" or similar.

**Response**: *Thanks for the suggestion. We have revised it to "results clearly indicate"*

L 46: "greenhouse gas emissions"

**Response**: *Done.*

L 62: The sentence is not clear. Please specify units of the change and in relation to what (e.g., flood
peak, precipitation intensity?)

**Response**: *Thanks for the suggestion. We have revised this sentence to "30% and 40% increase in the*
*precipitation intensity is expected for the 10- and 100-year return periods" (Lines 64-66).*

L 66-69 is again not clear. E.g., non-stationary changes reads awkward. Also, what do you mean by future
hydroclimate?

**Response***: Thanks for the suggestion. We have revised this sentence to "Therefore, it is important to*
*investigate the performance of drainage systems in a changing environment and to assess the potential*
*urban flooding under various scenarios to achieve better adaptations" (Lines 69-72).*

L71-77: As the article has a strong focus on mitigation and adaptation I suggest adding some relevant
references in those areas. See the work by (Alfieri et al., 2016; Arnbjerg-Nielsen et al., 2015; Moore et
al., 2016; Poussin et al., 2012) among others. The few ones currently listed in the article are somehow
hidden in the conclusions.

**Response**: *Thanks for the suggestion. We have expanded literature review and incorporated the*
*suggested references in the revision (Lines 80-83).*

ensembles of several GCMs, and second because "all five GCMs" sounds like if there were only five,
while CMIP5 includes way more than that.

**Response**: *Thanks for the suggestion. In the revision, we have deleted the statement "Unlike most*
*previous studies that only used data from one or two GCM in climate change impact studies on urban*
*floods".*

L 151: Rainfall is a climatic data. Please clarify.

**Response**: *Done.*

L 176: there are –> we considered

**Response**: *Corrected.*

L 181-182: This sentence should be supported by data, graphs or a reference to publications showing the
validation work against historical records.

**Response**: *Thanks for the suggestion. In the revision, we have updated Figure 5a (attached below) by*
*adding a graph on the city land use condition (e.g., green spaces and traffic network) and records of*
*historical flood locations obtained from local water authorities. It is shown that the simulated locations of*
*overloaded pipelines are in good agreement with historical records of flood points.*

[Figure]

***Figure 5*** *Spatial distribution of overloaded pipelines (red colour) induced by the 3-year (left column)*
*and 50-year extreme events (right column) without and with adaptations. The total percentage of*
*overloaded manholes (POM) and ratio of flood volume (RFV) are summarised for each scenario.*

*Descriptions of local land use, mainly the traffic network and green spaces, are provided as the*
*background image in (a).*

L 186-191: This part is difficult to read and understand. Please clarify and add some detail on how the
TFV – return period relationship was derived. Figure 2 currently doesn't help a lot as it is too general,
with no units nor tick marks. For example, if it the grey area is meant to indicate those events that
contribute the most to the annual damage, then it should take at least 50% of the area under the curve in
Figure 2, as its integral is proportional to the total flood risk.

**Response**: *Thanks for the suggestion. As responded to the third general comment, we have revised these*
*sentences to make it clearer and concise (Lines 236-247). Figure 2 is also updated following the*
*suggestions:*

[Figure]

***Figure 2*** *Illustration of flood volume and average total expected total flood volumes (TFVs) as a*
*function of return period under a stationary drainage system. The grey area denotes the average total*
*expected TFVs per year considering all kinds of floods.*

L 191- 195: This statement indicates a strong assumption which is not justified at this stage and sounds
like a speculation. Perhaps the authors want to introduce what is later on indicated by their findings, but I
think at this point this is unjustified, unless the point is supported by stronger evidence and/or some
references.

**Response**: *Thanks for the comment. In the revision, we have revised the relevant descriptions (Lines 242-*
*247).*

L204-205: What is the extent of the enhancement of pipeline diameters in the adapted scenario? I couldn't
find it anywhere in the text.

**Response**: *Thanks for the comment. The number of pipelines of the present-day and adapted systems was*
*323 and 488, with a total pipe length of 251.6 km and 375.4 km, respectively. In the adapted scenarios,*

*the mean pipeline diameter was about 1.73 m, which increased by 53% compared to that of the present-*

*day system. We have clarified this in the revision (Lines 255-258).*

L230-231: Is this 52% a simple average of the percent changes shown in Figure 3? Then I suggest to clarify, as it doesn't necessarily mean the overall projected change in flood risk.

**Response**: *Thanks for the comments. In the revision, we have added more details on the changes, rather*

*than showing the overall average value (Lines 289-292).*

L 254: More correctly "10 magnitudes of rainfall events".

**Response**: *Corrected.*

L 263: 19% should be 49%.

**Response**: *Corrected.*

L 332-333: Not just uncertainties but modeling assumptions as well.

**Response**: *Thanks for the suggestion. We have added more discussions on the assumptions and*

*limitations in the revision (Lines 414-462).*

L 328-329: That's true but perhaps out of the scope of this article, as anyways there is no real damage model to evaluate economic flood losses.

**Response**: *Thanks for the comment. Yes, flood damage is not addressed in this study. We have revised the*

*relevant terms and descriptions in the revision.*

L 358-363: Following the discussions above one should be careful in calling these numbers "flood risk".

Please adapt according to the indications in the discussion points above.

**Response**: *Thanks for the suggestion. We have changed "flood damage" or "flood risk" to "flood*

*volume" throughout the text in the revision.*

L 605-606: I suggest including the period "2020-2040" in the caption for better understanding the graph.

Table 1: Which are the units in the table? Please specify units and the storm duration related to the precipitation intensity values listed (key parameter to understand such values).

**Response**: *Thanks for the suggestion. We have added the period "2020-2040" in the caption. This table*

*shows the future change factor of precipitation at various return periods. It is dimensionless. The changes*

*are multiplied to the present rainfall time series to obtain climate change scenarios as inputs to our*

*model (see response to general comment 2).*

Figure 5: Please choose a more visible way of indicating overloaded pipelines, perhaps with a thicker line and/or a different color. Also the POM is currently mistakenly written as "NOM" in the 6 panels.

**Response**: *Thanks for the pointing out the typo. We have replaced "NOM" with POM. The illustration of*

*overloaded pipelines is a direct output from the SWMM model. At present, it is not easy to highlight the*

*pipelines given the hard-coded model user interface. Instead, we tried to update the figure with larger*

*color contrast for better illustration. In addition, we have added city land use information (i.e., green*

*spaces and traffic network) and records of historical flood pints obtained from the local water authorities*

*in the updated figure.*

[Figure]

**Figure 5** *Spatial distribution of overloaded pipelines (red colour) induced by the 3-year (left column) and 50-year extreme events (right column) without and with adaptations. The total percentage of overloaded manholes (POM) and ratio of flood volume (RFV) are summarised for each scenario.*

*Descriptions of local land use, mainly the traffic network and green spaces, are provided as the*
*background image in (a).*

Figure 6: Add units in the axis labels. E.g.: "[-]" for dimensionless. Also, note the typo in the x-axis label.

**Response**: *Thanks for the suggestion. We have updated the figure in the revision.*

Figure 7: Negative values for risk reduction means increasing risk. Please reverse graphs with positive
values (plus fix the typo rish -> risk)

**Response**: *Thanks for the suggestion. We have updated the figure and corrected the typo in the revision.*

_______________________________

**Anonymous Referee #2:**

**SHORT COMMENTS IN THE JOURNAL STYLE**

**Scientific questions:**

Adaptation effects on drainage performance in a context of climate change (CC) is relevant. Novel concepts. Try to quantify the impact adaptation measures is potentially new if appropriately developed in single case studies. Substantial conclusions. Not attended yet, due to insufficiently explained datasets and methods. Scientific methods and assumptions. Not clearly outlined. Results vs interpretations / conclusions. Unattended. Description. Pretty obscure. Authors proper credit. Ok! but not all is new. Title. OK! but to be revised in case of revision. Summary. Unbalanced on Climate trends when the most interesting part is adaptation. Overall presentation. Lacking of context outline. Language. To be revised by a mother tongue, that I am not. Formulae. Not expert enough to say. Parts to modify. Develop 1, 4 & 5, Clarify 2a & 2e, Reduce 2b, Delete 3b, Modify Fig. 1 & 5. References. Ok.

**Response***: We greatly appreciate the reviewer for the constructive comments and suggestions to improve our manuscript. In the revision, we have 1) added more details on the datasets and methods, 2) added more discussions on the assumptions and limitations, 3) modified the relevant statements and figures which are unclear or inaccurate, 4) revised the specific sections as suggested, 5) invited a native speaker to proof-read the paper. More details of our responses to each comment are provided as follows.*

*Note: the line numbers as mentioned in the response below refer to those in the cleaned version of manuscript.*

**EXTENDED COMMENTS**

Introduction All key definitions should be provided here. Flood risk is the probability an hazard has to generate damages (UNISDR, ISO etc: : :), not a probability of a disastrous flood only (that is hazard occurrence). Should be wise to specify to whom this work is addressed, since very essential information of the case study is missing (see next sections).

**Response***: Thanks for the comments. We agree with the reviewer that flood risk refers to the probability of a hazard to cause damage. In this study, we investigate the potential changes in flood volume (TFV) under various scenarios of climate changes and explore the role of adaptation and mitigation in regulating such changes. We acknowledge that the TFV is a hazard indicator, while flood damage is tightly linked to socio-economic conditions which is not addressed in this study. We have clarified this concept (Lines 239-247) and revised all relevant terms throughout the manuscript in the revision.*

*This study investigates the performance of drainage system under climate change scenarios, which has great implications for adaptation and mitigation strategies for the study region, which has experienced increasing flood events (Lines 130-141). Comparing the reduction of flood volume by climate mitigation (via reduction of greenhouse gas emissions) and local adaptation (via improvement of the drainage system) indicates that local adaptations are more effective than climate mitigation in reducing future flood volumes. This study also has important implications for the research community on drainage system design and modeling in a changing environment. We emphasize the importance of considering adaptations in assessing climate change impacts on future urban floods. In the revision, we have provided more detailed information on the case study region, research background and the implications following the suggestions (Lines 102-112, 130-141, 488-496).*

2. Material and Methods 2a) (i) A characterization of the hazard (rainfall) in Hohhot City is missing.

**Response***: Thanks for the comments. In the study region, most rain storms fall between June and August, a period that accounts for more than 65% of the annual precipitation. In the revision, we have provided more descriptions on the rainfall characterizations and flood hazards in the study region (Lines 126-129 and 134-142).*

*It should be noted that the input rainfall time series for the model are not the original historical observations. Rather, it is based on the storm intensity formula (SIF), which is used to estimate the design rainfall for each return period. The modeling practice mainly follows the standard procedure in urban drainage modeling in China, as documented in the national code for design of outdoor wastewater engineering (MOHURD, 2011). Specifically, the SIF represents an Intensity-Duration-Frequency (IDF) relationship, and is commonly used in the literature to estimate design rainfall hydrographs (Berggren et al., 2014; Cheng and AghaKouchak, 2014; Panthou et al., 2014; Willems, 2000; Zhou et al., 2012). Subsequently, the Chicago Design Storms (CDS) approach is applied to derive the design storms from the local SIF for the SWMM model as used in this study. The detailed procedures in using SIF to obtain the CDS design storms can be found in Chinese National Technical Guidelines for Establishment of Intensity-Duration-Frequency Curve and Design Rainstorm Profile (MOHURD, 2014) and have been well adopted in a number of Chinese urban drainage designs (Wu et al., 2016;Yin et al., 2016; Zhang et al., 2008; Zhang et al., 2015).*

*For the case study, the local rainfall is characterized by the SIF (q=635\*(1+0.841\*lg(P))/t^0.61), which*

*is obtained from local weather bureau. 10 return periods are considered in the paper and a 4-hour*

*rainfall time series is generated for each return period at a 10-minute interval. The technical details in*

*using SIF to derive the CDS rainfall are given in the following. As shown in the Equation 1, the q is the*

*average rainfall intensity, t is the storm duration and P is the design return period. The typical temporal*

*resolution considered in SIF for urban drainage simulations is minutes. A, b, c and D are regional*

*parameters governing the IDF relations among rainfall intensity, return period and storm duration. For a*

*given return period, the SIF is fitted into the Horner's equation as Eq.2:*

$$q = \frac{A(1 + Dlg(P))}{(t + b)^c} \qquad\qquad Eq.\ (1)$$

$$i = \frac{a}{(t + b)^c} \qquad\qquad Eq.\ (2)$$

*The synthetic hyetograph based on the Chicago method is computed using Eq. 2 and an additional*

*parameter r (where 0< r <1) which determines the relative location of peak intensity (with respect to*

*time), $t_p=r*t$. The time distribution of rainfall intensity is described after the peak $t_a = (1-r)*t$ and before*

*the peak $t_b=r*t$ by Eq. (3) and (4). $i_b$ is the instantaneous rainfall intensity before the peak , $i_a$ is the*

*instantaneous rainfall intensity after the peak.*

$$i_a = \frac{a[\frac{(1-c)t_a}{(1-r)} + b]}{(\frac{t_a}{(1-r)} + b)^{1+c}} \qquad\qquad Eq.\ (3)$$

$$i_b = \frac{a[\frac{(1-c)t_b}{r} + b]}{(\frac{t_b}{r} + b)^{1+c}} \qquad\qquad Eq.\ (4)$$

*By following the above procedure, a 4-hour rainfall time series can be generated for each return period*

*with the peak located in the center of the period. In the revision, we have added more details about the*

*rainfall and methods in the revision (Lines 189-220).*

| Maximum (Initial) Infiltration Capacity (Akan, 1993) | | |
|---|---|---|
| **Soil Type** | *(in/hr)* | *(mm/hr)* |
| Dry sandy soils with little or no vegetation | *5.0* | *127* |
| Dry loam soils with little or no vegetation | *3.0* | *76.2* |
| Dry clay soils with little or no vegetation | *1.0* | *25.4* |
| Dry sandy soils with dense vegetation | *10.0* | *254* |
| Dry loam soils with dense vegetation | *6.0* | *152* |
| Dry clay soils with dense vegetation | *2.0* | *51* |
| Moist sandy soils with little or no vegetation | *1.7* | *43* |
| Moist loam soils with little or no vegetation | *1.0* | *25* |
| Moist clay soils with little or no vegetation | *0.3* | *7.6* |
| Moist sandy soils with dense vegetation | *3.3* | *84* |
| Moist loam soils with dense vegetation | *2.0* | *5.1* |
| Moist clay soils with dense or no vegetation | *0.7* | *18* |

*To further address the concern, we have conducted a set of sensitivity experiments in the revision, see*

*added Table 1 and revised Figure 7. Specially, we used three possible infiltration values corresponding to*

*the first three soil types (i.e., dry sand, loam and clay soils with little or no vegetation) as listed in the*

*above table. The parameters associated with each possible infiltration value are shown in the table below:*

**Table 1** Infiltration parameters for three categories of soil in the SWMM simulation

| **Infiltration parameters\*** | *MaxRate* | *MinRate* | *Decay rate* | *DryTime* |
| | *[in/hr]* | *[in/hr]* | *[1/hr]* | *[days]* |
|---|---|---|---|---|
| *Dry loam with little or no vegetation* | *3* | *0.5* | *4* | *7* |
| *Dry sand with little or no* | *5* | *0.7* | *5* | *5* |

| | | | |
|---|---|---|---|
| *vegetation* | | | |
| *Dry clay with little or no vegetation* | *1* | *0.3* | *3* | *9* |

* To describe Horton infiltration method in SWMM, four basic infiltration parameters are required (Rossman and Huber, 2016) :
* MaxRate: Maximum infiltration rate on Horton curve
* MinRate: Minimum infiltration rate on Horton curve
* Decay: Decay rate constant of Horton curve
* DryTime: Time it takes for fully saturated soil to dry

*The original Figure 7 shows the comparison of benefits of climate mitigation and two adaptation*

*strategies in reducing flood volume, based on the soil category 'Dry loam with little or no vegetation'.*

*Here, we revised the Figure 7 by showing the uncertainty range (i.e. the error bar) arising from the*

*representation of different soil conditions in the drainage model. It is shown that magnitudes of estimated*

*benefits differ to some degree, nevertheless, the benefits of the designed adaptation measures in reducing*

*urban flood volumes were found to be robust regardless of soil conditions, and such benefits exceeded*

*those of climate change mitigation, confirming our major conclusions found in this study. We have*

*included the relevant descriptions and results in the revised manuscript (Lines 376-385).*

[Figure]

**Figure 7** *Comparison of benefits of climate mitigation and two adaptation strategies in reducing*

*urban flood volumes with changes in precipitation intensities for various return periods, and*

*with related variations (boundary bars) as a result of uncertainty arising from local soil*

*conditions.*

*the permeable soil and coverage rates in the study region, which prevents us from representing these*
*individual features/parameters in the model. Instead, the second adaptation scenario is designed to*
*investigate the effects of increased permeable surfaces on flood volume, and is reflective of the combined*
*effects of infiltration-related measures, including permeable pavements, infiltration trenches and green*
*roofs. That is, a simplified approach by altering the subcatchment imperviousness was adopted due to the*
*limitation of data availability in the study region. Specifically, we derived such information by comparing*
*the current and planned landuse maps and incorporated the changes in landuse and imperviousness (see*
*the updated Figure 1d) in the adaptation scenario. The figure 1d shows the difference in weighted mean*
*imperviousness (WMI) calculated for each subcatchments (different parts of the watersheds) in the*
*current and planned maps, which is used to indicate the area potential for adaptation based on the city*
*plan. For example, a subcatchment with higher positive changes in the WMI indicates that the area is*
*planned to have a land use type with lower imperviousness and therefore is assumed to be more suitable*
*for LID planning, and vice versa. We have clarified this with more discussions in the revision (Lines 260-*
*278).*

[Figure]

(iv) Can the authors provide some information about last disastrous floods in the case study? Areas affected the most, etc.

**Response***: Thanks for the comment. During the major flood event on 11 July 2016, the city, especially the western portion of the watershed, was hit by an extreme rainfall event that featured more than 100 mm of rain in 3 hours. The local meteorological department issued the red warning of rainstorm. The flood event led to the cancellation of at least 8 flights and 17 trains, and delays of several transportation systems. In particular, in the central area, the flood event caused severe traffic jams on major streets (see the photos below) and resulted in a number of flooded residential buildings. We have added more descriptions on this in the introduction of the study region in the revision (Lines 136-142).*

[Figure]

(Photos: yjhlnews.com(left) and chinanews.com(right))

*To provide more background information on historical flood events in the study area, we have included a*

*map describing historical flood records and city traffic network in Figure 5a. It is obvious that the central*

*portion of the city is the most affected region due to the low service level of its drainage system. We have*

*updated the Figure 5 and added more descriptions in the revised manuscript (Lines 314-315).*

[Figure]

***Figure 5****: Spatial distribution of overloaded pipelines (red colour) induced by the 3-year (left column)*
*and 50-year extreme events (right column) without and with adaptations. The total percentage of*
*overloaded manholes (POM) and ratio of flood volume (RFV) are summarised for each scenario.*
*Descriptions of local land use, mainly the traffic network and green spaces, are provided as the*
*background image in (a).*

2b) (i) It's quite normal to use more than one GCMs .

**Response***: Agreed. We have deleted relevant statements in the revision.*

(ii) Reader expects to learn from the expected changes in rainfall (mm and in which month) but no
information is provided on this topic.

**Response***: Thanks for the comments. Readers can refer to the Table 2 which summarizes the change*
*factors in extreme precipitation intensity of various return periods. It should be noted that the input*
*rainfall time series for the model are not the original historical observations. Rather, it is based on the*
*storm intensity formula (SIF), which is used to estimate the design rainfall for each return period. The*
*modeling practice mainly follows the standard procedure in urban drainage modeling in China, as*
*documented in the national code for design of outdoor wastewater engineering (MOHURD, 2011). Please*
*see the response to comment 2 for details. We have added more details on this in the revision (Lines 189-*
*233).*

2c) (i) Which rainfall information has been used to run SWMM [dataset length (years) and type (daily,
three hourly, hourly, etc.)]?

**Response***: Thanks for the comments. The rainfalls as inputs for the model are based on artificial rains in*
*the format of Chicago Design storms derived from historical rainfall records following the standard by*
*the local weather bureau and the national code for design of outdoor wastewater engineering (MOHURD,*
*2011). The rainfall period is 4 hours at sub-hour (i.e., 10 minute) time step. Please see the response to*
*comment 2 for details. We have added more details on this in the revision (Lines 189-220).*

2e) (i) The adaptation measures considered are to reduce the amount of water that run off. This is one side
of the problem. The other one is to slow down the water speed. And for this no measure is considered:
there is a wide range of measures for semi-arid contexts commonly used for this. I recommend to consider
it or explain why you don't.

**Response***: Thanks for the comments. We agree that slowing down the water speed could be an alternative*
*adaptation approach for attenuating runoff peak and reducing flood volume (Messner et al., 2006;*
*Floodsite, 2009). We note that the water speed is influenced by, among others, the gradient and flow*
*resistance of the bed of the water course (Ashley et al., 2007) and such information is not available at the*
*sub-catchment scale in the study region.*

*There are two main reasons that we did not consider the measures by attenuating the water speed in our*
*designed adaptation approach. First of fall, although some of the LID measures are primarily designed to*
*slow down the flow speed, i.e., vegetated swales, most of the LID measures can reduce both runoff volume*

*and flow speed at the same time. Constrained by the one-dimensional SWMM modelling approach in this*

*study, the performances of LID measures were mainly evaluated according to their effects in reducing*

*water volume from overloaded manholes (Oraei Zare et al., 2012;Lee et al., 2013). To examine whether*

*flood retention of a given event is induced by runoff volume or the internal speed control function in the*

*model is difficult and requires detailed data for model validations. Specifically, the required information*

*about surface roughness, soil conductivity, and seepage rate were unavailable at the subcatchment scale*

*in the study region. Based on the available datasets on current and future landuse maps, this study tends*

*to apply and assess adaptation measures that mainly affect the surface imperviousness. We have added*

*the discussions in the revised manuscript (Lines 436-449).*

(ii) How Authors have determined the impact of individual adaptation measures (permeable pavements,
trenches, green roof) over run off reduction? This should be explained.

**Response***: Thanks for the comments. As clarified in the response to 2a (iii), the second adaptation*

*measure is mainly designed to investigate the impacts of increased permeable surfaces on flood volume*

*reductions by altering the imperviousness of subcatchments to represent the infiltrated and detained*

*water volume in the runoff-generation process. That is, the individual measures related to permeable*

*pavements, trenches, green roof are not considered separately but represented in a combined and*

*simplified approach. Thus, we are not able to explicitly assess the performance of these individual*

*measures on flood reductions in details. We have added discussions on this in the revision (Lines 260-*
*278).*

3) Results 3b) (i) I don't understand the approach: Mitigation is expected to impact on CC at long term
(decades: : :). Drainage system is expected to reduce CC impacts at short-medium term (1-5 years). Is
obvious that adapting we can't expect to see effects on rainfall: : :

**Response***: Thanks for the comments. Mitigation refers to climate mitigations via reduction of greenhouse*
*gas emissions. The mitigation effects are assessed here by comparing the results based on RCP8.5*
*emission scenario (which is a business-as-usual scenario) and RCP2.6 scenario (which considers the*
*reduction of greenhouse gas emission). Climate mitigation via reducing greenhouse gas emissions is*
*expected to influence precipitation characteristics and thus the subsequent flood hazards (i.e., flood*
*volume in this study). Adaptation measures are localized and here refer to the specific design/update of*
*drainage system. The possible land surface-atmosphere interactions which would indirectly affect the*
*rainfall and floods are not considered in this study. We have clarified this in the revised manuscript*
*(Lines 154-160).*

4) Uncertainties & Limitations (i) The consideration of the state of drainage system could be a limitation
of this study? A drainage system obstructed by vegetation, waste or artefacts (cables, pipes, temporary
constructions) can make the outcomes of the SWMM quite distant from the real world. And change also
recommendations: : : that need to be extended to waste sector.

**Response***: Thanks for the comments. We agree with the reviewer that the state of drainage system could*
*affect its conveyance capacity and thus the system performance to various degrees. In some cases, floods*
*are not induced by the exceedance of drainage capacity, but by the deterioration of drainage system itself,*
*e.g., aging network, pipe deterioration, blockage, construction failures and local external factors*
*(Dawson et al., 2008; CIRIA, 1997; Davies et al., 2001). Previous studies with a focus on sewer*
*inspection and condition assessment, maintenance and rehabilitation strategies, have highlighted the*
*need for labor-intensive field investigations for collecting information on the waste status and relocations,*
*and such studies are often limited to certain areas (Ana and Bauwens, 2007; Fenner, 2000). In fact,*
*assessment of drainage conditions requires detailed datasets, which has been recognized as a great*
*challenge in applications. For example, in Europe, water service data collections mainly cover pipe*
*length, age, material, diameter and location (Stone et al., 2002; Ana and Bauwens, 2007), while the*
*assessment of pipe conditions are often managed by  separate and specialized programs.*

*Quantifying the impacts of drainage system states on urban flood volumes is not trivial, however, it was not within the scope of this study to take into account the actual state of the pipe system due to difficulties involved in collecting field data and selecting and using appropriate methods for reasonable assessment of pipe conditions. Such studies usually require comprehensive efforts on the material, data and method, (e.g., Dawson et al. 2008; Chae and Abraham 2001; Chughtai and Zayed 2008), which is not the focus of this paper. We acknowledge that the hydraulic performance may be overestimated without considering the drainage conditions and the waste section in the SWMM modeling approach (Pollert et al., 2005). In the revision, we have added more discussions on the impacts of pipe conditions on system performance, which should be addressed in the future study (Lines 422-434).*

Figures 1 & 5: scale is not showed: how large are blocs contoured by drainage network?
**Response***: Thanks for the comments. In the revision, relevant figures have been updated by including a*
*scale bar (see the attached figures below).*

[Figure]

*Figure 1* *Land use of the study region for the year 2010 (a) and 2020 (b). Pipe network description of current and planned drainage systems (c). Difference in Weighted Mean Imperviousness (WMI) between year 2010 and 2020 (d).*

[Figure]

**Figure 5** *Spatial distribution of overloaded pipelines (red colour) induced by the 3-year (left column) and 50-year extreme events (right column) without and with adaptations. The total percentage of overloaded manholes (POM) and ratio of flood volume (RFV) are summarised for each scenario. Descriptions of local land use, mainly the traffic network and green spaces, are provided as the background image in (a).*

6. Manuscript's title Show the name of the case study and the country. Limit to Adaptation, delete
mitigation, delete risk.

**Response**: *Thanks for the suggestion. In the revision, we replaced the "risk' with "volume" and added*
*the study region name and country. But we tend to keep "mitigation" in the title as we believe it is*
*important although we emphasize the importance of considering adaptation in assessing climate change*
*impacts on future urban floods. This is because the role of adaptation in reducing flood volume is*
*highlighted through comparing with the reduced flood volume by climate mitigation. Indeed, comparing*
*the reduction of flood volume by climate mitigation (via reduction of greenhouse gas emissions) and local*
*adaptation (via improvement of the drainage system) indicates that local adaptations are more effective*
*than climate mitigations in reducing future flood volume.*

[revised manuscript text omitted]

---

## Referee Report (RR1)

**Review of the paper, 'Impacts of future climate change on urban flood volumes in Hohhot City in Northern China: benefits of climate mitigation and adaptations [hess-2016-369]'**

By Qianqian Zhow, Guoyong Leng, and Maoyi Huang

In this paper, the authors assessed the benefits of mitigating climate change by reducing greenhouse gas emissions and locally adapting to climate change by modifying drainage systems to reduce urban flooding under various climate change scenarios through a case study conducted in Northern China. As the authors commented, this study accounted for the effects of both climate change mitigation and adaptation together in a consistent framework different from previous studies related to this issue. It would be the most important research outcome of this paper.

Now, the paper presents specific and easily identifiable advance in knowledge, which can be usefully applicable to the profession. With the subject within the scope of the journal, the authors describe the paper's research purpose, main findings, and conclusions in a more concise way. However, I have still several concerns about the paper's method, data, results and conclusions, which need to be modified for final publication. The details are summarized below:

1. **Cost-effectiveness Issue**: The authors compared the reduced total flood volumes by climate change mitigation and drainage system adaptation as functions of return period in Figure 7. From this result, the authors highlight the effectiveness of system adaptations in reducing future flood volumes and comment that this has important implications for the research community and decision-makers involved in urban flood management in the 'Summary and Conclusions' part. Because this study only focused on the future changes in urban flood volume followed by the applied different scenarios, there still remains a limitation of this study related to 'cost-effectiveness issue'. In the 'Uncertainties and Limitations' part, the authors added a sentence in lines 453-454 but it seems not enough to explain the limitation of this study. Of course, there are already many meaningful discussions in the 'Uncertainties and Limitations' part but I think the main discussion issue of this part should be 'cost-effectiveness' with valuable references.

2. **Introduction**: The authors need to clarify each condition for other researchers' outcomes. Please pay careful attention to the summary of other researchers' outcomes. For example,

    - Lines 64-66: For example, in Danish design guidelines for urban drainage, a 30% and 40% increase in the precipitation intensity is expected for the 10- and 100-year return periods, respectively (Arnbjerg-Nielsen, 2012). I read this reference, Arnbjerg-Nielsen (2012) but there were research outcomes about the delta change for Denmark. It seems hard to figure out where this sentence is originated from.

    - Lines 75-77: For example, Ashley et al. (2005) showed that flooding risks may increase by almost 30 times in comparison to current situations, and effective adaptation measures are required to cope with the increasing risks in the UK. Please refer to any information about flooding risks and current situations.

    - Lines 77-79: Larsen et al. (2009) estimated that future extreme one-hour precipitation will increase by 20%~60% throughout Europe. Please refer to more specific information about the future year.

3. **Materials and Methods – a. Study region**: The authors need to re-organize the structure of this part for helping readers understand the contents clearly (For example, 1) General comments; 2) Major flood event on 11 July 2016; 3) Necessity for adaptation policies; 4) Plan for the year 2020)

4. **Materials and Methods – c. Urban drainage modeling**: The authors need to summarize only important points, which are directly used for urban drainage modeling in the Storm Water Management Model. Please move several parts to the 'Supplementary Materials' or 'Appendix' part such as Table 1, Equations 1-4 and the related explanations.

5. **4-hour rainfall time series**: Do the authors have any reasons for selecting 4-hour rainfall time series in this study? In lines 223-224, the authors commented that the annual maximum daily precipitation was determined for both historical and future periods.

6. **Figure 2**: The authors need to revise the explanation about Figure 2 (lines 236-245) and the title of the Figure 2 for helping readers understand the contents in a more concise way.

7. **Abbreviation**: Please carefully use the word abbreviation. For example,

   - Line 236: The TFV -> The total flood volume (TFV)

   - Line 282: climate change -> climate change (CC)

8. **Weighted mean imperviousness (WMI)**: Please define how to calculate the WMI.

9. **Figure 3**: The authors need to revise the explanation about Figure 3 (lines 282-292) for helping readers understand the contents in a more concise way (for example, 1) max; 2) median; 3) min; 4) multi-model ensemble median).

   - 1yr -> 1

10. **Figure 4**: Please add the x-axis title.

11. **Ratio of flood volume (RFV) in Figure 5**: Please define how to calculate the RFV.

12. **TFV reduction (%)**: Please define how to calculate the TFV reduction. In Figures 4 and 7, the readers can see the related results, mainly composed of negative and positive numbers, respectively. Especially, in Figure 7, the authors need to explain about the negative values for the return periods of 100 and 200 years.

13. **Lines 83-85**: There are already many journal papers related to this issue. The authors revise the 'Introduction' part to emphasize the novelty of this study.

14. **Lines 122-123**: Please add the related reference here.

15. **Lines 134-136**: Please add the related reference here.

16. **Lines 200-202**: Please add the related reference here.

17. **Lines 251-255**: If this explanation is about the adaptation plan for the year 2020, please clarify the plan for the year 2020 in the 'Introduction' part and this part to help readers understand the contents in a more concise way.

18. **Lines 260-261**: It seems better to start this paragraph with the explanation about the second adaptation scenario. The authors need to re-organize this paragraph.

19. **Lines 269-272**: Please give additional information on this part. It is hard to understand the meaning of this sentence.

20. **Lines 305-308**: Please explain the main reason of these results with discussions.

21. **Line 315**: Please clarify where the historical flood points are.

---

## Author Response (AR2)

**Impacts of future climate change on urban flood volumes in Hohhot City**

**in Northern China: benefits of climate mitigation and adaptations**

Qianqian Zhou[1,2], Guoyong Leng[2,*], Maoyi Huang[3]

[1]School of Civil and Transportation Engineering, Guangdong University of Technology,

Waihuan Xi Road, Guangzhou 510006, China

[2]Joint Global Change Research Institute, Pacific Northwest National Laboratory, College Park

MD 20740, USA

[3]Earth System Analysis and Modeling Group, Pacific Northwest National Laboratory, Richland,

WA 99352, USA

________________________

*Corresponding author address: Guoyong Leng, Joint Global Change Research Institute, Pacific
Northwest National Laboratory, College Park MD, 20740.
E-mail: guoyong.leng@pnnl.gov

**Contents**

**Abstract**

The author would like to thank all reviewers for their dedicated time reviewing the manuscript and for their useful and constructive suggestions. All comments by the reviewers were carefully addressed and the manuscript has substantially benefited from the proposed changes. Here below, the author would like to clarify the changes regarding all comments, which are repeated in grey boxes. The following convention is applied to denote modification in the original manuscript: new text.

**1 Reviewer #1**

**Reviewer #1 Comment 1**

I see that the authors made a substantial revision of the article by addressing most of my previous comments. I think the article can become suitable for publication, provided that the comments below are adequately addressed.

**Response**

We greatly appreciates the reviewer for the thoughtful and encouraging comments on this manuscript. In the revision, we have tried our best to address the comments. More details of response to each specific comment are provided below.

**Reviewer #1 Comment 2**

Line 154: As I said in the previous review, please remove "all", as there are more than five GCMs in CMIP5.

**Response**

Thanks for your suggestion. We have removed "all" in the revision.

**Reviewer #1 Comment 3**

Line 227:  There is currently no information nor figures/tables on the outcomes of the goodness of fit tests, hence I suggest to mention at least some quantitative results in the text.

**Response**

Thanks for your comments. Please note that our test returns a decision for the null hypothesis of an extreme value distribution rather than a quantitative value, based on MATLAB tool (https://www.mathworks.com/help/stats/adtest.html). Results show that our test fails to reject the null hypothesis at the 5% significance level, suggesting that the fitted extreme value distribution could be used to describe the extreme precipitation distribution in the study region.

In the revision, the sentence "The goodness-of-fit was tested by calculating the Kolmogorov–

Smirnov and Anderson–Darling statistics" was revised to

Kolmogorov–Smirnov and Anderson–Darling statistics show that the hypothesis regarding the extreme value distribution is not rejected. That is, the fitted distribution could be well used to describe the extreme precipitation distribution in the study region.

> ### Reviewer #1 Comment 4
>
> Line 381: "boundary bars" are commonly referred to as "whiskers".

**Response**

Thanks for your comments. In the revision, we have changed "boundary bars" to "whiskers".

> ### Reviewer #1 Comment 5
>
> Figure 3: In this figure it is not clear which lines/bars refer to TFV and which other to the
> percent increase. Please clarify in the legend (and/or in the caption).

**Response**

Thanks for your comments. We have re-plotted the figure and added more descriptions in the figure caption. Please check the updated figure and caption below.

[Figure]

**Figure 3** Changes in total flood volume (TFV) as a function of precipitation intensity at various return periods under RCP8.5 scenario without mitigation and adaptations. Red solid line represents the multi-model ensemble median TFV with shaded areas denoting the ensemble range. Red dashed line is the TFV under present condition. Box plots show the relative changes in TFV by 2020–2040 relative to present condition. Box edges illustrate the 25th and 75th percentile, the central mark is the median and whiskers mark the 5th and 95th percentiles.

**2 Reviewer #2**

**Reviewer #2 Comment 1**

In this paper, the authors assessed the benefits of mitigating climate change by reducing greenhouse gas emissions and locally adapting to climate change by modifying drainage systems to reduce urban flooding under various climate change scenarios through a case study conducted in Northern China. As the authors commented, this study accounted for the effects of both climate change mitigation and adaptation together in a consistent framework different from previous studies related to this issue. It would be the most important research outcome of this paper.

Now, the paper presents specific and easily identifiable advance in knowledge, which can be usefully applicable to the profession. With the subject within the scope of the journal, the authors describe the paper's research purpose, main findings, and conclusions in a more concise way. However, I have still several concerns about the paper's method, data, results and conclusions, which need to be modified for final publication.

**Response**

We greatly appreciates the reviewer for the thoughtful and encouraging comments on this
manuscript. In the revision, we have tried our best to address the comments. More details of
response to each specific comment are provided below.
Note: line numbers in the response refer to those in the cleaned version of manuscript.

* * *
**Reviewer #2 Comment 2**

Cost-effectiveness Issue: The authors compared the reduced total flood volumes by climate change mitigation and drainage system adaptation as functions of return period in Figure 7. From this result, the authors highlight the effectiveness of system adaptations in reducing future flood volumes and comment that this has important implications for the research community and decision-makers involved in urban flood management in the 'Summary and Conclusions' part. Because this study only focused on the future changes in urban flood volume followed by the applied different scenarios, there still remains a limitation of this study related to 'cost-effectiveness issue'. In the 'Uncertainties and Limitations' part, the authors added a sentence in lines 453-454 but it seems not enough to explain the limitation of this study. Of course, there are already many meaningful discussions in the 'Uncertainties and Limitations' part but I think the main discussion issue of this part should be 'cost effectiveness' with valuable references.

**Response**

Thanks for your comments. We agree that a limitation of this study is lack of assessment of costs
and benefits of adaptation measures from the economic perspective. In the revision, we have
added the following discussions in lines 454-465.

In particular, the cost-effectiveness of the proposed adaptation measures should be
accounted for. Indeed, a major limitation of this study is lack of assessment of costs and
benefits of adaptation measures from the economic perspective. In fact, besides the
effectiveness of proposed adaptation measures in reducing flood volume, assessment of the
associated economic costs is essential for flood risk management (Rojas et al., 2013; Veith
et al., 2003; Hinkel et al., 2014; Aerts et al., 2014; Ward et al., 2017). For example, Ward
et al. (2017) showed that investments in urban flood protections with dykes are not
economically attractive everywhere. Higher investment and maintenance costs may
prohibit the implementation of adaptation strategies as proposed in this study. Future
efforts should therefore be devoted to building a framework for assessing the costs and
benefits of urban flood reduction measures and examine whether the reduced losses are
higher than the costs of investments and maintenance of these measures.

Reviewer #2 Comment 3

Introduction: The authors need to clarify each condition for other researchers' outcomes. Please pay careful attention to the summary of other researchers' outcomes. For example,

Lines 64-66: For example, in Danish design guidelines for urban drainage, a 30% and 40% increase in the precipitation intensity is expected for the 10-and 100-year return periods, respectively (Arnbjerg-Nielsen, 2012). I read this reference, Arnbjerg-Nielsen (2012). but there were research outcomes about the delta change for Denmark. It seems hard to figure out where this sentence is originated from.

**Response**

Thanks for your comments. We have revised this sentence to:

For example, Arnbjerg-Nielsen (2012) reported that the design intensities in Denmark are projected to increase by 10-50% for return periods ranging from 2 to 100 years."

Reviewer #2 Comment 4

Lines 75-77: For example, Ashley et al. (2005) showed that flooding risks may increase by almost 30 times in comparison to current situations, and effective adaptation measures are required to cope with the increasing risks in the UK. Please refer to any information about flooding risks and current situations.

**Response**

Thanks for your comments. We have revised this sentence to:

For example, Ashley et al. (2005) showed that flood risk (i.e., occurrence of pluvial flooding) in four UK catchments may increase by almost 30 times by 2080s compared to current conditions around the year 2000, and effective adaptation measures are required to cope with the increasing risks.

**Reviewer #2 Comment 5**

Lines 77-79: Larsen et al. (2009) estimated that future extreme one-hour precipitation will increase by 20%~60% throughout Europe. Please refer to more specific information about the future year.

**Response**

Thanks for your comments. We have revised this sentence to:

Larsen et al. (2009) estimated that future extreme one-hour precipitation will increase by

20%~60% throughout Europe by 2071–2100 relative to 1961–1990

**Reviewer #2 Comment 6**

Materials and Methods – a. Study region: The authors need to re-organize the structure of this part for helping readers understand the contents clearly (For example, 1) General comments; 2) Major flood event on 11 July 2016; 3) Necessity for adaptation policies; 4) Plan for the year 2020).

**Response**

Thanks for your comments. We have re-organized the structure of this sector following your suggestions. Please check the revision in lines 121-146.

**Reviewer #2 Comment 7**

Materials and Methods – c. Urban drainage modeling: The authors need to summarize only important points, which are directly used for urban drainage modeling in the Storm Water Management Model. Please move several parts to the 'Supplementary Materials' or 'Appendix' part such as Table 1, Equations 1-4 and the related explanations?

**Response**

Thanks for your comments. In the revision, we have refined the methodology section and moved

Table 1, Equations 1-4 and related explanations to the Supplementary Materials following your suggestion.

> **Reviewer #2 Comment 8**
>
> 4-hour rainfall time series: Do the authors have any reasons for selecting 4-hour rainfall
> time series in this study? In lines 223-224, the authors commented that the annual maximum
> daily precipitation was determined for both historical and future periods.

**Response**

Thanks for your comments. In this study, a 4-hour rainfall time series was generated for each return period at 10-minute interval. The duration of design rainfalls (i.e. 4-hour rainfall) is selected based on the time of concentration (ToC) of the watershed in the study region, according to the design principles as outlined in [Butler and Davies, 2010; Chow et al., 2013]. In the revision, we have added the following explanations in Lines 197-200.

A 4-hour rainfall time series was generated for each return period at 10-minute intervals.

The duration of design rainfalls (i.e., 4-hour rainfall) is selected based on the time of concentration (ToC) of the watershed in the study region, according to the design principles as outlined in Butler and Davies (2010) and Chow et al. (2013).

Yes, the annual maximum daily precipitation was determined for both historical and future periods. In this study, we used the delta change factor to derive future climate scenarios as inputs into our flood drainage model. Specifically, for each year, the annual maximum daily precipitation was determined for both historical and future periods. The generalised extreme value (GEV) distribution was then fitted separately to the two sets of daily values (Coles 2001;

Katz et al. 2002). The goodness-of-fit was tested by calculating the Kolmogorov–Smirnov and

Anderson–Darling statistics. The value corresponding to each return period was estimated based on the GEV distribution and the changes between future and historical periods were calculated as the change factors. The derived change factor for each return period was then multiplied by the historical design CDS rainfall time series to derive future climate scenarios as inputs into our urban drainage model. This approach was adopted due to two reasons: 1) Future hourly rainfall data is not readily available; 2) the relative climate change signal simulated by GCMs are argued to be more reliable than the simulated absolute values (Ho et al. 2012).

More details can be found in the revised methodology section and supplementary materials.

Discussions on the methodology can also be found in the section "Uncertainties and Limitations"

in the revision.

**Reviewer #2 Comment 9**

Figure 2: The authors need to revise the explanation about Figure 2 (lines 236-245) and the title of the Figure 2 for helping readers understand the contents in a more concise way.

**Response**

Thanks for your comments. We have added the following descriptions about Figure 2 in Lines

226-231, and the figure caption was also revised accordingly.

A log-linear relationship is assumed to characterize the changes in flood volume with the increase in precipitation intensity as indicated by return periods (Figure 2a) following

Zhou et al. (2012) and Olsen et al. (2015). Generally, more TFV (i.e., system overloading)

is expected with increase in rainfall intensity. In Figure 2b, the TFVs were linked to their specific occurrence probabilities. The total grey area under the curve denotes the TFVs integrated across various return periods and represents the total expected TFVs per year.

The contribution of an individual flood event to total average TFVs is dependent not only on the flood volume, but also its corresponding probability of occurrence.

**Reviewer #2 Comment 10**

Abbreviation: Please carefully use the word abbreviation. For example, Line 236: The TFV -> The total flood volume (TFV); Line 282: climate change -> climate change (CC).

**Response**

Thanks for your comments. We have added the full names for these abbreviations in the revision.

* * *
**Reviewer #2 Comment 11**

Weighted mean imperviousness (WMI): Please define how to calculate the WMI.
* * *
**Response**

Thanks for your suggestion. The WMI of a subcatchment is calculated as the average of impervious factors of all landuse types in the subcatchment, weighted by the area of landuse type as follows:

$$\text{WMI} = \sum_i \text{IF}_i \times A_i / \sum_i A_i$$

Where $\text{IF}_i$ and $A_i$ are the impervious factor and area for land use type i, respectively.

In the revision, we have added the equation and explanations for calculating WMI in Lines 255-

261.

* * *
**Reviewer #2 Comment 12**

Figure 3: The authors need to revise the explanation about Figure 3 (lines 282-292) for
helping readers understand the contents in a more concise way (for example, 1) max; 2)
median; 3) min; 4) multi-model ensemble median). □ 1yr -> 1
* * *
**Response**

Thanks for your comments. In the revision, we have re-plotted figure 3 and added more details in the figure caption. Please check the updated figure below.

[Figure]

**Figure 3** Changes in total flood volume (TFV) as a function of precipitation intensity at various return periods under RCP8.5 scenario without mitigation and adaptation. Red solid line represents the multi-model ensemble median TFV with shaded areas denoting the ensemble range. Red dashed line is the TFV under present condition. Box plots show the relative changes in TFV by 2020–2040 relative to present condition. Box edges illustrate the 25th and 75th percentile, the central mark is the median and whiskers mark the 5th and 95th percentiles.

**Reviewer #2 Comment 13**

Figure 4: Please add the x-axis title

**Response**

Thanks for your suggestion. We have added the x-axis title in the revision.

**Reviewer #2 Comment 14**

Ratio of flood volume (RFV) in Figure 5: Please define how to calculate the RFV.

**Response**

Thanks for your suggestion. The RFV is defined as the ratio of flooded volume from overloaded manholes to input rainfall volume. We have added the definition in the revision (Lines 304-305)

and revised the figure caption accordingly.

> ### Reviewer #2 Comment 15
>
> TFV reduction (%): Please define how to calculate the TFV reduction. In Figures 4 and 7,
> the readers can see the related results, mainly composed of negative and positive numbers,
> respectively. Especially, in Figure 7, the authors need to explain about the negative values
> for the return periods of 100 and 200 years.

**Response**

Thanks for your comments. The TFV reduction is calculated as the percentage difference in TFV

with mitigation or adaptation compared to that without mitigation or adaptation (i.e., $100 *$

$(TFV_{wo} - TFV_w)/TFV_{wo}$). Here, three scenarios including two adaptation strategies (i.e., Pipe and Pipe+LID) and one climate mitigation are considered and the reduced TFV are inter- compared among the three scenarios in Figure 7 to demonstrate that the proposed adaptation strategies are more effective than climate mitigation in reducing future flooded volume.

In Figure 7, the grey, blue and red bars indicate the TFV reductions with two adaptation strategies and climate mitigation, respectively. Positive values indicate that climate mitigation and/or local adaptation are effective in reducing the system overloading, namely the TFVs, and vice versa. It is shown that climate mitigation can lead to reduction of flood volume ranging from 10 to 40% compared to the scenario without mitigation. Importantly, local adaptations are more effective than climate mitigation in reducing flood volume, as indicated by the much larger values of TFV reductions. As noted by the reviewer, there are minor negative values of TFV

reductions for the return period of 100 and 200 years under climate mitigation scenario. The negative value is attributed to the slightly higher increase in precipitation intensity under climate mitigation scenario (RCP26) than the scenario without mitigation (RCP85) as simulated by two of five climate models (i.e., GFDL-ESM2m and NorESM1-M, see Table 1), which translates to a slight increase in flood volume under climate mitigation scenario. This climate internal variability is partly cancelled by other three climate models, thus leading to very minor negative value when considering the multi-model ensemble mean. This calls for the improvement of climate model performance and the use of more climate models (GCMs) to derive more robust climate projections in the future.

In the revision, we have added relevant explanations and discussions in Lines 351-361.

Overall, climate mitigation can lead to a reduction of flood volume by 10-40% compared to the scenario without mitigation. Notably, there are minor negative values of TFV

reductions for the return period of 100 and 200 years under climate mitigation scenario.

The negative value is attributed to the slightly higher increase in precipitation intensity under climate mitigation scenario (i.e. RCP26) than the scenario without mitigation (i.e.

RCP85) simulated by two of five climate models (i.e., GFDL-ESM2m and NorESM1-M, see Table 1), which translated to slightly larger flood volume under climate mitigation scenario. This climate internal variability is partly cancelled by other three climate models, thus leading to very minor negative value by the multi-model ensemble mean. This calls for the use of more climate models (GCMs) to derive more robust projections in the future studies.

**Reviewer #2 Comment 16**

Lines 83-85: There are already many journal papers related to this issue. The authors revise
the 'Introduction' part to emphasize the novelty of this study.

**Response**

Thanks for your comments.  In the revision, we have revised the introduction to emphasize the novelty of this study in Lines 85-93.

However, previous studies on the effects of climate change mitigation and adaptation are typically conducted separately, and it is unclear which strategy is more effective in reducing urban floods. This study aims to advance our understanding on urban floods within the context of change climate, through investigating the benefits of climate change mitigation (by reducing greenhouse gas emissions [GHG]) and local adaptation (by improving drainage systems) in reducing future urban flood volumes in a consistent framework.

**Reviewer #2 Comment 17**

Lines 122-123: Please add the related reference here.

**Response**

Thanks for your comments. We have added the related reference in the revision.

**Reviewer #2 Comment 18**

Lines 134-136: Please add the related reference here.

**Response**

Thanks for your comments. We have added the related reference in the revision.

**Reviewer #2 Comment 19**

Lines 200-202: Please add the related reference here.

**Response**

Thanks for your comments. We have added the related reference in the revision. Please note that this paragraph is moved to supplementary materials following your suggestion #7.

**Reviewer #2 Comment 20**

Lines 251-255: If this explanation is about the adaptation plan for the year 2020, please
clarify the plan for the year 2020 in the 'Introduction' part and this part to help readers
understand the contents in a more concise way.

**Response**

Thanks for your comments. We have added relevant descriptions on this in Lines 113-115.

We then designed two plausible adaptation strategies for the study region and investigated
how much urban flood volume can be reduced with the adapted systems by 2020s. We also
compared the benefits of global-scale climate change mitigation and local adaptation in
reducing urban flood volumes to advance our understanding of the effective measures for
coping with future urban floods.

**Reviewer #2 Comment 21**

Lines 260-261: It seems better to start this paragraph with the explanation about the second
adaptation scenario. The authors need to re-organize this paragraph.

**Response**

Thanks for your comments. We have added the following text in Lines 246-251.

The second adaptation scenario was designed to increase the permeable surfaces (e.g.,
green spaces) and reduce the regional imperviousness in the study region on the basis of
pipe capacity enhancement. This scenario is referred to as the Low Impact Development
(LID) scenario, and it was used to explore the effectiveness of urban green measures, such
as the use of permeable pavements, infiltration trenches, and green roofs. Changes in land
imperviousness in LID scenario have direct impacts on the performance of drainage system
in managing surface runoff.

**Reviewer #2 Comment 22**

Lines 269-272: Please give additional information on this part. It is hard to understand the
meaning of this sentence.

**Response**

Thanks for your comments. We have added more details on this in Lines 255-262.

Due to a lack of detailed information about the permeable soil and coverage rates in the
study region, the effects of these specific measures cannot be modelled individually. Here,

351 we used a simplified approach by altering the subcatchment imperviousness to reflect the
352 combined effects of infiltration-related measures. We derived such information by
353 calculating the difference in land use type and imperviousness between the current and
354 planned city maps using a geographical information system (GIS). Figure 1d shows the
355 difference in weighted mean imperviousness ($WMI = \sum_i (IF_i \times A_i) / \sum_i A_i$, Where $IF_i$ and
356 $A_i$ is the impervious factor and area for land use type $i$, respectively.) for each
357 subcatchment in the current and planned maps, using the commonly applied impervious
358 factors (Pazwash, 2011; Butler and Davies, 2004) for each land use type. The difference in
359 WMI was used to indicate the potential for adaptation based on the city plan.

**Reviewer #2 Comment 23**

Lines 305-308: Please explain the main reason of these results with discussions.

362 **Response**

363 Thanks for your comments. We have added the following explanations and discussions in Lines
364 289-296.

365 Notably, the peak of the total TFV curve was projected to shift from the 1-year event under
366 the RCP8.5 scenario to the 3-year event under the RCP2.6 scenario (Figure 4b), indicating
367 a substantial reduction in the TFVs (especially at the 1-year return period) (Figure 4c). The
368 lower total TFVs under RCP2.6 scenario could be attributed to the smaller magnitude of
369 rainfall intensity than RCP8.5 scenario (Table 1), demonstrating the important role of
370 climate mitigation in reducing urban flood volumes. Overall, climate change mitigation
371 can reduce future flood volumes by 13% compared to the scenario without mitigation, as
372 indicated by the multi-model ensemble median.

**Reviewer #2 Comment 24**

Line 315: Please clarify where the historical flood points are.

375 **Response**

Thanks for your comments. The historical flood points are included in Figure 5a. We have revised the figure and clarified their locations in the figure caption.

[Figure]

**Figure 5** Spatial distribution of overloaded pipelines (red colour) induced by the 3-year (left
column) and 50-year extreme events (right column) without and with adaptations. The total
percentage of overloaded manholes (POM) and ratio of flood volume (RFV) to input rainfall
volume are summarised for each scenario. Historical flood points and local land use, mainly the
traffic network and green spaces, are shown in (a).

e.   Design of adaptation scenarios

In this study, two adaptation scenarios were designed to explore the role of adaptation in reducing urban flood volume within the context of climate change by 2020s. The first scenario adapted was designed to update the drainage system as planned by thelocal water authorities to cope with the designed standard of a 3-year design event. It involved two main improvements of the current drainage system—enhancing the pipeline diameter and expanding the pipe network.

The design was implemented in the SWMM model as shown in Figure 1c. The number of pipelines of the present-day and adapted systems was 323 and 488, with a total pipe length of

251.6 km and 375.4 km, respectively. In the adapted scenarios, the mean pipeline diameter was about 1.73 m, which increased by 53% compared to that of the present-day system.

A variety of site-specific factors, such as the imperviousness of land area in the drainage basin, can also influence the performance of a drainage system in managing surface runoff. The second adaptation scenario was designed to increase the permeable surfaces (e.g., green spaces) and reduce the regional imperviousness in the study region on the basis of pipe capacity enhancement.

This scenario is referred to as the Low Impact Development (LID) scenario, and it was used to explore the effectiveness of urban green measures, such as the use of permeable pavements, infiltration trenches, and green roofs. Changes in land imperviousness in LID scenario have direct impacts on the performance of drainage system in managing surface runoff. Due to a lack of detailed information about the permeable soil and coverage rates in the study region, the effects of these specific measures cannot be modelled individually. Here, we used a simplified approach by altering the subcatchment imperviousness to reflect the combined effects of infiltration-related measures. We derived such information by comparingcalculating the difference in land use type and imperviousness between the current and planned city maps using a geographical information system (GIS

). Figure 1d shows the difference in weighted mean imperviousness ($WMI = \sum_i (IF_i \times A_i) / \sum_i A_i$, Where $IF_i$ and

$A_i$ is the impervious factor and area for land use type $i$, respectively.) for each subcatchment in the current and planned maps, using the commonly applied impervious factors (Pazwash, 2011;

Butler and Davies, 2004) for each land use type. The difference in WMI was used to indicate the potential for adaptation based on the city plan. For example, a subcatchment with higher positive changes in the WMI indicates that the area is planned to have a land use type with lower imperviousness and therefore is assumed to be more suitable for LID planning, and vice versa.

3. Results a. Impacts of future climate change on urban flood volumes

Figure 3 shows the projected climate change (CC) impacts on urban flooding using the present- day drainage system of the near future (i.e., 2020–2040) under the RCP 8.5 scenario. Without climate change mitigation or adaptation, the TFV was projected to increase significantly with the increase of extreme rainfall events for most of investigated return periods (Table 2). Note that the lower bounds for return periods of 1, 3, and 1000 years fall below the current TFV curve due to the decrease in precipitation intensities. Despite the large uncertainty associated with climate projections, in particular with the 1-, 10-, and 1000-year return periods, the poor service performance of the current system in coping with urban flooding was evident. Overall, the urban flood volume was projected to increase by 52% on average by the multi-model ensemble median by 2020–2040; the largest increase (258%) was projected for the 1-year event and the smallest increase (12%) for the 100-year event.

b. Benefits of climate change mitigation in reducing urban flood volumes

Figure 4 shows the comparison of TFVs under the RCP 8.5 scenario (i.e., a business-as-usual scenario) and the RCP 2.6 scenario (i.e., a climate change mitigation scenario). Although large uncertainties exist arising from climate models, it is clear that the simulated TFVs are much smaller under the RCP 2.6 scenario than under the RCP 8.5 scenario, demonstrating the benefits of climate mitigation in reducing local urban flood volumes. Such benefits are especially evident for floods for smaller return periods. For example, an increase of 936 m$^3$ in flood volume is projected with the increase in 1-year extreme rainfall under the business-as-usual climate change scenario (i.e., RCP 8.5), 52% of which would be reduced if climate change mitigation is in place (i.e., under RCP 2.6).

Notably, the peak of the total TFV curve was  projected to shift from the 1-year event under the RCP8.5 scenario to the 3-year event under the RCP2.6 scenario (Figure 4b

), indicating a substantial reduction in the  TFVs (especially at the 1-year return period) (Figure 4c). The lower total TFVs under RCP2.6 scenario could be attributed to the smaller magnitude of rainfall intensity than RCP8.5 scenario (Table 1), demonstrating the important role of climate mitigation in reducing urban flood volumes. Overall, climate change mitigation can reduce future flood volumes by 13% compared to the scenario without mitigation, as indicated by the multi-model ensemble median.

c.   Benefits of adaptation in reducing urban flood volumes

[revised manuscript text omitted]

d.   Climate mitigation versus drainage adaptation

Figure 7 shows the reduced TFVs by climate change mitigation and drainage system adaptation as functions of return period. It is evident that both mitigation and adaptation measures are effective in reducing future urban flood volumes. However, such benefits are projected to weaken gradually with the increase in rainfall intensity (i.e., larger return periods). Overall, climate mitigation can lead to a reduction of flood volume by 10-40% compared to the scenario without mitigation. Notably, there are minor negative values of TFV reductions for the return period of 100 and 200 years under climate mitigation scenario. The negative value is attributed to the slightly higher increase in precipitation intensity under climate mitigation scenario (i.e.

RCP26) than the scenario without mitigation (i.e. RCP85) simulated by two of five climate models (i.e., GFDL-ESM2m and NorESM1-M, see Table 1), which translated to slightly larger flood volume under climate mitigation scenario. This climate internal variability is partly cancelled by other three climate models, thus leading to very minor negative value by the multi- model ensemble mean. This calls for the use of more climate models (GCMs) to derive more robust projections in the future studies.

[revised manuscript text omitted]
. Indeed, a major limitation of this study is lack of assessment of costs and benefits of adaptation measures from the economic perspective. In fact, besides the effectiveness of proposed adaptation measures in reducing flood volume, assessment of the associated economic costs is essential for flood risk management (Rojas et al., 2013; Veith et al., 2003; Hinkel et al.,

2014; Aerts et al., 2014; Ward et al., 2017). For example, Ward et al. (2017) showed that investments in urban flood protections with dykes are not economically attractive everywhere.

Higher investment and maintenance costs may prohibit the implementation of adaptation strategies as proposed in this study. Future efforts should therefore be devoted to building a framework for assessing the costs and benefits of urban flood reduction measures and examine whether the reduced losses are higher than the costs of investments and maintenance of these measures.

Nevertheless, given these limitations, this study stands out from previous climate impact assessment studies of urban flood volumes by having proposed two feasible adaptation strategies and compared their benefits to those from global-scale climate change mitigations through GHG

reductions within a consistent framework. Depending on the progress on data collection and the demands of local authorities, more advanced methods for pipe assessment (e.g., considering the changing pipe conditions), LID measures (detailed modelling of LID control), and two- dimensional surface flooding for assessment of flood damage and risk are planned in a future study to provide a more comprehensive analysis of the adaptation measures.

5.      Summary and Conclusions

The potential impacts of future climate change on current urban drainage systems have received increasing attention during recent decades because of the devastating impacts of urban flooding on the economy and society (Chang et al., 2013; Zhou et al., 2012; Abdellatif et al., 2015).

However, few studies have explored the role of both climate change mitigation and drainage adaptations in coping with urban flooding in a changing climate. This study investigated the performance of a drainage system in a typical city in Northern China in response to various future scenarios. In particular, we assessed the potential changes in urban flood volume and explored the role of both mitigation and adaptation in reducing urban flood volumes in a consistent manner.

Our results show significant increases in urban flood volumes due to increases in precipitation extremes, especially for return periods of less than 10 years. Overall, urban flood volume in the study region is projected to increase by 52% by the multi-model ensemble median in the period of 2020–2040. Such increases in flood volume can be reduced considerably by climate change mitigation through reduction of GHG emissions. For example, the future TFVs under 1-year extreme rainfall events can be reduced by 50% when climate change mitigation is in place.

Besides global-scale climate change mitigation, regional/local adaptation can be implemented to cope with the adverse impacts of future climate change on urban flood volumes. Here, the adaptation measures as designed in this study were demonstrated to be much more effective in reducing future flood volumes than climate change mitigation measures. In general, the reduced flood volumes achieved by adaptation were more than double those achieved by climate change mitigation.

Through a comprehensive investigation of future urban floods, this study provides much-needed insights into urban flood management for similar urban areas in China, most of which are equipped with highly insufficient drainage capacities. By comparing the reduction of flood volume by climate change mitigation (via reduction of GHG emissions) and local adaptation (via improvement of drainage systems), this study highlights the effectiveness of system adaptations in reducing future flood volumes. This has important implications for the research community and decision-makers involved in urban flood management. We emphasise the importance of accounting for both global-scale climate change mitigation and local-scale adaptation in assessing future climate impacts on urban flood volumes within a consistent framework.

**Acknowledgements**

This research was supported by the Public Welfare Research and Ability Construction Project of

Guangdong Province, China (Grant No. 2017A020219003), the Water Conservancy Science and

Technology Innovation Project of Guangdong province, China (Grant No. 201710), the Natural

[revised manuscript text omitted]

Ward, P. J., Jongman, B., Aerts, J. C., Bates, P. D., Botzen, W. J., Loaiza, A. D., ... &
Winsemius, H. C. (2017). A global framework for future costs and benefits of river-flood
protection in urban areas. Nature Climate Change. 7, 642–646.

[revised manuscript text omitted]

**List of Figures**

**Figure 1** Land use of the study region for the year 2010 (a) and 2020 (b). Pipe network
description of current and planned drainage systems (c). Difference in Weighted Mean
Imperviousness (WMI) between year 2010 and 2020 (d).
**Figure 2** Illustration of  total flood volume (TFVs) as a
function of return  periods (a)
and estimation of average total expected TFVs per year (i.e., the
grey area in b) under a stationary drainage system.
**Figure 3**  Changes in total flood volume (TFV) as a function of
precipitation intensity at various return periods under  RCP8.5 scenario without mitigation
and adaptation. Red solid line represents the multi-model ensemble median TFV with shaded
areas denoting the  ensemble range. Red dashed line is the TFV under present condition.
Box plots show the relative changes in TFV by 2020–2040 relative to present condition. Box
edges illustrate the 25th and 75th percentile, the central mark is the median and whiskers mark
the 5th and 95th percentiles.
**Figure 4** Comparison of (a) flood volume, (b) total TFVs (i.e., the piece-wise integral of flood
volume versus the expected frequency with changes in precipitation intensity of various return
periods under RCP8.5 (blue) and RCP2.6 (red). (c) is  the TFV reduction
calculated as the percentage difference in TFVs under RCP2.6 compared to RCP8.5 (i.e.,
benefits of climate mitigation)  at various return periods.
**Figure 5** Spatial distribution of overloaded pipelines (red colour) induced by the 3-year (left
column) and 50-year extreme events (right column) without and with adaptations. The total
percentage of overloaded manholes (POM) and ratio of flood volume (RFV) to input rainfall
volume are summarised for each scenario.  Historical flood points and local land
use, mainly the traffic network and green spaces, are shown in
(a).
**Figure 6** Future changes in flood volumes (CTFVs) relative to historical conditions under the
current drainage system (yellow) and two adaptation scenarios (i.e., Pipe in red and Pipe+LID in
green) at various return periods.
**Figure 7** Comparison of benefits of climate mitigation and two adaptation strategies in reducing
urban flood volumes with changes in precipitation intensities for various return periods, and with
related variations (boundary bars) as a result of uncertainty arising from local soil conditions.

[Figure]

**Figure 1** Land use of the study region for the year 2010 (a) and 2020 (b). Pipe network
description of current and planned drainage systems (c). Difference in Weighted Mean
Imperviousness (WMI) between year 2010 and 2020 (d).

[Figure]

**Figure 2** Illustration of  flood volume  (TFVs)
as a function of return
periods (a) and estimation of average total expected TFVs per year
(i.e., the grey area in b) under a stationary drainage system.

[Figure]

[Figure]

**Figure 3**  Changes in total flood volume (TFV) as a function of
precipitation intensity at various return periods under  RCP8.5 scenario without mitigation
and adaptation. Red solid line represents the multi-model ensemble median TFV with shaded
areas denoting the ensemble range. Red dashed line is the TFV under present condition.
Box plots show the relative changes in TFV by 2020–2040 relative to present condition. Box
edges illustrate the 25th and 75th percentile, the central mark is the median and whiskers mark
the 5th and 95th percentiles.

[Figure]

[Figure]

**Figure 4** Comparison of (a) flood volume, (b) total TFVs (i.e., the piece-wise integral of flood
volume versus the expected frequency with changes in precipitation intensity of various return
periods under RCP8.5 (blue) and RCP2.6 (red). (c) is  the  TFV reduction
calculated as the percentage difference in TFVs under RCP2.6 compared to RCP8.5 (i.e.,
benefits of climate mitigation)  at various return periods.

[Figure]

[Figure]

**Figure 5** Spatial distribution of overloaded pipelines (red colour) induced by the 3-year (left
column) and 50-year extreme events (right column) without and with adaptations. The total
percentage of overloaded manholes (POM) and ratio of flood volume (RFV) to input rainfall
volume are summarised for each scenario. Descriptions of Historical flood points and local land
use, mainly the traffic network and green spaces, are provided as the background imageshown in
(a).

[Figure]

**Figure 6** Future changes in flood volumes (CTFVs) relative to historical conditions under the
current drainage system (yellow) and two adaptation scenarios (i.e., Pipe in red and Pipe+LID in
green) at various return periods.

[Figure]

**Figure 7** Comparison of benefits of climate mitigation and two adaptation strategies in reducing
urban flood volumes with changes in precipitation intensities for various return periods, and with
related variations (boundary bars) as a result of uncertainty arising from local soil conditions.